# Symmetry-Informed Geometric Representation for Molecules, Proteins, and Crystalline Materials

**Shengchao Liu**[1,2], **Weitao Du**[3], **Yanjing Li**[4], **Zhuoxinran Li**[5], **Zhiling Zheng**[6], **Chenru Duan**[7], **Zhiming Ma**[3], **Omar Yaghi**[6], **Anima Anandkumar**[8], **Christian Borgs**[6], **Jennifer Chayes**[6], **Hongyu Guo**[9], **Jian Tang**[1,10,11]

[1]Mila - Québec Artificial Intelligence Institute  [2]Université de Montréal
[3]University of Chinese Academy of Sciences  [4]Carnegie Mellon University
[5]University of Toronto  [6]University of California, Berkeley  [7]Massachusetts Institute of Technology
[8]California Institute of Technology  [9]National Research Council Canada
[10]HEC Montréal  [11]CIFAR AI Chair
#Correspondence: *shengchao.liu@umontreal.ca, jian.tang@hec.ca*

## Abstract

Artificial intelligence for scientific discovery has recently generated significant interest within the machine learning and scientific communities, particularly in the domains of chemistry, biology, and material discovery. For these scientific problems, molecules serve as the fundamental building blocks, and machine learning has emerged as a highly effective and powerful tool for modeling their geometric structures. Nevertheless, due to the rapidly evolving process of the field and the knowledge gap between science (*e.g.*, physics, chemistry, & biology) and machine learning communities, a benchmarking study on geometrical representation for such data has not been conducted. To address such an issue, in this paper, we first provide a unified view of the current symmetry-informed geometric methods, classifying them into three main categories: invariance, equivariance with spherical frame basis, and equivariance with vector frame basis. Then we propose a platform, coined Geom3D, which enables benchmarking the effectiveness of geometric strategies. Geom3D contains 16 advanced symmetry-informed geometric representation models and 14 geometric pretraining methods over 52 diverse tasks, including small molecules, proteins, and crystalline materials. We hope that Geom3D can, on the one hand, eliminate barriers for machine learning researchers interested in exploring scientific problems; and, on the other hand, provide valuable guidance for researchers in computational chemistry, structural biology, and materials science, aiding in the informed selection of representation techniques for specific applications. The source code is available on the GitHub repository.

## 1  Introduction

Artificial intelligence (AI) for molecule discovery has recently seen many developments, including small molecular property prediction [13, 17, 23, 38, 66, 78, 101, 103, 104, 119, 130, 132, 135], small molecule design and optimization [6, 54, 57, 85, 137], small molecule reaction and retrosynthesis [40, 111, 116], protein property prediction [27, 141], protein folding and inverse folding [48, 64, 92], protein design [15, 41, 46, 88, 91], and crystalline material design [33, 125, 128]. One of the most fundamental building blocks for these tasks is the geometric structure of molecules. Exploring effective methods for robust representation learning to leverage such geometric information fully remains an open challenge that interests both machine learning (ML) and science researchers.

To this end, symmetry-informed geometric representation [1] has emerged as a promising approach. By leveraging physical principles (*i.e.*, group theory for depicting symmetric particles) into spatial

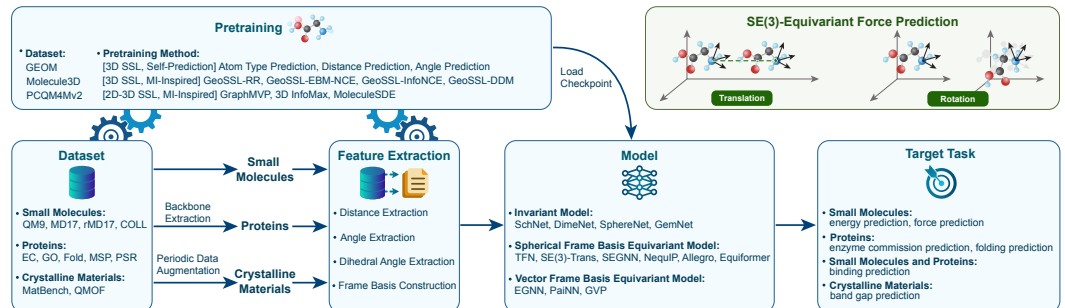

Figure 1: Pipeline for Geom3D, including dataset preprocessing, feature extraction, geometric pretraining and representation, and target tasks. We additionally demonstrate the **SE(3)-equivariant force prediction task**.

representation, they facilitate a more robust representation of small molecules, proteins, and crystalline materials. Nevertheless, pursuing geometric learning research is still challenging due to its evolving nature and the knowledge gap between science (*e.g.*, physics) and machine learning communities. These factors contribute to a substantial barrier for machine learning researchers to investigate scientific problems and hinder efforts to reproduce results consistently. To overcome this, we introduce Geom3D, a benchmarking of the geometric representation with four advantages, as follows. [1]

**(1) A unified and novel aspect in understanding symmetry-informed geometric models.**
The molecule geometry needs to satisfy certain physical constraints regarding the 3D Euclidean space. For instance, the molecules' force needs to be equivariant to translation and rotation (see SE(3)-equivariance in Fig. 1). In this work, we classify the geometric methods into three categories: *invariant* model, SE(3)-equivariant model with *spherical frame basis* and *vector frame basis*. The invariant models only consider features that are constant w.r.t. the SE(3) group, while the two families of equivariant models can be further unified using the *frame basis* to capture equivariant symmetry. An illustration of three categories is in Fig. 2. Building equivariant models on the *frame basis* provides a novel and unified view of understanding geometric models and paves the way for intriguing more ML researchers to explore scientific problems.

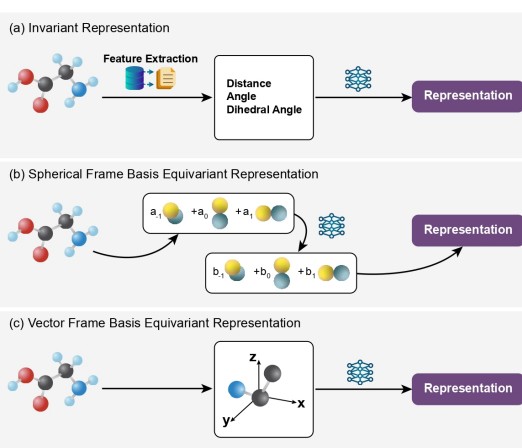

Figure 2: Three categories of geometric modules. (a) Invariant models only consider type-0 features. Equivariant models use either (b) spherical harmonics frames or (c) vector frames by projecting the coordinate vectors.

**(2) A unified platform for various scientific domains.** There exist multiple platforms and tools for molecule discovery, but they are (1) mainly focusing on molecule's 2D graph representation [77, 102, 145]; (2) using 3D geometry with customized data structures or APIs [3, 105]; or (3) covering only a few geometric models [76]. Thus, it is necessary to have a platform benchmarking the geometric models, especially for researchers interested in solving scientific problems. In this work, we propose Geom3D, a geometric modeling framework based on PyTorch Geometric (PyG) [31], one of the most widely-used platforms for graph representation learning. Geom3D benchmarks 16 geometric models on solving 52 scientific tasks, and these tasks include the three most fundamental molecule types: small molecules, proteins, and crystalline materials. Each of them requires distinct domain-specific preprocessing steps, *e.g.*, crystalline materials molecules possess periodic structures and thus need a particular periodic data augmentation. By leveraging such a unified framework, Geom3D serves as a comprehensive benchmarking tool, facilitating effective and consistent analysis components to interpret the existing geometric representation functions in a fair and convenient comparison setting.

**(3) A framework for a wider range of ML tasks.** The geometric models in Geom3D can serve as a building block for exploring extensive ML tasks, including but not limited to studying the molecule dynamic simulation and scrutinizing the transfer learning effect on molecule geometry. For example, pre-

---

[1]In what follows, we may use "molecule" to refer to "small molecule" for brevity.

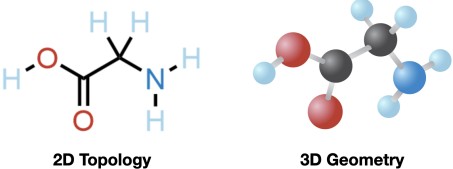
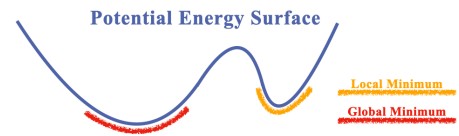

**2D Topology**  **3D Geometry**

(a) An example of small molecule structure.

(b) An illustration of the potential energy surface.

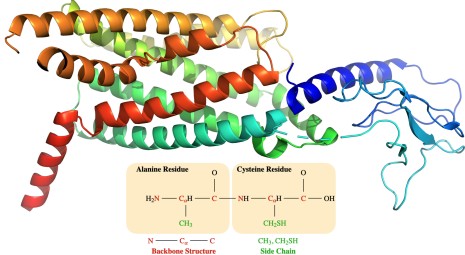
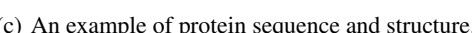
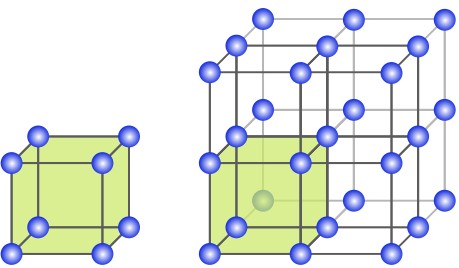

(c) An example of protein sequence and structure.

(d) An example of crystalline material.

Figure 3: Fig. 3(a) illustrates 2D topology and 3D conformation for molecule Glycine. Fig. 3(c) displays the 3D structure of protein. Fig. 3(d) shows a simple cubic crystal of the element Po. Fig. 3(b) is a demo of PES.

training is an important strategy to quickly transfer knowledge to target tasks, and recent works explore geometric pretraining on 3D conformations (including supervised and self-supervised) [59, 80, 136] and multi-modality pretraining on 2D topology and 3D geometry [30, 79, 86]. Other transfer learning venues include multi-task learning [82, 84] and out-of-distribution or domain adaptation [58, 133, 134], yet no geometry information has been utilized. All of these directions are promising for future exploration, and Geom3D serves as an auxiliary tool to accomplish them. For example, as will be shown in Sec. 4, we leverage Geom3D to effectively evaluate 14 pretraining methods with benchmarks.

**(4) A framework for exploring data preprocessing and optimization tricks.** When comparing different symmetry-informed geometric models, we find that in addition to the model architecture, there are two important factors affecting the performance: the data preprocessing (*e.g.*, energy and force rescaling and shift) and optimization methods (*e.g.*, learning rate, learning rate schedule, number of epochs, random seeds). In this work, we explore the effect of four preprocessing tricks and around 2-10 optimization hyperparameters for each model and task. In general, we observe that each model may benefit differently in different tasks regarding the preprocessing and optimization tricks. However, data normalization is found to help improve performance hugely in most cases. We believe that Geom3D is an effective tool for exploring and understanding various engineering tricks.

## 2 Data Structures for Geometric Data

**Small molecule 3D conformation.** Molecules are sets of points in the 3D Euclidean space, and they move in a dynamic motion, as known as the potential energy surface (PES). The region with the lowest energy corresponds to the most stable state for molecules, and molecules at these positions are called **conformations**, as illustrated in Fig. 3(b). For notation, we mark each 3D molecular graph as $g = (X, R)$, where $X$ and $R$ are for the atom types and positions, respectively.

**Crystalline material with periodic structure.** The crystalline materials or extended chemical structures possess a characteristic known as periodicity: their atomic or molecular arrangement repeats in a predictable and consistent pattern across all three spatial dimensions. This is the key aspect that differentiates them from small molecules. In Fig. 3(d), we show an original unit cell (marked in green) that can repeatedly compose the crystal structure along the lattice. To model such a periodic structure, we adopt the data augmentation from CGCNN [129]: for each original unit cell, we shift it along the lattice in three dimensions and connect edges within a cutoff value (hyperparameter). For more details on the two augmentation variants, please check Appendix A.

**Protein with backbone structure.** Protein structures can be classified into four primary levels, and the primary structure represents the linear arrangement of *amino acids*, and each amino acid is a molecule consisting of atoms. Geometric methods mainly focus on the tertiary structure, *i.e.*, the 3D geometry of each atom, encompassing the complete organization of a single protein. However,

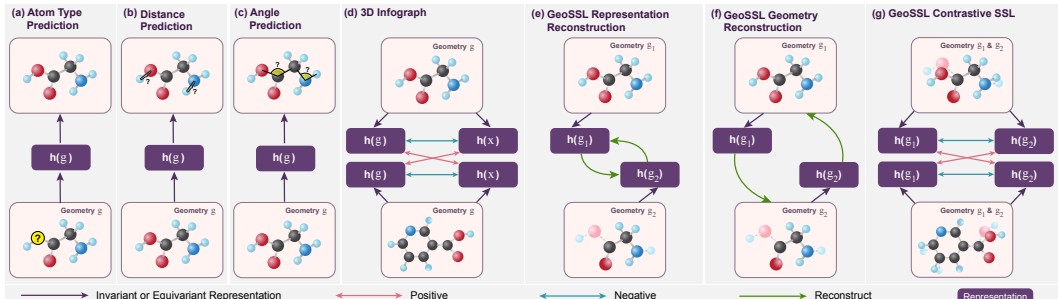

Figure 4: Pipelines for seven single-modal geometric pretraining methods. (a-c) conduct self-prediction. (d) maximizes the MI between nodes and graphs. (e-g) are GeoSSL, maximizing the MI between views $g_1$ and $g_2$.

atom-level modeling for proteins is consuming due to the large volume of atoms and the GPU memory limit. One solution is modeling each amino acid's *backbone structure*. The backbone structure of each amino acid is $N - C_\alpha - C$, and the $C_\alpha$ is bonded to the side chain. 20 common types of side chains corresponding to 20 amino acids, as illustrated in Fig. 3(c). Thus, modeling the backbone structure can balance the computational efficiency and the key geometric information.

## 3 Symmetry-Informed Geometric Representation

### 3.1 Group Symmetry and Equivariance

Symmetry means the object remains invariant after certain transformations [127], and it is everywhere on Earth, such as in animals, plants, and molecules. Formally, the set of all symmetric transformations satisfies the axioms of a group. Therefore, the group theory and its representation theory are common tools to depict such physical symmetry. **Group** is a set $G$ equipped with a group product $\times$ satisfying:

$$(1) \exists e \in G, \ \boldsymbol{a} \times \boldsymbol{e} = \boldsymbol{e} \times \boldsymbol{a}, \forall \boldsymbol{a} \in G; \quad (2) \ \boldsymbol{a} \times \boldsymbol{a}^{-1} = \boldsymbol{a}^{-1} \times \boldsymbol{a} = \boldsymbol{e}; \quad (3) \ \boldsymbol{a} \times (\boldsymbol{b} \times \boldsymbol{c}) = \boldsymbol{a} \times \boldsymbol{b} \times \boldsymbol{c}. \quad (1)$$

**Group representation** is a mapping from the group $G$ to the group of linear transformations of a vector space $X$ with dimension $d$ (see [138] for more rigorous definition):

$$\rho_X(\cdot) : G \to \mathbb{R}^{d \times d} \qquad \text{s.t.} \quad \rho(\boldsymbol{e}) = 1 \ \wedge \ \rho_X(\boldsymbol{a})\rho_X(\boldsymbol{b}) = \rho_X(\boldsymbol{a} \times \boldsymbol{b}), \ \forall \boldsymbol{a}, \boldsymbol{b} \in G. \quad (2)$$

During modeling, the $X$ space can be the input 3D Euclidean space, the equivariant vector space in the intermediate layers, or the output force space. This enables the definition of equivariance as below.

**Equivariance** is the property for the geometric modeling function $f : X \to Y$ as:

$$f(\rho_X(\boldsymbol{a})\boldsymbol{x}) = \rho_Y(\boldsymbol{a})f(\boldsymbol{x}), \ \ \forall \boldsymbol{a} \in G, \boldsymbol{x} \in X. \quad (3)$$

As displayed in Fig. 1, for molecule geometric modeling, the property should be rotation-equivariant and translation-equivariant (*i.e.*, SE(3)-equivariant). More concretely, $\rho_X(\boldsymbol{a})$ and $\rho_Y(\boldsymbol{a})$ are the SE(3) group representations on the input (*e.g.*, atom coordinates) and output space (*e.g.*, force space), respectively. SE(3)-equivariant modeling in Eq. (3) is essentially saying that the designed deep learning model $f$ is modeling the whole transformation trajectory on the molecule conformations, and the output is the transformed $\hat{y}$ accordingly. Further, we want to highlight that, in addition to the network architecture or representation function, the input features can also be represented as an equivariant feature mapping from the 3D mesh to $\mathbb{R}^{\tilde{d}}$ [11], where $\tilde{d}$ depends on input data, *e.g.*, $\tilde{d} = 1$ (for atom type dimension) + 3 (for atom coordinate dimension) on small molecules. Such features are called steerable features in [5, 11] when only considering the subgroup SO(3)-equivariance.

**Invariance** is a special type of equivariance, defined as:

$$f(\rho_X(\boldsymbol{a})\boldsymbol{x}) = f(\boldsymbol{x}), \ \ \forall \boldsymbol{a} \in G, \boldsymbol{x} \in X, \quad (4)$$

with $\rho_Y(\boldsymbol{a})$ as the identity $\forall \boldsymbol{a} \in G$. The group representation helps define the equivariance condition for $f$ to follow. Then, the question boils down to how to design such an equivariant $f$. In the following, we will discuss geometric modelings from a novel and unified perspective using the frame. In the next sections, we will provide a novel and unified aspect of understanding the advanced geometric representation and pretraining methods using the frame basis (details in Appendix H).

## 3.2 Invariant Geometric Representation Learning

One simple way of achieving SE(3) group symmetry is invariant modeling. It means the geometric model only considers the type-0 features [112], *i.e.*, features that are invariant with respect to rotation and translation. Existing works have been adopting the invariant features for modeling, including pairwise distance (SchNet [109]), bond angles (DimeNet [68]), and torsion angles (SphereNet [89] and GemNet [67]). Note that the torsion angles are angles between two planes defined by pairwise bonds.

## 3.3 Equivariant Geometric Representation Learning

Invariant modeling only captures the type-0 features. However, equivariant modeling of higher-order particles may bring in extra expressiveness. For example, the elementary particles in high energy physics [98] inherit higher order symmetries in the sense of SO(3) representation theory, which makes the equivariant modeling necessary. Such higher-order particles include type-1 features like coordinates and forces in molecular conformation. There are many approaches to design such SE(3)-equivariant model satisfying Eq. (3). There are two main venues, as will be discussed below.

**Spherical Frame Basis.** This research line utilizes the irreducible representations [37] for building SO(3)-equivariant representations, and the first work is TFN [112]. Its main idea is to project the 3D Euclidean coordinates into the spherical harmonics space, which transforms equivariantly according to the irreducible representations of SO(3), and the translation-equivariant can be trivially guaranteed using the relative coordinates. Following this, there have been variants combining it with the attention module (Equiformer [73]) or with more expressive network architectures (SEGNN [4], Allegro [95]).

**Vector Frame Basis.** An alternative philosophy of equivariant modeling utilizes the vector (in physics) frame basis. It constructs three vectors bases, serving as a reference frame to help locate the vectors in each corresponding local environment. Works along this line for molecule discovery include DeePMD [140] for dynamics simulation, 3D-EMGP [59] and MoleculeSDE [79] for geometric pretraining, and ClofNet [20] for conformation generation. For macromolecules like protein, the equivariant vector frame has been used for protein design (StructTrans [53]) and protein folding (AlphaFold2 [64]). We also want to highlight that, from a mathematical perspective, equivariance and invariance can be transformed to each other by the scalarization technique. Please check [49] for details.

The spherical frame basis can be easily extended to higher-order particles, yet it may suffer from the high computational cost. On the other hand, the vector frame basis is specifically designed for the 3D point clouds; thus, it is more efficient but cannot generalize to higher-order particles. Meanwhile, we would like to acknowledge other equivariant modeling paradigms, including using orbital features [99] and elevating 3D Euclidean space to SE(3) group [32, 52]. Please check Appendix F for details.

## 3.4 Geometric Pretraining

Recent studies have started to explore **single-modal of geometric pretraining** on molecules. The GeoSSL paper [80] covers a wide range of geometric pretraining algorithms. The type prediction, distance prediction, and angle prediction predict the masked atom type, pairwise distance, and bond angle, respectively. The 3D InfoGraph predicts whether the node- and graph-level 3D representation are for the same molecule. GeoSSL is a novel geometric pretraining paradigm that maximizes the mutual information (MI) between the original conformation $g_1$ and augmented conformation $g_2$, where $g_2$ is obtained by adding small perturbations to $g_1$. RR, InfoNCE, and EBM-NCE optimize the objective in the latent representation space, either generative or contrastive. GeoSSL-DDM [80, 136] optimizes the same objective function using denoising score matching. 3D-EMGP [60] has the same strategy and utilizes an equivariant module to denoise the 3D noise directly. Another research line is the **multi-modal of topological and geometric pretraining**. GraphMVP [86] first proposes one contrastive objective (EBM-NCE) and one generative objective (VRR) to optimize the MI between the 2D topologies and 3D geometries in the representation space. 3D InfoMax [114] is a special case of GraphMVP, with the contrastive part only. MoleculeSDE [79] extends GraphMVP by introducing two SDE models for solving the 2D and 3D reconstruction. We illustrate these algorithms in Figs. 4 and 8.

## 3.5 Discussion: Reflection-antisymmetric in Geometric Learning

Till now, we have discussed the SE(3)-equivariance, *i.e.*, the translation and rotation equivariance. As highlighted in the recent work [61, 79], the molecules needlessly satisfy the reflection-equivariant,

Table 1: Results of 26 models on 12 quantum mechanics prediction tasks in QM9, with 110K for training, 10K for validation, and 11K for testing. The task unit is specified, and the evaluation is the mean absolute error (MAE).

| Featurization | Model | $\alpha\downarrow$ $\alpha_0^3$ | $\nabla\mathcal{E}\downarrow$ meV | $\mathcal{E}_{HOMO}\downarrow$ meV | $\mathcal{E}_{LUMO}\downarrow$ meV | $\mu\downarrow$ D | $C_v\downarrow$ $\frac{cal}{mol\cdot K}$ | $G\downarrow$ meV | $H\downarrow$ meV | $R^2\downarrow$ $\alpha_0^2$ | $U\downarrow$ meV | $U_0\downarrow$ meV | ZPVE$\downarrow$ meV |
|---|---|---|---|---|---|---|---|---|---|---|---|---|---|
| 1D FPs | MLP | 2.231 | 196.72 | 131.27 | 164.94 | 0.526 | 0.919 | 2158.64 | 2358.23 | 68.621 | 2340.61 | 2314.77 | 155.921 |
| | RF | 3.801 | 207.02 | 165.72 | 183.04 | 0.534 | 1.485 | 3391.79 | 3729.94 | 94.512 | 3705.75 | 3678.25 | 253.132 |
| | XGB | 2.748 | 199.71 | 139.88 | 165.43 | 0.516 | 1.062 | 2563.93 | 2804.27 | 82.959 | 2786.28 | 2769.29 | 180.989 |
| 1D SMILES | CNN | 0.364 | 165.22 | 124.65 | 114.81 | 0.566 | 0.173 | 156.66 | 170.59 | 20.403 | 166.18 | 169.89 | 10.070 |
| | BERT | 0.313 | 117.50 | 84.93 | 98.88 | 0.446 | 0.176 | 170.01 | 183.43 | 18.002 | 183.84 | 188.60 | 13.410 |
| 1D SELFIES | CNN | 0.345 | 157.04 | 115.51 | 113.00 | 0.499 | 0.168 | 136.42 | 146.56 | 20.080 | 143.00 | 140.01 | 10.149 |
| | BERT | 0.348 | 123.11 | 91.15 | 90.80 | 0.461 | 0.203 | 168.20 | 187.50 | 19.125 | 204.93 | 195.98 | 17.328 |
| 2D Graph | GCN | 1.338 | 145.82 | 96.21 | 106.66 | 0.434 | 0.526 | 1198.12 | 1291.57 | 37.585 | 1281.03 | 1303.39 | 85.103 |
| | ENN-S2S | 1.401 | 270.59 | 129.18 | 132.84 | 0.577 | 0.760 | 1487.21 | 955.24 | 34.609 | 1800.79 | 1521.32 | 51.226 |
| | GraphSAGE | 1.601 | 131.45 | 88.78 | 93.21 | 0.402 | 0.544 | 1473.42 | 1617.73 | 38.112 | 1553.01 | 1565.65 | 95.344 |
| | GAT | 1.132 | 135.90 | 94.70 | 98.52 | 0.406 | 0.291 | 911.82 | 991.31 | 26.583 | 1161.29 | 592.67 | 55.061 |
| | GIN | 1.165 | 175.82 | 90.66 | 110.74 | 0.539 | 0.691 | 848.24 | 1090.36 | 35.110 | 1498.23 | 1364.18 | 108.331 |
| | D-MPNN | 0.568 | 118.42 | 85.01 | 86.20 | 0.441 | 0.241 | 423.14 | 458.39 | 24.816 | 470.01 | 445.91 | 29.291 |
| | PNA | 0.681 | 148.88 | 88.72 | 97.31 | 0.361 | 0.409 | 664.98 | 692.74 | 23.855 | 616.70 | 694.92 | 57.217 |
| | Graphormer | 2.836 | 79.27 | 54.24 | 52.42 | 0.330 | 0.080 | 2066.28 | 2546.01 | 131.158 | 2229.88 | 2525.51 | 144.595 |
| | AWARE | 0.297 | 144.91 | 133.89 | 98.86 | 0.602 | 0.129 | 86.62 | 94.47 | 22.180 | 93.59 | 95.73 | 5.275 |
| | GraphGPS | 0.209 | 75.98 | 54.75 | 54.53 | 0.288 | 0.089 | 528.50 | 693.19 | 12.488 | 296.00 | 411.16 | 49.888 |
| 3D Graph | SchNet | 0.060 | 44.13 | 27.64 | 22.55 | 0.028 | 0.031 | 14.19 | 14.05 | 0.133 | 13.93 | 13.27 | 1.749 |
| | DimeNet++ | 0.044 | 36.22 | 20.01 | 16.66 | 0.028 | **0.022** | **7.45** | **6.14** | 0.323 | **6.33** | 7.18 | **_1.118_** |
| | SE(3)-Trans | 0.137 | 56.52 | 34.65 | 34.41 | 0.050 | 0.063 | 65.28 | 70.70 | 1.747 | 68.92 | 68.88 | 5.428 |
| | EGNN | 0.062 | 49.56 | 30.08 | 24.98 | 0.029 | 0.030 | 10.01 | 9.14 | **0.089** | 9.28 | 9.08 | 1.519 |
| | PaiNN | 0.049 | 42.73 | 24.46 | 20.16 | 0.016 | 0.025 | 8.43 | 7.88 | 0.169 | 8.18 | 7.63 | 1.419 |
| | GemNet-T | **_0.041_** | 35.46 | 17.85 | _15.86_ | 0.021 | 0.023 | 7.61 | 7.08 | 0.271 | 6.42 | **_5.88_** | 1.232 |
| | SphereNet | 0.047 | 38.93 | 21.45 | 18.25 | 0.027 | 0.025 | 8.16 | 13.68 | 0.288 | 6.77 | 7.43 | 1.295 |
| | SEGNN | 0.048 | 33.61 | _17.66_ | 17.01 | 0.021 | 0.026 | 11.60 | 12.45 | 0.404 | 11.29 | 12.20 | 1.590 |
| | Allegro | 0.097 | 102.44 | 61.86 | 63.17 | 0.176 | 0.032 | 42.08 | 44.96 | 1.977 | 44.64 | 44.43 | 2.949 |
| | NequIP | 0.066 | 61.94 | 42.00 | 31.64 | 0.036 | 0.028 | 22.08 | 23.36 | 0.415 | 23.23 | 23.02 | 1.899 |
| | Equiformer | 0.051 | **_33.46_** | 17.93 | 16.85 | **_0.015_** | 0.023 | 14.49 | 14.60 | 0.433 | 14.88 | 13.78 | 2.342 |

but instead, they should be reflection-antisymmetric [79]. One classic example is that the energy of small molecules is reflection-antisymmetric in a binding system. Each of the two equivariant categories discussed in Sec. 3.3 can solve this problem easily. The spherical frame basis can achieve this by adding the reflection into the Wigner-D matrix [4], and the vector frame basis can accomplish this using the cross-product during frame construction [79].

# 4 Geometric Datasets and Benchmarks

In Sec. 3, we introduce a novel aspect for understanding symmetry-informed geometric models. In this section, we discuss utilizing Geom3D framework for benchmarking 16 geometric models over 52 tasks. For the detailed dataset acquisitions and task specifications (*e.g.*, *dataset size*, *splitting*, and *task unit*), please check Appendix B. Geom3D also covers 7 1D models and 10 2D graph neural networks (GNNs) and benchmarks the 14 pretraining algorithms to learn a robust geometric representation. Additionally, we want to highlight Geom3D enables exploration of important data preprocessing and optimization tricks for performance improvement, as will be introduced next.

## 4.1 Small Molecules: QM9

QM9 [100] is a dataset consisting of 134K molecules, each with up to 9 heavy atoms. It includes 12 tasks that are related to the quantum properties. For example, U0 and U298 are the internal energies at temperatures of 0K and 298.15K, respectively. On the QM9 dataset, we can easily get the 1D descriptors (Fingerprints/FPs [106], SMILES [126], SELFIES [70]), 2D topology, and 3D conformation. This enables us to build models on each of them respectively: (1) We benchmark 7 models on 1D descriptors, including multi-layer perception (MLP), random forest (RF), XGBoost (SGB), convolution neural networks (CNN), and BERT [18]. (2) We benchmark 10 2D GNN models on the molecular topology, including GCN [23, 66], ENN-S2S [38], GraphSAGE [43], GAT [119], GIN [130], D-MPNN [132], PNA [13], Graphormer [135], AWARE [17], GraphGPS [101]. (3) We benchmark 11 3D geometric models on the molecular conformation, including SchNet [109], DimeNet++ [68], SE(3)-Trans [35], EGNN [108], PaiNN [110], GemNet-T [67], SphereNet [89], SEGNN [4], Allegro [95], NequIP [3], Equiformer [73]. The evaluation metric is the mean absolute error.

The results of these 28 models are in Table 1, and two important insights are observed: (1) There is no one universally best geometric model, yet DimeNet++, PaiNN, GemNet, and Equiformer perform well in most tasks. However, PaiNN takes less than 20 GPU hours, and the other three models take up to 5 GPU days per task. (2) The geometric conformation is important for quantum property prediction. The performance of 3D models is better than all the 1D and 2D models *by orders of magnitudes*.

Table 2: Results on 6 energy ($\frac{kcal}{mol}$) and force ($\frac{kcal}{mol\cdot\text{Å}}$) prediction tasks in MD17 and rMD17 (w/o normalization), and the metric is the mean absolute error (MAE). The data split and complete results are in Appendices B and I.

| Model | Energy /Force | MD17 | | | | | | rMD17 | | | | | |
|---|---|---|---|---|---|---|---|---|---|---|---|---|---|
| | | Aspirin ↓ | Ethanol ↓ | Malonaldehyde ↓ | Naphthalene ↓ | Salicylic ↓ | Toluene ↓ | Aspirin ↓ | Ethanol ↓ | Malonaldehyde ↓ | Naphthalene ↓ | Salicylic ↓ | Toluene ↓ |
| SchNet | Energy | 0.475 | 0.109 | 0.300 | 0.167 | 0.212 | 0.149 | 0.534 | 1.757 | 0.260 | 0.124 | 2.618 | 0.119 |
| | Force | 1.203 | 0.386 | 0.794 | 0.587 | 0.826 | 0.568 | 1.243 | 0.449 | 0.862 | 0.587 | 0.878 | 0.574 |
| DimeNet++ | Energy | 4.168 | 1.238 | 1.385 | 1.846 | 2.445 | 1.484 | 2.438 | 1.456 | 2.317 | 1.648 | 1.555 | 1.210 |
| | Force | 7.212 | 0.753 | 1.842 | 8.515 | 1.752 | 1.037 | 2.009 | 1.213 | 7.029 | 0.629 | 0.934 | 0.921 |
| EGNN | Energy | 17.892 | 0.436 | 0.896 | 12.177 | 6.964 | 4.051 | 17.35 | 0.402 | 0.534 | 12.164 | 7.794 | 15.021 |
| | Force | 3.042 | 0.924 | 1.566 | 1.136 | 1.177 | 1.202 | 3.825 | 0.989 | 1.334 | 1.183 | 1.571 | 1.165 |
| PaiNN | Energy | 27.626 | **0.063** | **0.102** | 0.622 | 0.371 | 0.165 | 30.156 | 1.17 | **0.070** | 5.297 | 5.219 | 0.045 |
| | Force | 0.572 | 0.230 | 0.338 | 0.132 | 0.288 | 0.141 | 0.573 | 0.316 | 0.377 | 0.161 | 0.321 | 0.231 |
| GemNet-T | Energy | 0.684 | 4.598 | 4.966 | 0.482 | **0.128** | **0.098** | 5.389 | 1.615 | 9.496 | 0.031 | 21.411 | 959.745 |
| | Force | 0.558 | 0.219 | 0.433 | 0.212 | 0.326 | 0.174 | 0.555 | 0.233 | 0.337 | 0.154 | 0.371 | 0.400 |
| SphereNet | Energy | 0.244 | 1.603 | 1.559 | 0.167 | 0.188 | 0.113 | 0.304 | 0.072 | 0.138 | 0.093 | 0.771 | 20.479 |
| | Force | 0.546 | 0.168 | 0.667 | 0.315 | 0.479 | 0.194 | 0.622 | 0.217 | 0.500 | 0.279 | 2.088 | 0.254 |
| SEGNN | Energy | 17.774 | 0.151 | 0.247 | 0.655 | 2.173 | 0.624 | 15.721 | 0.13 | 0.182 | 1.11 | 1.494 | 0.814 |
| | Force | 9.003 | 0.893 | 1.249 | 0.895 | 2.220 | 1.138 | 8.549 | 0.846 | 1.185 | 0.926 | 2.056 | 1.241 |
| NequIP | Energy | 8.333 | 0.971 | 2.293 | 1.032 | 2.952 | 1.303 | 9.618 | 0.936 | 2.313 | 2.089 | 3.302 | 1.306 |
| | Force | 23.769 | 5.832 | 12.099 | 5.247 | 14.048 | 6.8 | 22.904 | 6.027 | 12.372 | 5.529 | 15.693 | 7.094 |
| Allegro | Energy | 1.138 | 0.258 | 1.33 | 0.824 | 1.114 | 0.441 | 1.366 | 1.002 | 0.417 | 1.756 | 1.035 | 0.437 |
| | Force | 3.405 | 1.412 | 4.191 | 3.743 | 4.934 | 1.968 | 3.186 | 2.799 | 2.125 | 3.815 | 4.781 | 2.048 |
| Equiformer | Energy | 0.308 | 0.096 | 0.183 | **0.097** | 0.189 | 0.209 | 0.375 | 0.064 | 0.085 | **0.069** | **0.143** | **0.104** |
| | Force | **0.286** | **0.142** | **0.230** | **0.068** | **0.200** | **0.080** | **0.305** | **0.162** | **0.240** | **0.070** | **0.218** | **0.077** |

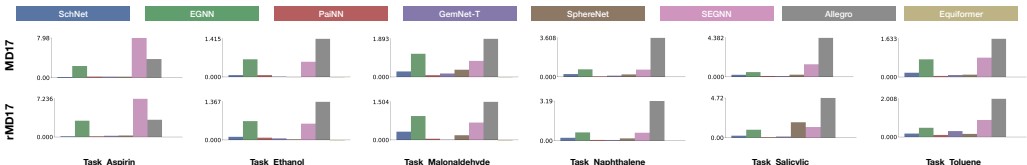

Figure 5: Ablation study on the effect of data normalization. Here are visualizations on performance differences on 6 tasks and 2 datasets, with MAE(force pred w/o normalization) - MAE(force pred w/ normalization).

## 4.2 Small Molecules: MD17 and rMD17

MD17 [8] is a dataset of molecular dynamics simulation. It has 8 tasks corresponding to eight organic molecules, and each task includes the molecule positions along the PES (see Fig. 3(b)). The goal is to predict each atom's energy and interatomic forces for each molecule's position. We follow the literature [68, 89, 109, 110] of using 8 subtasks, 1K for training and 1K for validation, while the test set (from 48K to 991K) is much larger. However, the MD17 dataset contains non-negligible numerical noises [9], and it is corrected by the revised MD17 (rMD17) dataset [10]. 100K structures were randomly chosen for each task/molecule in MD17, and the single-point force and energy calculations were performed for each structure using the PBE/def2-SVP level of theory. The calculations were conducted with tight SCF convergence and a dense DFT integration grid, significantly minimizing the computational noises.

The results on MD17 and rMD17 are in Table 2. We select 12 tasks for illustration, and more comprehensive results can be found in Appendix I. We can observe that, in general, PaiNN, GemNet and Equiformer perform well on MD17 and rMD17 tasks. We also report **ablation study on data normalization**. NequIP [3] and Allegro [95] introduce a normalization trick: multiplying the predicted energy with the mean of ground-truth force (reproduced results in Appendix J). We plot the performance gap, MAE(w/o normalization) - MAE(w/ normalization), in Fig. 5, and observe most of the gaps are positive, meaning that adding data normalization can lead to generally better performance.

## 4.3 Small Molecules: COLL

The COLL dataset [36] comprises energy and force data for 140K random snapshots obtained from molecular dynamics simulations of molecular collisions. These simulations were conducted using the semiempirical GFN2-xTB method. To obtain the data, DFT calculations were performed utilizing the revPBE functional and def2-TZVP basis set, which also incorporated D3 dispersion corrections. The task is to predict the energy and force for each atom in the molecule, and we consider 10 advanced geometric models for benchmarking. The results are in Table 3, and GemNet, SphereNet, and Equiformer reach more optimal performance.

Table 3: Results on energy and force prediction in COLL. 120k for training, 10k for val, 9.48k for test. The metric is the mean absolute error (MAE).

| Model | Energy ($eV$) ↓ | Force ($eV/\text{Å}$) ↓ |
|---|---|---|
| SchNet | 0.178 | 0.130 |
| DimeNet++ | 0.036 | 0.049 |
| EGNN | 1.808 | 0.234 |
| PaiNN | 0.030 | 0.052 |
| GemNet-T | **0.017** | **0.028** |
| SphereNet | 0.032 | 0.047 |
| SEGNN | 7.085 | 0.642 |
| NequIP | 0.120 | 0.113 |
| Allegro | 0.161 | 0.130 |
| Equiformer | 0.036 | 0.030 |

Table 4: Results on 2 binding affinity prediction tasks. We select three evaluation metrics for LBA: the root mean squared error (RMSD), the Pearson correlation ($R_p$) and the Spearman correlation ($R_S$). LEP is a binary classification task, and we use the area under the curve for receiver operating characteristics (ROC) and precision-recall (PR) for evaluation. We run cross-validation with 5 seeds, and the mean and std are reported.

| Model | LBA | | | LEP | |
|---|---|---|---|---|---|
| | RMSD ↓ | $R_P$ ↑ | $R_C$ ↑ | ROC ↑ | PR ↑ |
| SchNet | $1.521 \pm 0.02$ | $0.474 \pm 0.01$ | $0.452 \pm 0.01$ | $0.450 \pm 0.03$ | $0.379 \pm 0.03$ |
| DimeNet++ | $1.672 \pm 0.09$ | $0.550 \pm 0.01$ | $0.556 \pm 0.01$ | $0.590 \pm 0.06$ | $0.496 \pm 0.05$ |
| EGNN | $1.494 \pm 0.04$ | $0.503 \pm 0.04$ | $0.483 \pm 0.05$ | $\mathbf{0.657 \pm 0.05}$ | $\mathbf{0.559 \pm 0.05}$ |
| PaiNN | $\mathbf{1.434 \pm 0.02}$ | $0.583 \pm 0.02$ | $\mathbf{0.580 \pm 0.02}$ | $0.585 \pm 0.02$ | $0.432 \pm 0.03$ |
| GemNet-T | – | – | – | $0.659 \pm 0.05$ | $0.506 \pm 0.05$ |
| SphereNet | $1.581 \pm 0.02$ | $0.538 \pm 0.01$ | $0.529 \pm 0.01$ | $0.523 \pm 0.04$ | $0.432 \pm 0.05$ |
| SEGNN | $1.416 \pm 0.03$ | $0.566 \pm 0.02$ | $0.550 \pm 0.02$ | $0.574 \pm 0.03$ | $0.485 \pm 0.03$ |
| NequIP | $1.606 \pm 0.02$ | $0.537 \pm 0.01$ | $0.520 \pm 0.01$ | $0.538 \pm 0.12$ | $0.481 \pm 0.07$ |
| Allegro | $1.567 \pm 0.02$ | $0.547 \pm 0.00$ | $0.534 \pm 0.00$ | $0.627 \pm 0.04$ | $0.525 \pm 0.03$ |
| Equiformer | $1.392 \pm 0.03$ | $\mathbf{0.598 \pm 0.02}$ | $0.578 \pm 0.02$ | $0.618 \pm 0.06$ | $0.510 \pm 0.05$ |

Table 5: Results on 10 protein tasks from six datasets: ECSingle, ECMultiple, Fold (Fold, Sup., Fam.), GO (MF, BP, CC), MSP, and PSR. The evaluation metrics are Accuracy (ACC, %), $F_{max}$ (definition in Appendix B), ACC, $F_{max}$, receiver operating characteristics (ROC), and Spearman's $\rho$, respectively.

| | ECSingle | ECMultiple | Fold | | | GO | | | MSP | PSR | |
|---|---|---|---|---|---|---|---|---|---|---|---|
| | | | Fold | Sup. | Fam. | MF | BP | CC | | | |
| | ACC ↑ | $F_{max}$ ↑ | ACC ↑ | ACC ↑ | ACC ↑ | $F_{max}$ ↑ | $F_{max}$ ↑ | $F_{max}$ ↑ | ROC ↑ | Global $\rho$ ↑ | Mean $\rho$ ↑ |
| IEConv | – | – | 45.0 | 69.7 | 98.9 | – | – | – | – | – | – |
| GVP-GNN | 65.5 | 0.712 | 34.8 | 52.7 | 95.0 | 0.476 | 0.312 | 0.389 | 0.574 | 0.744 | 0.302 |
| GearNet | 78.8 | 0.799 | 29.1 | 43.1 | 95.9 | 0.477 | 0.283 | 0.373 | – | – | – |
| ProNet | 86.4 | 0.823 | 52.7 | 70.3 | 99.3 | 0.559 | 0.367 | 0.414 | 0.634 | $\mathbf{0.818}$ | 0.462 |
| CDConv | $\mathbf{86.9}$ | $\mathbf{0.862}$ | $\mathbf{60.0}$ | $\mathbf{79.9}$ | $\mathbf{99.5}$ | $\mathbf{0.649}$ | $\mathbf{0.435}$ | $\mathbf{0.450}$ | $\mathbf{0.717}$ | 0.817 | $\mathbf{0.500}$ |

## 4.4 Small Molecules & Proteins Binding: LBA & LEP

The binding affinity measures the strength of the binding interaction between a small molecule (ligand) to the target protein. In Geom3D, we consider modeling both the ligands and proteins with their 3D structures. During binding, a cavity in a protein can potentially possess suitable properties for binding a small molecule, and it is called a pocket [113]. Due to the large volume of protein, Geom3D follows existing works [118] by only taking the binding pocket instead of the whole protein structure. Specifically, Geom3D models up to 600 atoms for each ligand and protein pair. For the benchmarking, we consider two binding affinity tasks. (1) The first task is ligand binding affinity (LBA) [123]. It is gathered from [124], and the task is to predict the binding affinity strength between a ligand and a protein pocket. (2) The second task is ligand efficacy prediction (LEP) [34]. The input is a ligand and both the active and inactive conformers of a protein, and the goal is to classify whether or not the ligand can activate the protein's function. The results on two binding tasks are in Table 4, and we can observe that PaiNN, SEGNN, and Equiformer are generally outstanding on the two tasks.

## 4.5 Proteins: ECSingle, ECMultiple, Fold, GO, MSP, and PSR

**ECSingle** is a classification task [45] that classifies 37K proteins into 384 four-level Enzyme Commission (EC) types. This task aims to recognize the fundamental role of proteins as bio-catalysts or enzymes, which are essential in facilitating biological reactions. The EC numbering system [63] serves as a comprehensive numerical classification scheme, systematically organizing the varied functionalities of enzymes and providing a structured approach to understanding their biological roles.

**ECMultiple** is a multi-label classification task proposed by Gligorijevic et al. [39], where 19K proteins are associated with 538 distinct EC categories, including both three-level and four-level types and a single protein can be concurrently labeled with several three-level or four-level EC numbers.

**Fold** is a task classifying 16K proteins into 1,195 fold patterns [47, 74]. It is an important biological task in predicting the 3D structures from 1D amino acid sequences. We further consider three testsets (Fold, Superfamily, and Family) based on the sequence and structure similarity [94].

**GO** (Gene Ontology) is a dataset [39] with 36K proteins for GO term classification, where the GO term provides a consistent description of gene product attributes across species and databases [12]. Concretely, each protein contains up to three types of GO terms, corresponding to three types of classification tasks: (1) Molecular Function (MF) has 489 classes; (2) Biological Process (BP) has 1,943

Table 6: Results on the 8 tasks from MatBench and 1 task from QMOF (with optimal DA). The data split and task unit are in Appendix B, and the metric is the mean absolute error (MAE).

| Model | MatBench | | | | | | | | QMOF |
| | Per. $E_{\text{form}} \downarrow$ 18,928 | Dielectric $\downarrow$ 4,764 | $log_{10}G \downarrow$ 10,987 | $log_{10}K \downarrow$ 10,987 | $E_{\text{exfo}} \downarrow$ 636 | Phonons $\downarrow$ 1,265 | Band Gap $\downarrow$ 106,113 | $E_{\text{form}} \downarrow$ 132,752 | Band Gap $\downarrow$ 20,425 |
|---|---|---|---|---|---|---|---|---|---|
| SchNet | 0.040 | 0.334 | 0.081 | 0.060 | 65.201 | 42.586 | 0.327 | 0.026 | 0.236 |
| DimeNet++ | **0.037** | 0.357 | 0.081 | 0.058 | 68.685 | 38.339 | **0.208** | 0.025 | 0.234 |
| EGNN | 0.038 | 0.331 | 0.087 | 0.064 | 78.015 | 74.846 | 0.211 | 0.026 | 0.256 |
| PaiNN | 0.038 | 0.317 | **0.080** | **0.053** | 67.752 | 44.602 | **0.022** | 0.190 | 0.207 |
| GemNet-T | 0.042 | 0.325 | 0.088 | 0.061 | 68.425 | 48.986 | 0.186 | 0.026 | **0.207** |
| SphereNet | 0.043 | 0.388 | 0.087 | 0.061 | 72.987 | 36.300 | 0.217 | 0.029 | 0.251 |
| SEGNN | 0.046 | 0.360 | 0.087 | 0.059 | 65.052 | 43.638 | 0.330 | 0.047 | 0.330 |
| Equiformer | 0.046 | **0.280** | 0.087 | 0.057 | **62.977** | **37.381** | 0.202 | 0.027 | 0.234 |

classes; and (3) Cellular Component (CC) has 320 classes. Notice that each protein can be associated with multiple GO terms in each GO term type, thus all three tasks are multi-label classifications.

**MSP & PSR** are two protein tasks from a collection of benchmark datasets for machine learning in structural biology [118]. MSP (Mutation Stability Prediction) aims to predict whether the stability of a protein increases after mutation. The dataset is a mutation dataset containing 4K proteins. It is constructed by incorporating single-point mutations given in the SKEMPI database [56]. PSR (Protein Structure Ranking) is a regression task based on the Critical Assessment of Structure Prediction (CASP) [71]. In CASP, a protein structure is predicted and a quality score, the global distance test (GDT_TS), is calculated between the predicted structure and experimentally determined structure. This task aims to predict this score for 44K proteins.

The results of 5 models are in Table 5. CDConv [29] outperforms other models by a large margin on almost all 10 tasks, while ProNet [122] performs second well in general, and reaches the best result on the PSR task with global $\rho$ metric. Notice that certain entries in the table are temporarily left blank due to memory constraints encountered. More detailed dataset specifications are in Appendix B.

### 4.6 Crystalline Materials: MatBench and QMOF

**MatBench** [21] is explicitly created to evaluate the performance of machine learning models in predicting properties of inorganic bulk materials covering mechanical, electronic, and thermodynamic material properties [21]. Here we consider 8 regression tasks with crystal structures, including predicting the formation energy (Perovskites, $E_{\text{form}}$), exfoliation energies ($E_{\text{exfo}}$), band gap, shear and bulk modulus ($log_{10}G$ and $log_{10}K$), etc. Please check Appendix B for more details.

**Quantum MOF (QMOF)** [107] is a dataset of over 20K metal-organic frameworks (MOFs) and coordination polymers derived from DFT. The task is to predict the band gap, the energy gap between the valence band and the conduction band. The results of 8 geometric models on 8 MatBench tasks and 1 QMOF task are in Table 6, and we can observe that the performance of all the models is very close, while DimeNet++, PaiNN, GemNet-T, and Equiformer are slightly better.

We also conduct **ablation study on periodic data augmentation** on crystal materials. We note that there are two data augmentation (DA) methods: gathered and expanded. Gathered DA means that we shift the original unit cell along three dimensions, and the translated unit cells will have the *same* node indices as the original unit cell, *i.e.*, a multi-edge graph. How-

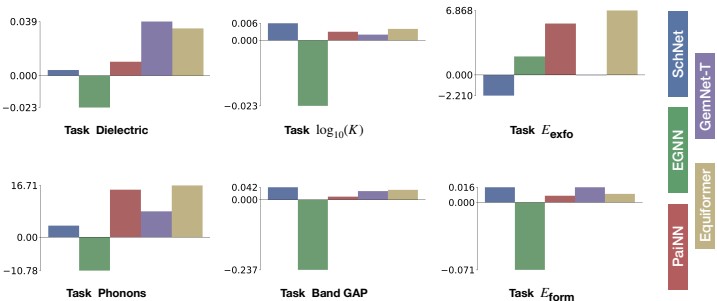

Figure 6: Ablation study on the performance gap with data augmentation (DA): MAE(expanded DA) - MAE(gathered DA).

ever, expanded DA will assume the translated unit cells have different node indices from the original unit cell. (A visual demonstration is in Appendix A). We conduct an ablation study on the effect of these two DAs, and we plot MAE(expanded DA) - MAE(gathered DA) on six tasks in Fig. 6. It reveals that for most of the models (except EGNN), using gathered DA can lead to consistently better performance, and thus it is preferred. For more qualitative analysis, please check Appendix J.

Table 7: QM9 downstream results after pretraining, and the backbone model is SchNet. We take 110K for training, 10K for validation, and 11K for testing. The evaluation metric is the mean absolute error (MAE).

| Pretraining | $\alpha\downarrow$ | $\nabla\mathcal{E}\downarrow$ | $\mathcal{E}_{\text{HOMO}}\downarrow$ | $\mathcal{E}_{\text{LUMO}}\downarrow$ | $\mu\downarrow$ | $C_v\downarrow$ | $G\downarrow$ | $H\downarrow$ | $R^2\downarrow$ | $U\downarrow$ | $U_0\downarrow$ | ZPVE$\downarrow$ |
|---|---|---|---|---|---|---|---|---|---|---|---|---|
| – (random init) | 0.060 | 44.13 | 27.64 | 22.55 | 0.028 | 0.031 | 14.19 | 14.05 | 0.133 | 13.93 | 13.27 | 1.749 |
| Supervised | 0.062 | 40.31 | 25.57 | 21.69 | 0.030 | 0.030 | 14.36 | 14.68 | 0.308 | 15.21 | 16.13 | 1.638 |
| Type Prediction | 0.073 | 45.38 | 28.76 | 24.83 | 0.036 | 0.032 | 16.66 | 16.28 | 0.275 | 15.56 | 14.66 | 2.094 |
| Distance Prediction | 0.065 | 45.87 | 27.61 | 23.34 | 0.031 | 0.033 | 14.83 | 15.81 | 0.248 | 15.07 | 15.01 | 1.837 |
| Angle Prediction | 0.066 | 48.45 | 29.02 | 24.40 | 0.034 | 0.031 | 14.13 | 13.77 | 0.214 | 13.50 | 13.47 | 1.861 |
| 3D InfoGraph | 0.062 | 45.96 | 29.29 | 24.60 | 0.028 | 0.030 | 13.93 | 13.97 | 0.133 | 13.55 | 13.47 | 1.644 |
| GeoSSL-RR | 0.060 | 43.71 | 27.71 | 22.84 | 0.028 | 0.031 | 14.54 | 13.70 | 0.122 | 13.81 | 13.75 | 1.694 |
| GeoSSL-InfoNCE | 0.061 | 44.38 | 27.67 | 22.85 | 0.027 | 0.030 | 13.38 | 13.36 | **0.116** | 13.05 | 13.00 | 1.643 |
| GeoSSL-EBM-NCE | 0.057 | 43.75 | 27.05 | 22.75 | 0.028 | 0.030 | 12.87 | 12.65 | 0.123 | 13.44 | 12.64 | 1.652 |
| 3D InfoMax | 0.057 | 42.09 | 25.90 | 21.60 | 0.028 | 0.030 | 13.73 | 13.62 | 0.141 | 13.81 | 13.30 | 1.670 |
| GraphMVP | 0.056 | 41.99 | 25.75 | 21.58 | 0.027 | 0.029 | 13.43 | 13.31 | 0.136 | 13.03 | 13.07 | 1.609 |
| GeoSSL-DDM-1L | 0.058 | 42.64 | 26.32 | 21.87 | 0.028 | 0.030 | 12.61 | 12.81 | 0.173 | 12.45 | 12.12 | 1.696 |
| GeoSSL-DDM | 0.056 | 42.29 | **25.61** | 21.88 | 0.027 | 0.029 | 11.54 | 11.14 | 0.168 | 11.06 | **10.96** | 1.660 |
| MoleculeSDE (VE) | 0.056 | 41.84 | 25.79 | 21.63 | 0.027 | 0.029 | **11.47** | **10.71** | 0.233 | **11.04** | 10.95 | **1.474** |
| MoleculeSDE (VP) | **0.054** | **41.77** | 25.74 | **21.41** | **0.026** | **0.028** | 13.07 | 12.05 | 0.151 | 12.54 | 12.04 | 1.587 |

## 4.7 Geometric Pretraining on Small Molecules

We run 14 pretraining algorithms, including one supervised pretraining: the pretraining dataset (*e.g.*, PCQM4Mv2 [51]) possess the energy or energy gap label for each conformation, which can be naturally adopted for pretraining. The benchmark results of using SchNet as the backbone model pretrained on PCQM4Mv2 and fine-tuning on QM9 tasks are in Table 7. We observe that MoleculeSDE and GeoSSL-DDM utilizing the geometric denoising diffusion models outperform other pretraining methods in most cases. On the other hand, supervised pretraining (pretrained on energy gap $\nabla\mathcal{E}$) reaches outstanding performance on $\nabla\mathcal{E}$ downstream task, yet the generalization to other tasks is modest. Please check Appendix I for more pretraining results with different backbone models.

## 5 Conclusion and Future Directions

Geom3D provides a unified view on the SE(3)-equivariant models, together with the implementations. Indeed these can serve as the building blocks to various tasks, such as geometric pretraining (as displayed in Sec. 4.7) and the conformation generation (ClofNet [20], MoleculeSDE [79]), paving the way for building more foundational models and solving more challenging tasks.

**Limitations on models and tasks.** Geom3D includes 10 topological models, 16 geometric models, 14 geometric pretraining methods, and 52 diverse tasks. We would also like to acknowledge there exist many more tasks (*e.g.*, Atom3D [118], Molecule3D [131], OC20 [7]) and more geometric models (*e.g.*, OrbNet [99], MACE [2], Uni-Mol [144], and LieTransformer [52]). The continual updating may necessitate the collective efforts of our entire community, exemplifying our collaborative endeavors.

**Foundation model as future exploration.** Recently, there have been certain explorations on building the foundation models for molecule discovery, especially by incorporating textual data on the molecule's functionalities [25, 26, 83, 87, 88, 115, 139, 143]. However, existing works mainly focus on the 1D sequence or 2D topology, while the 3D geometric structure of molecules is rarely considered. We believe that Geom3D can offer essential support for future explorations along this direction.

## Reproducibility and Tutorials

The codes of Geom3D have been released on this GitHub repository. Both the raw and preprocessed datasets have been released on this HuggingFace link. The checkpoints of all models have been released on this HuggingFace link. We further added four tutorials on using Geom3D on customized data, energy prediction, force prediction, and geometric pretraining. These tutorials can sufficiently demonstrate how users can inject new methods into Geom3D platform, showcasing its potential as a fundamental building block for tackling a wide range of machine learning tasks.

## Acknowledgement

The authors would like to thank Zichao Rong, Chengpeng Wang, Jiarui Lu, Farzaneh Heidari, Zuobai Zhang, Limei Wang, and Hanchen Wang for their helpful discussions. This project is supported by the Natural Sciences and Engineering Research Council (NSERC) Discovery Grant, the Canada CIFAR AI Chair Program, collaboration grants between Microsoft Research and Mila, Samsung Electronics Co., Ltd., Amazon Faculty Research Award, Tencent AI Lab Rhino-Bird Gift Fund, and a National Research Council of Canada (NRC) Collaborative R&D Project. This project was also partially funded by IVADO Fundamental Research Project grant PRF-2019-3583139727.

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
