# Appendix

## Table of Contents

## A    Data Structure and Data Preprocessing

Recall that as illustrated in Fig. 2, we split existing geometric models into three big venues: invariant modeling, equivariant modeling with spherical frame, and equivariant modeling with vector frame. All these modelings are application-agnostic, *i.e.*, they can be naturally adapted to small molecules, proteins, and crystal materials.

In this section, we would like to scrutinize the key data structure of such three data types. They are critical factors when we design geometric modeling. For instance:

- Small molecules often have <100 atoms in the 3D Euclidean space, and thus they can be easily fed into GPU memory.
- Proteins are macromolecules with tens of thousands of atoms. Thus, geometric models on small molecules cannot be easily adapted. Existing research works are working on modeling the backbone-level and residue-level, *i.e.*, only modeling the most important atoms (e.g., $C_\alpha, C, N$) in proteins.
- Crystal materials are molecules with periodic structures, and typical solutions consider periodic data augmentation before feeding them into geometric models.

In the next, we will explain in more detail of these three data structures.

### A.1    Small Molecules

In the machine learning and computational chemistry domain, existing works are mainly focusing on the molecule 1D description [70, 106, 126] and 2D topology graph [13, 17, 23, 38, 43, 66, 78, 101, 119, 130, 132, 132, 135]. Especially as the 2D graph, where the atoms and bonds are treated as nodes and edges, respectively. To model this graph structure, a message-passing graph neural network model family has been proposed.

Simultaneously, molecules can be naturally treated as 3D point clouds in Euclidean space, where atoms are the 3D points. In geometric modeling, as illustrated in Sec. 2, the inputs are atom types and atom positions, *i.e.*, $g = (X, R)$.

In Table 1, we provide a comparison of models on 1D descriptions, 2D topological graphs, and 3D geometric conformations. The observation verifies the necessity of using conformation for quantum property prediction tasks.

### A.2    Proteins

Protein structures can be classified into four primary levels. The primary structure represents the linear arrangement of amino acids within a polypeptide chain. Secondary structure arises from local interactions between adjacent amino acids, resulting in the formation of recognizable patterns like alpha helices and beta sheets. The tertiary structure encompasses the complete three-dimensional organization of a single protein, involving additional folding and structural modifications beyond the secondary structure. Quaternary structure emerges when multiple polypeptide chains or subunits interact to form a protein complex.

Specifically for geometric modeling, we are now focusing on the protein tertiary structure, which can be constructed based on different structural levels, namely the all-atom level, backbone level, and residue level. We explain the details below, and you can find an illustration in Fig. 3(c).

- At the all-atom level, the graph nodes represent individual atoms, capturing the fine-grained details of the protein structure.
- At the backbone level, the graph nodes correspond to the backbone atoms ($N - C_\alpha - C$), omitting the side chain information. This level of abstraction focuses on the essential backbone structure of the protein.
- At the residue level, the graph nodes represent amino acid residues. The position of each residue can be represented by the position of its $C_\alpha$ atom or calculated as the average position of the backbone atoms within the residue. This level provides a higher-level representation of the protein structure, grouping atoms into residue units.

### A.3    Crystalline Materials

**Periodic structure.** The crystalline materials or extended chemical structures possess a characteristic known as periodicity: their atomic or molecular arrangement repeats in a predictable and consistent pattern across all three spatial dimensions. This is the key aspect that differentiates them from small molecules. In Fig. 3(d), we

show an original unit cell (marked in green) that can repeatedly compose the crystal structure along the lattice. To model such a periodic structure, we adopt the data augmentation (DA) from CGCNN [129], yet with two variants as explained below.

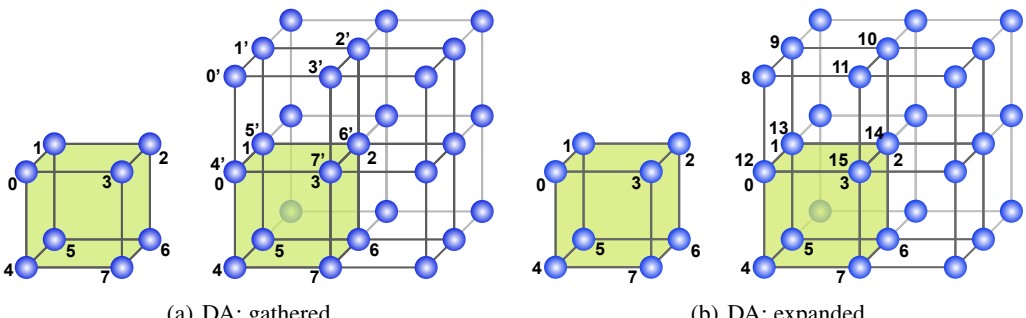

(a) DA: gathered.   (b) DA: expanded.

Figure 7: An illustration for crystalline material data augmentation (DA). Notice that in Fig. 7(a), the shifted unit cells and the original unit cells share the same corresponding node indices; for the demonstration clarity, we mark them with ′, *e.g.*, ) 0 and 0′ are the indices for the same nodes.

**Data augmentation 1: Gathered.**   Gathered DA means that we will shift the original unit cell along three dimensions, and the translated unit cells will have the same node indices as the original unit cell. An example is in Fig. 7(a).

**Data augmentation 2: Expanded.**   Expanded DA refers that we shift the original unit cell in the same way as Gathered, but the translated unit cells have different node indices from the original unit cell. An example is in Fig. 7(b).

Once we have these two augmentations, we have the augmented nodes and corresponding periodic coordinates. The edge connection needs to satisfy three conditions simultaneously:

- The pairwise distance should be larger than 0 and no larger than the threshold $\tau$, *i.e.*, the distance is within $(0, \tau)$.
- At least one of the linked nodes (bonded atoms) belongs to the anchor unit cell.
- No self-loop.

In specific, we give an example of the two DAs below. We take the same simple cubic crystal in Fig. 7 for illustration, and we assume that the edge length in the unit cell is $l$. The threshold for building the edge is $\tau = l$.

- Gathered DA. $(0, 1)$ satisfies the conditions; $(0, 3')$ violets the condition; $(0, 4')$ violets the conditions; $(0', 1')$ violets the conditions.
- Expanded DA. $(0, 1)$ satisfies the conditions; $(0, 11)$ violets the conditions; $(0, 12)$ violets the condition; $(8, 9)$ violets the conditions.

In terms of implementation, this can be easily achieved by calling the pymatgen [97] package. Such data augmentation is merely one way of handling the periodic data structure in crystalline materials. There could be more potential ways, and we would like to leave them for future exploration.

Thus, after the data augmentation, we can feed the augmented 3D point clouds into the geometric models. This modeling process is the same as that of the small molecules.

# B  Dataset Acquisition and Preparation & Benchmark Hyperparameters

For the dataset download, please check this GitHub repository for detailed instructions.

## B.1  Small Molecules: QM9

**Task specification.** QM9 [100] is a dataset of 134K molecules, consisting of 9 heavy atoms. It includes 12 tasks that are related to the quantum properties. For example, U0 and U298 are the internal energies at 0K and 298.15K, respectively, and H298 and G298 are the other two energies that can be transferred from U298, respectively. The other 8 tasks are quantum mechanics related to the DFT process.

**Task unit.** We list the units for 12 QM9 tasks below.

Table 8: Units for 12 tasks in QM9.

| $\alpha$ | $\nabla\mathcal{E}$ | $\mathcal{E}_{\text{HOMO}}$ | $\mathcal{E}_{\text{LUMO}}$ | $\mu$ | $C_v$ | $G$ | $H$ | $R^2$ | $U$ | $U_0$ | ZPVE |
|---|---|---|---|---|---|---|---|---|---|---|---|
| $\alpha_0^3$ | $meV$ | $meV$ | $meV$ | $D$ | $\frac{cal}{mol\cdot K}$ | $meV$ | $meV$ | $\alpha_0^2$ | $meV$ | $meV$ | $meV$ |

**Dataset size and split.** There are 133,885 molecules in QM9, where 3,054 are marked as "uncharacterized" and have been filtered out because they are rearranged during geometry optimization. This leads to 130,831 molecules. For data splitting, we use 110K for training, 10K for validation, and 11K for testing.

**Others.** Current work is using different optimization strategies and different data splits (in terms of the splitting size). During the benchmark, we find that: (1) The performance on QM9 is very robust to either using (i) 110K for training, 10K for validation, 10,831 for test or using (ii) 100K for training, 13,083 for validation and 17,748 for test. (2) The optimization, especially the learning rate scheduler, is very critical. During the benchmarking, we find that using cosine annealing learning rate schedule [90] is generally the most robust.

## B.2  Small Molecules: MD17

**Task specification.** MD17 [8] is a dataset on molecular dynamics simulation. It includes eight tasks, corresponding to eight organic molecules, and each task includes the molecule positions along the potential energy surface (PES), as shown in Fig. 3(b). The goal is to predict the energy-conserving interatomic forces for each atom in each molecule position.

**Task unit.** The MD17 aims for energy and force prediction. The unit is $\frac{kcal}{mol}$ for energy and $\frac{kcal}{mol\cdot\text{Å}}$ for force.

**Dataset size and split.** We follow the literature [68, 89, 109, 110] of using 1K for training and 1K for validation, while the test set (from 48K to 991K) is much larger, and we list them below.

Table 9: Dataset size and splits on MD17.

| Pretraining | Aspirin | Benzene | Ethanol | Malonaldehyde | Naphthalene | Salicylic | Toluene | Uracil |
|---|---|---|---|---|---|---|---|---|
| Train | 1K | 1K | 1K | 1K | 1K | 1K | 1K | 1K |
| Validation | 1K | 1K | 1K | 1K | 1K | 1K | 1K | 1K |
| Test | 209,762 | 47,863 | 553,092 | 991,237 | 324,250 | 318,231 | 440,790 | 131,770 |

**Others.** There are multiple ways to predict the energy, *e.g.*, using the SE(3)-equivariant to predict the forces directly. In Geom3D, we first predict the energy for each position; then, we take the gradient w.r.t. the input position. The Python codes are attached below:

```python
from torch.autograd import grad

positions = batch.positions # input positions
energy = model_3D(batch) # energy prediction
force = -grad(outputs=energy, inputs=positions) # force prediction
```

Notice that this holds for all the force prediction tasks, like rMD17 and COLL, which will be introduced below.

Additionally, in Appendix J.2, we will discuss the data normalization for MD prediction.

## B.3  Small Molecules: rMD17

**Task specification.** The revised MD17 (rMD) dataset [10] is constructed based on the original MD17 dataset. 100K structures were randomly chosen for each type of molecule present in the MD17 dataset. Subsequently, the single-point force and energy calculations were performed for each of these structures using the PBE/def2-SVP

level of theory. The calculations were conducted with tight SCF convergence and a dense DFT integration grid, significantly minimizing noise.

**Task unit.** The rMD17 aims for energy and force prediction. The unit is $\frac{kcal}{mol}$ for energy and $\frac{kcal}{mol \cdot \text{Å}}$ for force.

**Dataset size and split.** We use 950 for training, 50 for validation, and 1000 for test.

## B.4 Small Molecules: COLL

**Task specification.** COLL dataset [36] is a collection of configurations obtained from molecular dynamics simulations on molecular collisions. Around 140,000 snapshots were randomly taken from the trajectories of the collision, for each of which the energy and force were calculated using density functional theory (DFT).

**Task unit.** The rMD17 aims for energy and force prediction. The unit is $eV$ for energy and $eV/\text{Å}$ for force.

**Dataset size and split.** The published COLL dataset has split the whole data into 120,000 training samples, 10,000 validation samples, and 9,480 testing samples.

## B.5 Small Molecules & Proteins Binding: LBA & LEP

**Task specification.** Ligand-protein binding is formed between a small molecule (ligand) and a target protein. During the binding process, there is a cavity in a protein that can potentially possess suitable properties for binding a small molecule, called pocket [113]. Due to the large volume of protein, Geom3D follows existing works [118] by only taking the binding pocket, where there are no more than 600 atoms for each molecule and protein pair. For the benchmarking, we consider two binding affinity tasks. (1) The first task is ligand binding affinity (LBA) [123]. It is gathered from [124], and the task is to predict the binding affinity strength between a small molecule and a protein pocket. (2) The second task is ligand efficacy prediction (LEP) [34]. We have a molecule bounded to pockets, and the goal is to detect if the same molecule has a higher binding affinity with one pocket compared to the other one.

**Task unit.** LBA is to predict $pK = -\log(K)$, where K is the binding affinity in Molar units. LEP has no unit since it is a classification task.

**Dataset size and split.** The dataset size and splitting are listed below.

Table 10: Dataset size and splits on LBA & LEP. For LBA, we use split-by-sequence-identity-30: we split protein-ligand complexes such that no protein in the test dataset has more than 30% sequence identity with any protein in the training dataset. For LEP, we split the complex pairs by protein target.

| Pretraining | LBA | LEP |
|---|---|---|
| Train | 3,507 | 304 |
| Validation | 466 | 110 |
| Test | 490 | 104 |
| Split | split-by-identity-30 | split-by-target |

## B.6 Proteins: ECSingle

**Task specification.** The Enzyme Commission(EC) Number is a numerical classification of enzymes according to the catalyzed chemical reactions [50]. Therefore, the functions of enzymes and the chemical reaction type they catalyze can be represented by different EC numbers. An example of EC number is $EC3.1.1.4$: 3 represents Hydrolases (the first number represents enzyme class); 3.1 represents Ester Hydrolases (the second number represents enzyme subclass); 3.1.1 represents Carboxylic-ester Hydrolases (the third number represents enzyme sub-subclass); 3.1.1.4 represents Phospholipases (the fourth number represents the specific enzyme). The EC dataset was constructed by Hermosilla et al. [45] for the protein function prediction task. The enzyme reaction data with Enzyme Committee annotations were originally collected from the SIFTS database [14]. Then, all the protein chains were clustered using a 50% similarity threshold. EC numbers that were annotated for at least five clusters were selected and five proteins with less than 100% similarities were selected from each cluster, annotated by the EC number.

**Task unit.** No unit is available since it is a classification task.

**Dataset size and split.** ECSingle contains 37,428 protein chains, which were split into 29,215 for training, 2,562 for validation, and 5,651 for testing.

## B.7 Proteins: ECMultiple

**Task specification.** In a manner analogous to the ECSingle task, the ECMultiple task classifies proteins into various three-level and four-level EC numbers. Given a protein's potential for multifunctionality, it can be associated with multiple three- or four-level EC numbers. As illustrated in the ECSingle task, a three-level EC number is represented as 3.1.1.-, while a four-level number takes the form 3.1.1.4. As [39], the testing set is divided into subsets based on sequence identity relative to proteins in the training set. Specifically, the testing set is segmented for proteins exhibiting less than 30%, 40%, 50%, 70%, and 95% sequence similarity to those in the training set. $F_{max}$ is used as the evaluation metric since it is a multi-label classification task.

$F_{max}$, or the protein-centric maximum F Score [39], is proposed to evaluate the multi-label classification task, where each label can be considered as a binary classification task. Denote $\lambda \in [0, 1]$ as the threshold for the binary classification, $p_i^j$ as the probability for the $i$-th protein to be classified as true for class j, and $b_i^j \in 0, 1$ as the label for the $i$-th protein in the $j$-th class. Then, for each threshold value $\lambda$, we first calculate the precision and recall for the $i$-th protein:

$$\text{precision}_i(\lambda) = \frac{\sum_j^J((p_i^j \geq \lambda) \cap b_i^j)}{\sum_j^J(p_i^j \geq \lambda)}$$

$$\text{recall}_i(\lambda) = \frac{\sum_j^J(p_i^j \geq \lambda)}{\sum_j^J(b_i^j)}. \tag{5}$$

Then, we calculate the average precision and recall for each $\lambda$ over all the proteins. The denominator of the average precision represents the number of proteins with at least one prediction over the threshold:

$$\text{precision}(\lambda) = \frac{\sum_i^N precision_i(\lambda)}{\sum_i^N((\sum_j^J(p_i^j \geq \lambda)) \geq 1)}$$

$$\text{recall}(\lambda) = \frac{\sum_i^N recall_i(\lambda)}{N}. \tag{6}$$

Finally, $F_{max}$ is the maximum F score for $\lambda \in [0, 1]$:

$$F_{max} = \max_{\lambda \in [0,1]} \frac{2 \cdot precision(\lambda) \cdot recall(\lambda)}{precision(\lambda) + recall(\lambda)}. \tag{7}$$

**Task unit.** No unit is available since it is a classification task.

**Dataset size and split.** ECMultiple contains 19,198 protein chains, which are split into 15,550 for training, 1,729 for validation, and 1,919 for testing. The testing set is further split into: 720 for Tesing_<30%, 902 for Tesing_<40%, 1,117 for Tesing_<50%, 1,476 for Tesing_<70%, and 1,919 for Tesing_<95%. Notice that these sub-testing sets can have overlaps.

## B.8 Proteins: Fold

**Task specification.** Proteins can be hierarchically divided into different levels: Family, Superfamily, and Fold based on their sequence similarity, structure similarity, and evolutionary relations [94]. Proteins with (1) $\geq$30% residue identities or (2) lower residue identities but have similar functions are grouped into the same Family. A Superfamily is for families whose proteins have low residue identities but their structural and functional features suggest a possible same evolutionary origin. A Fold is for proteins sharing the same major secondary structures with the same arrangement and topological connections.

Based on the SCOP 1.75 database, all the fold categories can be grouped into seven structural classes with in total of 1195 fold types [75]: (a) all $\alpha$ proteins (primarily formed by $\alpha-$helices, 284 folds), (b) all $\beta$ proteins (primarily formed by $\beta-$sheets, 174 folds), (c) $\alpha/\beta$ proteins ($\alpha$-helices and $\beta$-strands interspersed, 147 folds), (d) $\alpha + \beta$ proteins ($\alpha$-helices and $\beta$-strands segregated, 376 folds), (e) multi-domain proteins (66 folds), (f) membrane and cell surface proteins and peptides (58 folds), and (g) small proteins (90 folds). DeepSF [47] proposed a three-level redundancy removal at fold superfamily/family levels, resulting in three subsets for testing.

- **Fold testing set** Firstly, the proteins are split into Fold-level training set and testing set, where the training set and testing set don't share the same superfamily.
- **Superfamily testing set** Then, the Fold-level training set is split into Superfamily-level training set and testing set, where they don't share the same family.
- **Family testing set** Finally, the Superfamily-level training set is split into Family-level training set and testing set, where for proteins in the same family, 80% of them are used for training and 20% of them are used for testing.

**Task unit.** No unit is available since they are classification tasks.

**Dataset size and split.** FOLD contains 16,292 proteins, and we follow [47]: 12,312 training samples, 736 validation samples, 3,244 testing samples. The testing samples contain 3 sub testsets: 718 for folding testset, 1,254 for superfamily testset, and 1,272 for family testset.

## B.9 Proteins: GO

**Task specification.** Gene Ontology (GO) categorizes terms into three distinct classifications:

- **Molecular Function (MF)** This describes activities at the molecular level. It captures the broad concept of a function without delving into specifics such as its location, the time it occurs, or the molecular or complex entity executing this function. An exemplar term under this category is "oxidoreductase activity."
- **Biological Process (BP)** This pertains to broader biological objectives achieved through one or more molecular functions. For instance, "cell division" falls under this category.
- **Cellular Component (CC)** This denotes the specific locations within or outside a cell where a gene product is active. It encompasses both subcellular structures and macromolecular complexes, with terms like "cytosolic large ribosomal subunit" being illustrative of this category [12].

For each of these categories, proteins can be affiliated with several GO terms. In the context of the testing set, it is divided based on sequence similarity with the training set proteins, specifically for those with less than 30%, 40%, 50%, 70%, and 95% similarity. Given that this is a multi-label classification task, the evaluation metric employed is $F_{max}$ described before.

**Task unit.** No unit is available since it is a classification task.

**Dataset size and split.** GO contains 36,632 protein chains, which were split into 29,894 for training, 3,322 for validation, and 3,416 for testing. The testing set is further split into: 1,717 for Tesing_<30%, 1,937 for Tesing_<40%, 2,199 for Tesing_<50%, 2,733 for Tesing_<70%, and 3,416 for Tesing_<95%. The sub-testing sets can overlap with each other.

## B.10 Proteins: MSP

**Task specification.** The Mutation Stability Prediction (MSP) task, as proposed in [118], aims to predict whether a protein's stability increases following a mutation, categorizing it as a binary classification task. The SKEMPI database, documented in [56], catalogs mutations present in protein–protein interactions along with their effects on binding affinity and various other attributes. To construct the MSP dataset, single-point mutations are modeled on the wild-type protein sequence, thereby producing the mutated protein variant. A single-point mutation refers to instances where a lone base pair is added, removed, or modified, which may subsequently alter the protein sequence. During the training process, the model independently generates representations for both the wild-type protein and its mutated counterpart. The proteins in the testing set have a <30% sequence similarity with proteins in the training set.

**Task unit.** No unit is available since it is a classification task.

**Dataset size and split.** MSP contains 4,184 protein chains, which were split into 2,864 for training, 937 for validation, and 347 for testing.

## B.11 Proteins: PSR

**Task specification.** The Critical Assessment of Structure Prediction (CASP) is a prestigious international competition that focuses on 3D protein structure prediction [71]. The Protein Structure Ranking (PSR) dataset has been curated from protein structures submitted to CASP, with the primary objective being the prediction of the Global Distance Test (GDT_TS) score between the predicted and experimentally determined structures. As such, this constitutes a regression task. As outlined on the official CASP website, the computation of the GDT_TS score includes the following steps:

(1) Superimposition of the predicted structure onto the true structure. (2) Calculation of pairwise distances between residues in the predicted structure and their respective counterparts in the true structure post-superimposition. (3) Determination of the percentage of residues that align within four distinct distance thresholds. (4) The GDT_TS score is derived by averaging the percentages obtained in the previous step.

In the CASP competition, participants are typically given an experimentally determined protein structure, along with a set of decoy structures. These decoys are generated using various computational modeling techniques and closely resemble the true protein in terms of structure. In other words, a true protein is associated with a set of decoys. The evaluation process in CASP not only assesses the proximity of predicted protein structures to the true

structure but also involves ranking the predicted structures relative to the decoys, for a more robust assessment. Correspondingly, we include both Global Spearman's $\rho$ and Mean Spearman's $\rho$ in the evaluation metrics. Global Spearman's $\rho$ is simply the correlation between the predicted GDT_TS score and the ground truth by calculating the Spearman's $\rho$ for all samples. In comparison, for Mean Spearman's $\rho$, we first separately calculate the Spearman's $\rho$ for decoys corresponding to the same protein and then take the average of these Spearman's $\rho$.

**Task unit.** GDT_TS has no units.

**Dataset size and split.** PSR contains 44,214 protein chains, which were split into 25,400 for training, 2,800 for validation, and 16,014 for testing.

## B.12 Crystalline Materials: MatBench

**Task specification.** MatBench [21] is a test suite for benchmarking 13 machine learning model performances for predicting different material properties. The dataset size for these tasks varies from 312 to 132k. The MatBench dataset has been pre-processed to clean up the task-irrelevant and unphysical-computed data. For benchmarking, we take 8 regression tasks with crystal structure data. These tasks are [16, 22, 55] Formation energy per Perovskite cell (Per. $E_{\text{form}}$), Refractive index (Dielectric), Shear modulus ($log_{10}G$), Bulk modulus($log_{10}K$), exfoliation energy ($E_{\text{exfo}}$), frequency at last phonon PhDOS peak (Phonons), band gap (Band Gap), and formation energy ($E_{\text{form}}$). Detailed explanations are as below:

- Perovskites: predicting formation energy from the crystal structure.
- Dielectric: predicting refractive index from the crystal structure.
- $log_{10}G$: predicting DFT log10 VRH-average shear modulus from crystal structure.
- $log_{10}K$: predicting DFT log10 VRH-average bulk modulus from crystal structure.
- $E_{\text{exfo}}$: predicting exfoliation energies from the crystal structure.
- Phonons: predicting vibration properties from the crystal structure.
- Band Gap: predicting DFT PBE band gap from the crystal structure.
- $E_{\text{form}}$: predicting DFT formation energy from the crystal structure.

**Task unit.** The unit for each task is listed below.

**Dataset size and split.** The dataset size for each task is listed above. For benchmarking, we take 60%-20%-20% as training-validation-testing for all tasks.

Table 11: Unit, dataset size, and naming specifications for MatBench.

| Column in MatBench | Perovskites | Dielectric | log gvrh | log kvrh | jdft2d | Phonons | Band Gap | E Form |
|---|---|---|---|---|---|---|---|---|
| Task Name in Table 6 | Per. $E_{\text{form}}$ | Dielectric | $log_{10}G$ | $log_{10}K$ | $E_{\text{exfo}}$ | Phonons | Band Gap | $E_{\text{form}}$ |
| Size | 18,928 | 4,764 | 10,987 | 10,987 | 636 | 1,265 | 106,113 | 132,752 |
| Unit | $eV$ | – | $log_{10}$ GPa | $log_{10}$ GPa | $meV$ | cm$^{-1}$ | $eV$ | $eV$/atom |

## B.13 Crystalline Materials: QMOF

**Task specification.** QMOF [107] is a database containing 20,425 metal–organic frameworks (MOFs) with quantum-chemical properties generated using density functional theory (DFT) calculations. The task is to predict the band gap, the energy gap between the valence band and the conduction band.

**Task unit.** The unit for the band gap task is $eV$.

**Dataset size and split.** As mentioned above, there are 20,425 MOFs, and we take 80%-10%-10% for training-validation-testing.

# C   Group Representation and Equivariance

Symmetry is everywhere on Earth, such as in animals, plants, and molecules. The group theory is the most expressive tool to depict such physical symmetry. In this section, we would like to go through certain key concepts in group theory.

**Symmetry** is the collection of all transformations under which an object is invariant. The readers can easily check that these transformations are automatically invertible and form a group, where the group multiplication is identified with the composition operation of two transformations. From a dynamical system point of view, symmetries are essential for reducing the degree of freedom of a system. For example, Noether's first theorem states that every differentiable symmetry of a physical system with conservative forces has a corresponding conservation law [96]. Therefore, symmetries form an important source of inductive bias that can shed light on the design of neural networks for modeling physical systems.

**Type-0, type-1 and higher-order particles** are critical concepts in describing physical word. In general, different orders of spherical harmonics exhibit varying degrees of angular variation across the sphere's surface. This variation refers to how quickly the function changes as you move around in different directions on the sphere. Lower-order spherical harmonics have smoother angular patterns with slower rates of change, while higher-order harmonics have more rapid changes in angular directions. In concrete, type-0 features include pairwise distance, and (dihedral) angle information, and type-1 features cover 3D coordinates, velocities, and forces.

## C.1   Group

A **group** is a set $G$ equipped with an operator (group product) $\times$, and they need to follow three rules:

1. It contains an identity element $e \in G$, s.t. $ae = ea = a, \forall a \in G$.
2. Associativity rule $(ab)c = a(bc)$.
3. Each element has an inverse $aa^{-1} = a^{-1}a = e$.

Below we list several well-known groups:

- **O(n)** is an n-dimensional **orthogonal group** that consists of rotation and reflections.
- **SO(n)** is a **special orthogonal group** that only consists of rotations.
- **E(n)** is an n-dimensional **Euclidean group** that consists of rotations, translations, and reflections.
- **SE(n)** is an n-dimensional **special Euclidean group**, which comprises arbitrary combinations of rotations and translations (no reflections).
- **Lie Group** is a group whose elements form a differentiable manifold. All the groups above are specific examples of the Lie Group.

## C.2   Group Representation and Irreducible Group Representation

**Group representation** is a mapping from the group $G$ to the group of linear transformations of a vector space $X$ with dimension $d$ (see [138] for more rigorous definition):

$$\rho_X(\cdot) : G \to \mathbb{R}^{d \times d} \qquad \text{s.t.} \quad \rho(e) = 1 \ \wedge \ \rho_X(a)\rho_X(b) = \rho_X(a \times b), \ \forall a, b \in G. \tag{8}$$

During modeling, the $X$ space can be the input 3D Euclidean space, the equivariant vector space in the intermediate layers, or the output force space. This enables the definition of equivariance as in Appendix C.3.

Group representation of SO(3) can be applied to any n-dimensional vector space. If we map SO(3) to the 3D Euclidean space (*i.e.*, $n = 3$), the group representation has the same formula as the rotation matrix.

**Irreducible representations of rotations** The irreducible representations (irreps) of SO(3) are indexed by the integers 0, 1, 2, ..., and we call this index $l$. The $l$-irrep is of dimension $2l + 1$. $l = 0$ (dimension 1) corresponds to scalars and $l = 1$ (dimension 3) corresponds to vectors.

## C.3   Equivariance and Invariance

**Equivariance** is the property for the geometric modeling function $f : X \to Y$, and we want to design a function $f$ that is equivariant as:

$$f(\rho_X(a)x) = \rho_Y(a)f(x), \ \ \forall a \in G, x \in X. \tag{9}$$

How to understand this in the molecule discovery scenarios? $\rho_X(g)$ is the group representation on the input space, like atom coordinates; and $\rho_Y(g)$ is the group representation on the output space $Y = f(X)$, *e.g.*, the force field space. Equivariance modeling in Eq. (9) is essentially saying that the designed deep learning model $f$ is modeling the whole transformation trajectory (*e.g.*, rotation for SO(3)-group) on the molecule conformations, and the output is the transformed $\hat{y}$ accordingly.

Note that in deep learning, a function with learned parameters can be abstracted as $f : W \times X \to Y$, where $w \in W$ is a choice of learned parameters (or weights). The parameters are scalars, *i.e.*, they don't transform

under a transformation of E(3)/SE(3). This implies that weights are scalars and are invariant under any choice of coordinate system.

**Invariance** is a special type of equivariance where

$$f(\rho_X(\boldsymbol{a})\boldsymbol{x}) = f(\boldsymbol{x}), \;\; \forall \boldsymbol{a} \in G, \boldsymbol{x} \in X, \tag{10}$$

with $\rho_Y(g)$ as the identity $\forall g \in G$.

Thus, group and group representation help define the equivariance condition for $f$ to follow. Then, the question turns to how to design such invariant or equivariant $f$.

- In Sec. 3.2, we introduced the invariant geometric models.
- In Sec. 3.3, we briefly discussed two main categories of equivariant geometric models: the spherical frame basis model and the vector frame basis model. In the following, we will introduce both in more detail in Appendices D and E, respectively.

Through lifting from the original geometric space to its frame bundle (see [49] for the precise definition), equivariant operations like covariant derivatives are realized in an invariant way. From a practical perspective, the lifting operation can be alternatively replaced by scalarization by equivariant frames. See [19, 20] for an illustration. Therefore, invariance and equivariance are just two equivariant descriptions of characterizing symmetry that can be transformed into each other through frames.

One thing we want to highlight is that convolutional neural networks (CNNs) on images are translation-equivariant on $\mathbb{R}^2$, which demonstrates the power of encoding symmetry into the deep neural network architectures.

# D Equivariance with Spherical Frame Basis

First, we would like to give a high-level idea of this basis:

- It introduces the spherical harmonics as the basis and maps all the points into such a space.
- The mapping from 3D Euclidean space to the spherical harmonics space satisfies the E(3)/SE(3)-equivariance property as defined Eq. (9).
- Based on such basis, we can design a message-passing framework to learn the desired properties.

Then, we would like to refer to Figure 2 in SEGNN ArXiv version v3. It nicely illustrates how the equivariance works in the spherical harmonics space.

**Spherical Harmonics**    The spherical harmonics are functions from points on the sphere to vectors, or more rigoriously:

**Definition 1.** *The spherical harmonics are a family o functions $Y^l$ from the unit sphere to the irrep $D^l$. For each $l = 0, 1, 2...,$ the spherical harmonics can be seen as a vector of $2l + 1$ functions $Y^l(\vec{x}) = \left(Y^l_{-l}(\vec{x}), Y^l_{-l+1}(\vec{x}), ..., Y^l_l(\vec{x})\right)$. Each $Y^l$ is equivariant to SO(3) with respect to the irrep of the same order,* i.e.,*

$$Y^l_m(R\vec{x}) = \sum_{n=-1}^{l} D^l(R)_{mn} Y^l_n(\vec{x}), \tag{11}$$

*where $R$ is any rotation matrix and $D^l$ are the irreducible representation of SO(3). They are normalized $\|Y^l(\vec{x})\| = 1$ when evaluated on the sphere $\|\vec{x}\| = 1$.*

According to Eq. (9), Eq. (11) satisfies the equivariance property: the input space $X$ is the 3D Euclidean space, and the output space $Y$ is the Spherical Harmonics space.

Some key points we would like to highlight:

- Sphere $\mathbb{S}^2$ is not a group, but it is a homogeneous space of SO(3).
- The decomposition into the irreducible group representations makes it steerable.
- The parameter $l$ is named the **rotation order**.

**Model Design**    With the spherical basis, we can design our own geometric models. Notice that during the modeling process, all the variables are tensors.

For instance, we can take the vector $\boldsymbol{r}_j - \boldsymbol{r}_i$ as the vector in $Y^l_m(\frac{\boldsymbol{r}_j - \boldsymbol{r}_i}{\|\boldsymbol{r}_j - \boldsymbol{r}_i\|})$. As shown in Eq. (11), this is rotation-equivariant. And we can easily see $\boldsymbol{r}_j - \boldsymbol{r}_i$ is translation-equivariant.

This term can be naturally adopted for the edge embedding under the message passing framework [38], and we can parameterize it with a radial term [112] as[2]:

$$\boldsymbol{h}_{i,j} = \text{Radial}(\|\boldsymbol{r}_j - \boldsymbol{r}_i\|) Y^l_m(\frac{\boldsymbol{r}_j - \boldsymbol{r}_i}{\|\boldsymbol{r}_j - \boldsymbol{r}_i\|}), \tag{12}$$

where the radial function is invariant with the pairwise distance as the input. This is for the message function. Then generally, for the update and aggregate function of node-level tensor $\boldsymbol{v}_i$, we have two options:

$$\boldsymbol{v}_i = \begin{cases} \boldsymbol{v}_i + \sum_{j \in \mathcal{N}(i)} \boldsymbol{h}_{i,j} + \boldsymbol{v}_j \\ \boldsymbol{v}_i + \sum_{j \in \mathcal{N}(i)} \boldsymbol{h}_{i,j} \otimes \boldsymbol{v}_j, \end{cases} \tag{13}$$

where the update can be done either with plus or multiplication. Note that $\otimes$ is the tensor product, which can be calculated using the *Clebsch-Gordan coefficients*. Please refer to [37] for more details.

---

[2]Notice that the index of angular momentum in the spherical frame is very important, yet we ignore them here for brevity. Please refer to the papers for more rigorous definitions.

# E   Equivariance with Vector Frame Basis

In physics, the vector frame is equivalent to the coordinate system. For example, we may assign a frame to all observers, although different observers may collect different data under different frames, the underlying physics law should be the same. In other words, denote the physics law by $f$, then $f$ should be an equivariant function.

Since there are three orthogonal directions in $\mathbf{R}^3$, a vector frame in $\mathbf{R}^3$ consists of three orthogonal vectors:

$$F = (\boldsymbol{e}_1, \boldsymbol{e}_2, \boldsymbol{e}_3).$$

Once equipped with a vector frame (coordinate system), we can project all geometric quantities to this vector frame. For example, an abstract vector $\boldsymbol{r} \in \mathbf{R}^3$ can be written as $\boldsymbol{r} = (r_1, r_2, r_3)$ under vector frame $F$, if: $\boldsymbol{r} = r_1 \boldsymbol{e}_1 + r_2 \boldsymbol{e}_2 + r_3 \boldsymbol{e}_3$. An equivariant vector frame further requires the three orthonormal vectors in $(\boldsymbol{e}_1, \boldsymbol{e}_2, \boldsymbol{e}_3)$ to be equivariant. Intuitively, an equivariant vector frame will transform according to the global rotation or translation of the whole system. Once equipped with an equivariant vector frame, we can project equivariant vectors into this vector frame:

$$\boldsymbol{r} = \tilde{r}_1 \boldsymbol{e}_1 + \tilde{r}_2 \boldsymbol{e}_2 + \tilde{r}_3 \boldsymbol{e}_3. \tag{14}$$

We call the process of $\boldsymbol{r} \rightarrow \tilde{r} := (\tilde{r}_1, \tilde{r}_2, \tilde{r}_3)$ the **projection** operation. Since $\tilde{r}_i = \boldsymbol{e}_i \cdot \boldsymbol{r}_i$ is expressed as an inner product between equivariant vectors, we know that $\tilde{r}$ consists of scalars.

We assign an equivariant vector frame to each node/edge to incorporate equivariant frames with graph message passing. Therefore, we call them the local frames. For example, consider node $i$ and one of its neighbors $j$ with positions $\boldsymbol{x}_i$ and $\boldsymbol{x}_j$, respectively. One way to construct the equivariant frame is the orthonormal frame using the Gram-Schmidt, like Clofnet [20] and MoleculeSDE [79]. The vector frame $\mathcal{F}_{ij} := (\boldsymbol{e}_1^{ij}, \boldsymbol{e}_2^{ij}, \boldsymbol{e}_3^{ij})$ is defined with respect to $\boldsymbol{x}_i$ and $\boldsymbol{x}_j$ as follows:

$$\left( \frac{\boldsymbol{x}_i - \boldsymbol{x}_j}{\|\boldsymbol{x}_i - \boldsymbol{x}_j\|}, \frac{\boldsymbol{x}_i \times \boldsymbol{x}_j}{\|\boldsymbol{x}_i \times \boldsymbol{x}_j\|}, \frac{\boldsymbol{x}_i - \boldsymbol{x}_j}{\|\boldsymbol{x}_i - \boldsymbol{x}_j\|} \times \frac{\boldsymbol{x}_i \times \boldsymbol{x}_j}{\|\boldsymbol{x}_i \times \boldsymbol{x}_j\|} \right). \tag{15}$$

The Gram-Schmidt orthogonalization makes sure that the Local-Frame($\boldsymbol{r}_i, \boldsymbol{r}_j$) is orthonormal. However, there also exist other ways to construct the vector frames, like using the protein backbone structures [29, 53]. Finally, it's worth mentioning that global frames can be built by pooling local frames. For example, a graph-level equivariant frame is obtained by aggregating node frames and implementing the Gram-Schmidt orthogonalization. However, the Newton dynamics experiments in [20] demonstrated that the global frame's performance is worse than edge local frames. Therefore, although edge-, node-, and global- frames are equal in terms of equivariance, the optimization properties of different equivariant frames depend varies according to different scientific datasets.

# F   Other Geometric Modeling (Featurization and Lie Group)

We also want to acknowledge other equivariant modeling methods.

**Featurization.** OrbNet [99] models the atomic orbital, the description of the location and wave-like behavior of electrons in atoms. This possesses a finer-grained featurization level than other methods. Voxel means that we discretize the 3D Euclidean space into bins, and recent work [33] empirically shows that this also applies to geometry learning tasks.

**Equivariance modeling with Lie group.** In previous sections, equivariant algorithms are viewed as mappings from a 3D point cloud (which discretizes the 3D Euclidean space) to another 3D point cloud, or as mappings to invariant quantities. From this point of view, the symmetry group $E(3)/SE(3)$ manifests itself as a group action transforming the Euclidean space. However, it is worth noting that this action is transitive in the sense that any two points in 3D Euclidean space can be transformed from one to the other through a combination of translation and rotation. In mathematical terms, the 3D Euclidean space is a **homogeneous space** of the group $E(3)$. Exploiting this observation, LieConv [32] and LieTransformer [52] elevate the 3D point cloud to the $E(3)$ group and perform parameterized group convolution (and attention) operations, ensuring equivariance, to obtain an equivariant embedding on the group $E(3)$. Finally, by projecting the result back to $\mathbf{R}^3$ (taking the quotient), an equivariant map from $\mathbf{R}^3$ to the output space is obtained. The main limitation of Lie group modeling lies in the convolution operation, which often involves high-dimensional integration and requires approximation for most groups. For more in-depth insights into the properties of convolution on groups, we refer readers to [5]. Another lifting of $\mathbf{R}^3$ is to lift it to the SO(3) frame bundle, such that the SO(3) group transforms one orthonormal frame to another orthonormal frame transitively. This lifting also inspires the design of [19, 20].

# G  Expressive Power: from Invariance to Equivariance

Equivariant neural networks are constructed for equivariant tasks. That is, to approximate an equivariant function. Compared with ordinary neural networks, a natural question arises: *Does an equivariant neural network have the universal approximation property whiten the equivariant function class?* By the novel D-spanning concept [24], this question is partially answered. The author further proposed two types of equivariant architectures that can enjoy the D-spanning property: 1. the G-equivariant polynomials enhanced TFN; 2. the minimal universal architecture constructed by tensor products. Therefore, at least in terms of universal approximation, an equivariant neural network doesn't necessarily require irreducible representations and Clebsch-Gordan decomposition. The reader can check [20] for how to realize the minimal universal architecture in an invariant way through equivariant frames and tensorized graph neural networks (*e.g.*, [65]). Informally, we conclude that an invariant graph neural network equipped with a powerful message-passing mechanism can achieve the universal approximation property. Another proof strategy of the universality of invariant scalars that doesn't rely on theories of tensorized graph neural networks can be found in [120].

However, the mainstream GNN is usually based on a 1-hop message passing mechanism (although tensorized graph neural networks have empirically shown competitive performances in molecular tasks) for computational efficiency. For 1-hop message passing mechanisms (including node-based transformers), our previous conclusion no longer holds, and vector (or higher order tensors) updates are necessary for enhancing the expressiveness power. The reader can consult the concrete example from PaiNN [110] to illustrate this point.

More precisely, We denote the nodes in Figure 1 of [110] as $\{a : white, b : blue, c : red, d : white\}$, and we consider whether the message of $b$ and $c$ received from their 1-hop neighbors can discriminate the two different geometric structures. For node $b$, the invariant geometric information we can get from 1-hop neighbors are the relative distance $d_{ab}$ and $d_{bc}$ and their intersection angle $\alpha_1$. Since the relative distances of the two structures remain equal, only the angle information is useful. Similarly, for node $c$, we have the intersection angle $\alpha_2$. Unfortunately, the intersection angles $\alpha_1$ and $\alpha_2$ of the two structures are still the same, and we conclude that invariant features are insufficient for discriminating the two different structures. On the other hand, [110] showed that by introducing directional vector features (type-1 equivariant steerable features), we are able to solve the problem in this special case, which proves the superiority of 'equivariance' over 'invariance' within 1-hop message passing mechanisms. Another invariant way of filling in this type of expressiveness gaps systematically is to introduce the information of frame transitions **FTE**, as was demonstrated in [19].

Vector update is just a special case of the more general higher-order tensor updates. To merge general equivariant tensors into our GNN, we can either utilize tensor products of vector frames [19], or introduce the concepts of spherical harmonics, which form a complete basis in the sense of irreducible representations of group $SO(3)$ and $O(3)$. However, to express the output of the tensor product between spherical harmonics as a combination of spherical harmonics is nontrivial. Fortunately, this procedure has been studied by quantum physicists, which is named after the Clebsch-Gordan decomposition (coefficients) [93]. Combining these blocks, we can build convolution or attention-based equivariant graph neural networks, see [37, 73] for detailed constructions.

# H Architecture for Geometric Representation

In this section, we are going to give a brief review of certain advanced geometric models, and a summary of more methods can be found in Table 12. Meanwhile, we will keep updating more advanced models.

We include all the hyperparameters in this GitHub repository. We are sure that we won't be able to tune all the hyperparameters, yet we want to claim that our reported results are reproducible using the hyperparameters listed above. In the future, we appreciate any contribution to do more searching on this.

Table 12: Categorization on geometric methods. For pretraining methods, the categorization is based on the pretraining algorithms and backbone models are not considered.

| | Model | Invariance | Equivariance | |
| | | | Spherical Frame | Vector Frame |
| --- | --- | --- | --- | --- |
| Representation | SchNet [109] | ✓ | | |
| | DimeNet [68] | ✓ | | |
| | SphereNet [89] | ✓ | | |
| | GemNet [67] | ✓ | | |
| | IEConv [45] | ✓ | | |
| | GearNet [142] | ✓ | | |
| | ProNet [122] | ✓ | | |
| | TFN [35] | | ✓ | |
| | SEGNN [4] | | ✓ | |
| | SE(3)-Trans [35] | | ✓ | |
| | NequIP [3] | | ✓ | |
| | Allegro [95] | | ✓ | |
| | Equiformer [73] | | ✓ | |
| | PaiNN [110] | | | ✓ |
| | GVP-GNN [62] | | | ✓ |
| | CDConv [29] | | | ✓ |
| | EGNN [108] | | ✓ | ✓ |
| Pretraining | GraphMVP [86] | ✓ | | |
| | 3D InfoMax [114] | ✓ | | |
| | GeoSSL-RR [81] | ✓ | | |
| | GeoSSL-EBM-NCE [81] | ✓ | | |
| | GeoSSL-InfoNCE [81] | ✓ | | |
| | GeoSSL-DDM [81] | ✓ | | |
| | GeoSSL-DDM-1L [136] | ✓ | | |
| | 3D-EMGP [59] | | | ✓ |
| | MoleculeSDE [79] | | | ✓ |

Generally, all the algorithms can be classified into two categories: SE(3)-invariant and SE(3)-equivariant. Note that, rigorously, SE(3)-invariant is also SE(3)-equivariant. Here we follow the definition in [37][3]:

- SE(3)-invariant models only operate on scalars ($l = 0$) which interact thought simple scalar multiplication. These scalars include pairwise distance, triplet-wise angle, etc, that will not change under rotation. In other words, the SE(3)-invariant pre-compute the invariant features and throw away the coordinate system.
- SE(3)-equivariant models keep the coordinate system and if the coordinate system changes, the outputs change accordingly. These models have been believed to empower larger model capacity [3, 37] with $l > 0$ quantities.

There are other variants, like the activation functions, the number of layers, normalization layers, etc. In this section, we will stick to the key module, *i.e.*, the SE(3)-invariant and SE(3)-equivariant modules for each backbone model.

The aggregation function is the same as:

$$h_i' = \text{aggregate}_{j \in \mathcal{N}(i)}(m_{ij}). \tag{16}$$

In the following, we will be mainly discussing the message-passing function as below.

---

[3]Also a video by Tess et al, link is here.

## H.1 Invariant Models

**SchNet**  SchNet [109] simply handles a molecule by feeding in the pairwise distance and throws them into the message-passing style GNN.

$$m_{ij} = \text{MLP}(h_j, \textbf{RBF}(d_{ij})). \tag{17}$$

where $\textbf{RBF}(\cdot)$ is the RBF kernel.

**DimeNet**  The directional message passing neural network (DimeNet and DimeNet++) [69]. The message passing function in DimeNet is two-hop instead of one-hop. Such message-passing step is similar to directed message-passing neural network (D-MPNN) [132], and it can reduce the redundancy during the message passing process.

$$m_{ji}^{l+1} = \sum_{k \in \mathcal{N}_j \setminus \{i\}} \text{MLP}(m_{ji}^l, \textbf{RBF}(d_{ji}), \textbf{SBF}(d_{kj}, \alpha_{\angle kij})), \tag{18}$$

where $\text{SBF}_{ln}(d_{kj}, \alpha_{\angle kij}) = \sqrt{\frac{2}{c^3 j_{l+1}^2(z_{ln})}} j_l(\frac{z_{ln}}{c} d_{kj}) Y_l^0(\alpha)$ is the spherical Fourier-Bessel (spherical harmonics) basis, a joint 2D basis for distance $d_{kj}$ and angle $\alpha_{\angle kij}$.

**SphereNet**  SphereNet [89] is an extension of DimeNet by further modeling the dihedral angle. It first adopts the spherical Fourier-Bessel (spherical harmonics) basis for dihedral angle modeling, namely

$$\text{SBF}(d, \theta, \phi) = j_l(\frac{\beta_{l_n}}{c} d) Y_l^m(\theta, \phi). \tag{19}$$

In addition, the basic operation of SphereNet is based on the quadruplets: r, s, $q_1$, and these three nodes formulate a reference plan to provide the polar angle to the point $q_2$. However, SphereNet provides an acceleration module, by projecting all the neighborhoods of s, in an anticlockwise direction, and the reference plan for each node $q_i$ is determined by r, s and $q_{i-1}$. Thus, the computational complexity is reduced by one order of magnitude. SphereNet further considers the following for distance and angle modeling:

$$\text{CBF}_{ln}(d_{kj}, \alpha_{\angle kij}) = \sqrt{\frac{2}{c^3 j_{l+1}^2(z_{ln})}} j_l(\frac{z_{ln}}{c} d_{kj}) Y_l^0(\alpha), \qquad \text{RBF}_{ln}(d_{kj}) = \sqrt{\frac{2}{c}} \frac{\sin(\frac{n\pi}{c} d)}{d}. \tag{20}$$

**GemNet**  GemNet [67] further extends DimeNet and SphereNet. It explicitly models the dihedral angle. Notice that both GemNet and SphereNet are using the SBF for dihedral angle modeling, yet the difference is that GemNet is using edge-based 2-hop information, *i.e.*, the torsion angle, while SphereNet is using the edge-based 1-hop information. Thus, GemNet is expected to possess richer information, while the trade-off is the larger computational efficiency (by one order of magnitude): GemNet has the complexity of $O(nk^3)$ while SphereNet is $O(nk^2)$.

**GearNet and ProNet for Macromolecules**  The invariant modeling for macromolecules follows the same strategy as the previous geometric models. The only difference, as illustrated in Appendix A, is that the modeling particles are the protein backbones ($N - C_\alpha - C$) or residues ($C_\alpha$). GearNet [142] and ProNet [122] are modeling the pairwise distance and dihedral angles at different scales.

## H.2 Equivariant Models with Spherical Frame Basis

**TFN**  Tensor field network (TFN) [112] first introduces using the SE(3)-equivariance group symmetry for modeling the geometric molecule data. As will be introduced later, the translation-equivariance can be easily achieved by considering the relative coordinates, *i.e.*, $\vec{r} = \boldsymbol{r}_i - \boldsymbol{r}_j$. Then the problem is simplified to design an SO(3)-equivariant model. To handle this, TFN first proposes a general **framework** by using the spherical harmonics as the basis satisfying the following for all $\boldsymbol{a} \in SO(3)$ and $\hat{\boldsymbol{r}}$:

$$Y_m^l(R(\boldsymbol{a})\hat{\boldsymbol{r}}) = \sum_{m'=-l}^{l} D_{mm'}^{(l)}(\boldsymbol{a}) Y_{m'}^{(l)}(\hat{\boldsymbol{r}}), \tag{21}$$

where $\hat{\boldsymbol{r}} = \vec{r}/\|\vec{r}\|$, and $D^l$ is the irreducible representations of SO(3) to $(2l + 1) \times (2l + 1)$-dim matrices (*i.e.*, the Wigner-D matrices). This is one design criterion for SE(3)-equivariant neural networks with the spherical harmonics frame. In specific, to design an SE(3)-equivariant network, we take the following form:

$$F(\vec{r}) = W(\boldsymbol{r})Y(\hat{\boldsymbol{r}}), \tag{22}$$

where $\boldsymbol{r} = \|\vec{r}\|$, $W(\cdot)$ is the learnable function. Thus we are separating the spherical harmonics basis and the radial signal. For modeling, we only need to learn the $W(\cdot)$ on the radial. Then we use the Clebsch-Gordan tensor product for message passing on node $i$, which is:

$$\boldsymbol{v}_i = \boldsymbol{v}_i + \sum_{j \in \mathcal{N}(i)} F(\vec{\boldsymbol{r}}_{ij}) \otimes \boldsymbol{v}_j, \tag{23}$$

where $\otimes$ is the Clebsch-Gordan tensor product. **Note** that for brevity and to give the audience a high-level idea of the spherical frame basis modeling, we omit the **rotation order** and the **channel index** in Eqs. (22) and (23). First, we want to acknowledge that the rotation order is the key to conducting the message passing along tensors, and please refer to the original paper for details. Then for the channel or depth of the message passing layers (notation $c$ in the TFN paper), they are important to expand the model capacity.

To sum up, by far, we can observe that TFN only considers the pairwise information (*i.e.*, 1-hop neighborhood) for SE(3)-equivariance.

**SE(3)-Transformer**    SE(3)-Transformer [35] extends the TFN by introducing an attention score, *i.e.*,

$$\boldsymbol{v}_i = \boldsymbol{v}_i + \sum_{j \in \mathcal{N}(i)} \alpha_{ij} F(\vec{\boldsymbol{r}}_{ij}) \otimes \boldsymbol{v}_j, \tag{24}$$

where $\alpha_{ij}$ is the attention score.

To calculate the attention score, first, we need to define the following:

$$\boldsymbol{q}_i = \bigoplus_{l \geq 0} \sum_{k \geq 0} W_Q^{lk} \boldsymbol{v}_i^k, \qquad \boldsymbol{k}_{ij} = \bigoplus_{l \geq 0} \sum_{k \geq 0} F_K^{lk}(\boldsymbol{r}_j - \boldsymbol{r}_i) \otimes \boldsymbol{v}_j^k, \tag{25}$$

where $k$ and $l$ correspond to the rotation order of the input and output tensor, $W_Q$ is a learnable linear matrix, $F_K$ follows the same formation as Eq. (22), and $\bigoplus$ is the direct sum. Then we can obtain the attention coefficients with dot product as:

$$\alpha_{ij} = \frac{\exp(\boldsymbol{q}_i^T \boldsymbol{k}_{ij})}{\sum_{j' \in \mathcal{N}_i \backslash i} \exp(\boldsymbol{q}_i^T \boldsymbol{k}_{ij'})} \tag{26}$$

**Equiformer**    SE(3)-Trans adopts the dot product attention, and Equiformer [73] extends this with an MLP attention and with higher efficiency. We also want to mention that during modeling, Equiformer has an option of adding extra atom and bond information, and we set this hyperparameter as False for a fair comparison when comparing with other geometric models.

**NequIP**    Neural Equivariant Interatomic Potentials (NequIP) [3] is a follow-up of TFN, which mainly focuses on improving the force prediction. Originally, TFN was directly predicting the $l = 1$ tensor for the force prediction. In NequIP, the output only includes the $l = 1$ tensor, while the force is obtained by taking the gradient with respect to the energy.

There are also other minor architecture design updates, such as adding the skip-connection [44]. Please refer [3] for more details.

**Allegro**    Allegro [95] is a follow-up of NequIP by further modeling a local frame around each atom. In specific, the standard message-passing framework is based on the nodes (or atoms here), while Allegro focuses on the edge-level information.

**Difference with Spherical Harmonics in Invariant Modeling**    As you may notice, the invariant models also adopt the spherical harmonics (or spherical Fourier-Bessel), *e.g.*, Eq. (18) in DimeNet and Eq. (19) in SphereNet and GemNet. However, their usage of the spherical harmonics is different from the spherical frame models discussed in this section.

- In invariant models, the spherical harmonics are used for embedding the angle information, either bond angles or dihedral angles. Such angles are type-0 features, and they are invariant w.r.t. the SO(3) group. Note that this embedding is related to quantum mechanics since the spherical harmonics appear as general solutions of the Schrödinger equations.
- In the spherical frame models, the spherical harmonics are used to serve as the basis for transforming the relative coordinates into tensors, utilizing the fact that spherical harmonics are equivariant functions with respect to the SO(3) group.

Thus, they may follow the same numerical calculation, but their physical meanings are different.

## H.3    Equivariant Models with Vector Frame Basis

From a very high-level view, we can view this as first constructing the tensor and then conducting the message-passing between the type-0 tensor and type-1 tensor.

**EGNN**    E(n)-equivariant graph neural network (EGNN) [108] has a very neat design to achieve the E(n)-equivariance property. It constructs the message update function for both the atom positions and atom attributes

simultaneously. Concretely, for edge embedding $e$, input node embedding $h$ and coordinate $v = r$, the $l$-th layer updates are:

$$
\begin{aligned}
\boldsymbol{m}_{ij} &= W_e\big(\boldsymbol{h}_i^l, \boldsymbol{h}_j^l, \|\boldsymbol{v}_i^l - \boldsymbol{v}_j^l\|, \boldsymbol{e}_{ij}\big) \\
\boldsymbol{v}_i^{l+1} &= \boldsymbol{v}_i^l + \sum_{j \neq i}(\boldsymbol{v}_i^l - \boldsymbol{v}_j^l), W_v(\boldsymbol{m}_{ij}) \\
\boldsymbol{m}_i &= \sum_{j \neq i} \boldsymbol{m}_{ij} \\
h_i^{l+1} &= W_h(\boldsymbol{h}_i^l, \boldsymbol{m}_i),
\end{aligned}
\tag{27}
$$

where $W_e, W_v, W_h$ are learnable parameters. The equivariance can be proved easily and with good efficiency. However, one inherent limitation of EGNN is that it is essentially a global vector frame model and utilizes only one projection (scalarization) dimension, and it does not satisfy the reflection-antisymmetric condition for certain tasks like binding.

**PaiNN**  Polarizable atom interaction neural network (PaiNN) [110] utilizes a multi-channel vector aggregation method, which contains more expressive equivariant vector information than Eq. (27). More precisely, each node of PaiNN maintains a multi-channel vector: $\boldsymbol{v}_i \in \mathbf{R}^{F \times 3}$, where $F$ denotes the channel number. Comparing with Eq. (27), the $\boldsymbol{v}_i \in \mathbf{R}^{1 \times 3}$ of EGNN restricted the expressiveness power. [19] provides a geometric explanation of the updating method of PaiNN ((9) of [110]) by the frame transition functions between local vector frames.

**CDConv for Macromolecules**  CDConv [29] models the residue-level information of proteins. In specific, it utilizes the 3D coordinates of $C_\alpha$ as the surrogates to the residue coordinates. Then it builds an orthogonal frame based on the 1D residue sequence. The experimental results show that it can reach promising results on protein structure tasks.

# I Complete Results

In the main body, due to space limitations, we cannot provide the results on certain tasks. Here we would like to provide more comprehensive results.

For the results not listed either in the main body or in this section, there are two possible reasons for us to exclude them: (1) We cannot reproduce them using the reported hyperparameters in the original paper, and we may need to do more hyperparameter tuning as the next steps. (2) Some models are too large to fit in the GPU memory, even with batch-size=1.

## I.1 Small Molecules: MD17 and rMD17

In Table 2, we select 6 subtasks in MD17 and 6 subtasks in rMD17. Next we will show the complete results of MD17 and rMD17 are in Tables 13 and 14.

Table 13: Results on 8 energy ($\frac{kcal}{mol}$) and force ($\frac{kcal}{mol \cdot Å}$) prediction tasks in MD17. The evaluation is the mean absolute error. No data normalization is used.

| Model | Energy/Force | Aspirin ↓ | Benzene ↓ | Ethanol ↓ | Malonaldehyde ↓ | Naphthalene ↓ | Salicylic ↓ | Toluene ↓ | Uracil ↓ |
|---|---|---|---|---|---|---|---|---|---|
| SchNet | Energy | 0.475 | 0.117 | 0.109 | 0.300 | 0.167 | 0.212 | 0.149 | 0.170 |
| | Force | 1.203 | 0.380 | 0.386 | 0.794 | 0.587 | 0.826 | 0.568 | 0.773 |
| DimeNet++ | Energy | 4.168 | 0.893 | 1.238 | 1.385 | 1.846 | 2.445 | 1.484 | 1.522 |
| | Force | 7.212 | 0.603 | 0.753 | 1.842 | 8.515 | 1.752 | 1.037 | 1.632 |
| EGNN | Energy | 17.892 | 1.142 | 0.436 | 0.896 | 12.177 | 6.964 | 4.051 | 0.854 |
| | Force | 3.042 | 0.736 | 0.924 | 1.566 | 1.136 | 1.177 | 1.202 | 1.367 |
| PaiNN | Energy | 27.626 | 0.095 | 0.063 | 0.102 | 0.622 | 0.371 | 0.165 | 0.111 |
| | Force | 0.572 | 0.053 | 0.230 | 0.338 | 0.132 | 0.288 | 0.141 | 0.201 |
| GemNet-T | Energy | 0.684 | 0.097 | 4.598 | 4.966 | 0.482 | 0.128 | 0.098 | 1.349 |
| | Force | 0.558 | 0.089 | 0.219 | 0.433 | 0.212 | 0.326 | 0.174 | 486.612 |
| SphereNet | Energy | 0.244 | 0.107 | 1.603 | 1.559 | 0.167 | 0.188 | 0.113 | 7.115 |
| | Force | 0.546 | 0.135 | 0.168 | 0.667 | 0.315 | 0.479 | 0.194 | 0.442 |
| SEGNN | Energy | 17.774 | 0.086 | 0.151 | 0.247 | 0.655 | 2.173 | 0.624 | 0.259 |
| | Force | 9.003 | 0.265 | 0.893 | 1.249 | 0.895 | 2.220 | 1.138 | 0.948 |
| NequIP | Energy | 8.333 | 0.355 | 0.971 | 2.293 | 1.032 | 2.952 | 1.303 | 1.266 |
| | Force | 23.769 | 2.383 | 5.832 | 12.099 | 5.247 | 14.048 | 6.800 | 8.060 |
| Allegro | Energy | 1.138 | 0.154 | 0.258 | 1.330 | 0.824 | 1.114 | 0.441 | 0.613 |
| | Force | 3.405 | 0.823 | 1.412 | 4.191 | 3.743 | 4.934 | 1.968 | 3.544 |
| Equiformer | Energy | 0.308 | 0.075 | 0.096 | 0.183 | 0.097 | 0.189 | 0.209 | 0.106 |
| | Force | 0.286 | 0.045 | 0.142 | 0.230 | 0.068 | 0.200 | 0.080 | 0.141 |

Table 14: Results on 10 energy ($\frac{kcal}{mol}$) and force ($\frac{kcal}{mol \cdot Å}$) prediction tasks in rMD17. The evaluation is the mean absolute error. Data normalization is used.

| Model | Energy/Force | Aspirin ↓ | Azobenzene ↓ | Benzene ↓ | Ethanol ↓ | Malonaldehyde ↓ | Naphthalene ↓ | Paracetamol ↓ | Salicylic ↓ | Toluene ↓ | Uracil ↓ |
|---|---|---|---|---|---|---|---|---|---|---|---|
| SchNet | Energy | 0.534 | 1.818 | 0.111 | 1.757 | 0.260 | 0.124 | 8.138 | 2.618 | 0.119 | 7.029 |
| | Force | 1.243 | 3.596 | 0.233 | 0.449 | 0.862 | 0.587 | 2.320 | 0.878 | 0.574 | 0.762 |
| DimeNet++ | Energy | 2.438 | 3.955 | 0.741 | 1.456 | 2.317 | 1.648 | 2.261 | 1.555 | 1.210 | 2.320 |
| | Force | 2.009 | 1.243 | 0.340 | 1.213 | 7.029 | 0.629 | 1.047 | 0.934 | 0.921 | 3.181 |
| EGNN | Energy | 17.350 | 21.333 | 0.315 | 0.402 | 0.534 | 12.164 | 26.902 | 7.794 | 15.021 | 1.669 |
| | Force | 3.825 | 2.330 | 0.529 | 0.989 | 1.334 | 1.183 | 2.313 | 1.571 | 1.165 | 1.323 |
| PaiNN | Energy | 30.156 | 0.107 | 0.010 | 1.170 | 0.070 | 5.297 | 0.117 | 5.219 | 0.045 | 2.478 |
| | Force | 0.573 | 0.326 | 0.032 | 0.316 | 0.377 | 0.161 | 0.440 | 0.321 | 0.231 | 0.235 |
| GemNet-T | Energy | 5.389 | 7.770 | 0.007 | 1.615 | 9.496 | 0.031 | 2.173 | 21.411 | 959.745 | 994.036 |
| | Force | 0.555 | 0.347 | 0.033 | 0.233 | 0.337 | 0.154 | 0.388 | 0.371 | 0.400 | 1.165 |
| SphereNet | Energy | 0.304 | 0.257 | 0.052 | 0.072 | 0.138 | 0.093 | 0.183 | 0.771 | 20.479 | 12.211 |
| | Force | 0.622 | 0.532 | 0.076 | 0.217 | 0.500 | 0.279 | 0.482 | 2.088 | 0.254 | 0.959 |
| SEGNN | Energy | 15.721 | 3.474 | 0.270 | 0.130 | 0.182 | 1.110 | 4.197 | 1.494 | 0.814 | 1.115 |
| | Force | 8.549 | 2.579 | 0.174 | 0.846 | 1.185 | 0.926 | 3.191 | 2.056 | 1.241 | 0.966 |
| NequIP | Energy | 9.618 | 1.993 | 3.048 | 0.936 | 2.313 | 2.089 | 5.136 | 3.302 | 1.306 | 1.738 |
| | Force | 22.904 | 6.406 | 1.523 | 6.027 | 12.372 | 5.529 | 17.574 | 15.693 | 7.094 | 10.220 |
| Allegro | Energy | 1.366 | 0.872 | 0.029 | 1.002 | 0.417 | 1.756 | 0.944 | 1.035 | 0.437 | 0.387 |
| | Force | 3.186 | 2.763 | 0.237 | 2.799 | 2.125 | 3.815 | 3.081 | 4.781 | 2.048 | 1.939 |
| Equiformer | Energy | 0.375 | 0.127 | 0.027 | 0.064 | 0.085 | 0.069 | 0.215 | 0.143 | 0.104 | 0.200 |
| | Force | 0.305 | 0.132 | 0.020 | 0.162 | 0.240 | 0.070 | 0.258 | 0.218 | 0.077 | 0.149 |

## I.2 Geometric Pretraining

**Single-modal Pretraining.** Recent studies have started to explore **single-modal geometric pretraining** on molecules. The GeoSSL paper [80] covers a wide range of geometric pretraining algorithms. The type prediction, distance prediction, and angle prediction predict the masked atom type, pairwise distance, and bond angle, respectively. The 3D InfoGraph predicts whether the node- and graph-level 3D representation are for the same molecule. GeoSSL is a novel geometric pretraining paradigm that maximizes the mutual information (MI) between the original conformation $g_1$ and augmented conformation $g_2$, where $g_2$ is obtained by adding small perturbations to $g_1$. RR, InfoNCE, and EBM-NCE optimize the objective in the latent representation space, either generative or contrastive. GeoSSL-DDM [80] optimizes the same objective function using denoising score matching. GeoSSL-DDM-1L [136] is a special case of GeoSSL-DDM with one layer of denoising. 3D-EMGP [60] geometric pretraining is specifically built on equivariant models, and the goal is to denoise the 3D coordinates directly using a diffusion model. We illustrate these seven algorithms in Fig. 4.

**2D-3D Multi-modal Pretraining.** Another promising direction is the **multi-modal pretraining on topology and geometry**. GraphMVP [86] first proposes one contrastive objective (EBM-NCE) and one generative objective (variational representation reconstruction, VRR) to optimize the mutual information between the 2D and 3D modalities. Specifically, VRR does the 2D and 3D reconstruction in the latent space. 3D InfoMax [114] is a special case of GraphMVP, with the contrastive part only. MoleculeSDE [79] extends GraphMVP by introducing two SDE models for solving the 2D and 3D reconstruction. An illustration of them is in Fig. 8.

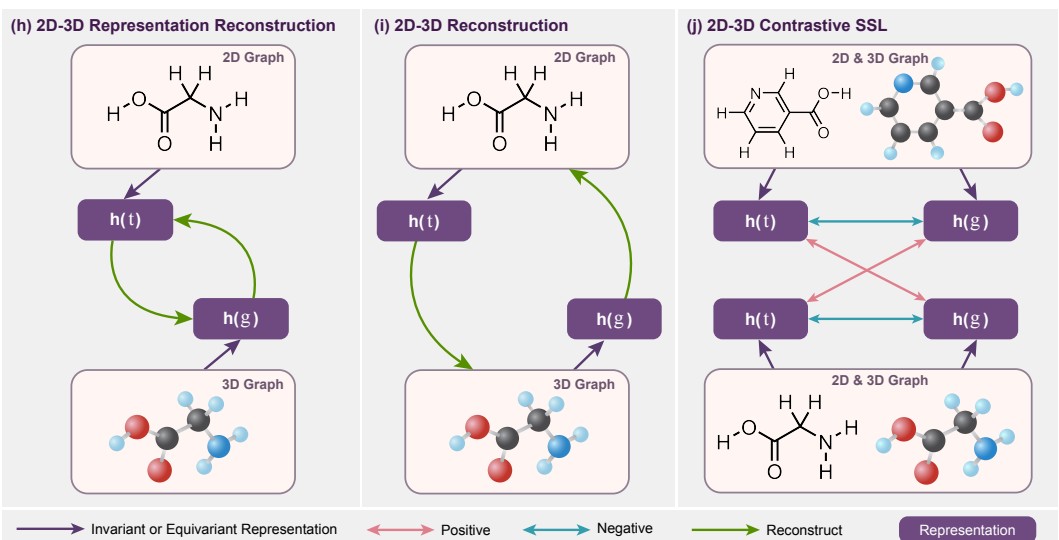

Figure 8: Pipelines for three multi-modal geometric pretraining methods.

In Table 7, we show the pretraining results of using **SchNet as the backbone** and fine-tuning on QM9. The pretraining results of using **SchNet as the backbone** and fine-tuning on MD17 are in Table 15. The pretraining results of using **PaiNN as the backbone** and fine-tuning on QM9 and MD17 are in Tables 16 and 17. For MD17, as will be discussed in Appendix J, we do not consider the data normalization trick. Notice that some pretraining results are skipped due to the collapsed performance.

Table 15: Pretraining results on eight force prediction tasks from MD17, and the backbone model is SchNet. We take 1K for training, 1K for validation, and 48K to 991K molecules for the test concerning different tasks. The evaluation is mean absolute error, and the best results are marked in **bold** and **bold**, respectively.

| Pretraining | Aspirin ↓ | Benzene ↓ | Ethanol ↓ | Malonaldehyde ↓ | Naphthalene ↓ | Salicylic ↓ | Toluene ↓ | Uracil ↓ |
|---|---|---|---|---|---|---|---|---|
| – (random init) | 1.203 | 0.380 | 0.386 | 0.794 | 0.587 | 0.826 | 0.568 | 0.773 |
| Supervised | 1.867 | 0.434 | 0.566 | 1.106 | 0.637 | 13.037 | 0.607 | 0.759 |
| Type Prediction | 1.383 | 0.402 | 0.450 | 0.879 | 0.622 | 1.028 | 0.662 | 0.840 |
| Distance Prediction | 1.427 | 0.396 | 0.434 | 0.818 | 0.793 | 0.952 | 0.509 | 1.567 |
| Angle Prediction | 1.542 | 0.447 | 0.669 | 1.022 | 0.680 | 1.032 | 0.623 | 0.768 |
| 3D InfoGraph | 1.610 | 0.415 | 0.560 | 0.900 | 0.788 | 1.278 | 0.768 | 1.110 |
| GeoSSL-RR | 1.215 | 0.393 | 0.514 | 1.092 | 0.596 | 0.847 | 0.570 | 0.711 |
| GeoSSL-InfoNCE | 1.132 | 0.395 | 0.466 | 0.888 | 0.542 | 0.831 | 0.554 | 0.664 |
| GeoSSL-EBM-NCE | 1.251 | 0.373 | 0.457 | 0.829 | 0.512 | 0.990 | 0.560 | 0.742 |
| 3D InfoMax | 1.142 | 0.388 | 0.469 | 0.731 | 0.785 | 0.798 | 0.516 | 0.640 |
| GraphMVP | 1.126 | 0.377 | 0.430 | 0.726 | 0.498 | 0.740 | 0.508 | 0.620 |
| GeoSSL-DDM-1L | 1.364 | 0.391 | 0.432 | 0.830 | 0.599 | 0.817 | 0.628 | 0.607 |
| GeoSSL-DDM | **1.107** | 0.360 | 0.357 | 0.737 | 0.568 | 0.902 | 0.484 | 0.502 |
| MoleculeSDE (VE) | **1.112** | **0.304** | **0.282** | **0.520** | **0.455** | **0.725** | **0.515** | **0.447** |
| MoleculeSDE (VP) | 1.244 | **0.315** | **0.338** | **0.488** | **0.432** | **0.712** | **0.478** | 0.468 |

Table 16: Pretraining results on 12 quantum mechanics prediction tasks from QM9, and the backbone model is PaiNN. We take 110K for training, 10K for validation, and 11K for testing. The evaluation is mean absolute error, and the best and the second best results are marked in **bold** and **bold**, respectively.

| Pretraining | $\alpha$ ↓ | $\nabla\mathcal{E}$ ↓ | $\mathcal{E}_{\text{HOMO}}$ ↓ | $\mathcal{E}_{\text{LUMO}}$ ↓ | $\mu$ ↓ | $C_v$ ↓ | $G$ ↓ | $H$ ↓ | $R^2$ ↓ | $U$ ↓ | $U_0$ ↓ | ZPVE ↓ |
|---|---|---|---|---|---|---|---|---|---|---|---|---|
| – | 0.049 | 42.73 | 24.46 | 20.16 | 0.016 | 0.025 | 8.43 | 7.88 | 0.169 | 8.18 | 7.63 | 1.419 |
| Supervised | 0.161 | 64.30 | 23.41 | 19.31 | 0.015 | 0.024 | 9.01 | 9.53 | 0.152 | 16.17 | 9.43 | 1.470 |
| Distance Prediction | 0.049 | 37.23 | 22.75 | 18.26 | 0.014 | 0.030 | 9.31 | 9.35 | 0.143 | 9.85 | 9.07 | 1.566 |
| 3D InfoGraph | 0.047 | 44.25 | 24.06 | 18.54 | 0.015 | 0.052 | 8.81 | 7.97 | 0.143 | 8.68 | 8.08 | 1.416 |
| GeoSSL-RR | 0.046 | 41.20 | 23.93 | 19.36 | 0.016 | 0.025 | 8.32 | 8.17 | 0.174 | 7.99 | 8.20 | 1.438 |
| GeoSSL-InfoNCE | 0.045 | 39.29 | 23.23 | 18.40 | 0.015 | 0.024 | 8.34 | 8.37 | **0.127** | 7.45 | 8.34 | 1.356 |
| GeoSSL-EBM-NCE | 0.045 | 38.87 | 22.71 | 17.89 | 0.014 | 0.082 | 8.28 | 7.35 | 0.130 | 7.85 | 7.68 | 1.338 |
| 3D InfoMax | 0.046 | 36.97 | 21.31 | 17.69 | 0.014 | 0.024 | 8.38 | 7.36 | 0.135 | 8.60 | 7.99 | 1.453 |
| GraphMVP | 0.044 | 36.03 | 20.71 | 17.02 | 0.014 | 0.024 | 8.31 | 7.36 | 0.132 | 7.57 | 7.34 | 1.337 |
| GeoSSL-DDM-1L | 0.045 | 36.13 | 20.59 | 17.26 | 0.014 | 0.024 | 9.45 | 8.43 | 0.128 | 8.88 | 8.16 | 1.380 |
| GeoSSL-DDM | **0.043** | 35.55 | 20.57 | **16.95** | 0.014 | 0.024 | 8.25 | 7.42 | **0.127** | 7.36 | 7.34 | 1.334 |
| 3D-EMGP (Gaussian) | 0.277 | 40.56 | 21.25 | 23.99 | 0.014 | 0.039 | 9.16 | 9.14 | 0.340 | 9.31 | 8.59 | 1.433 |
| MoleculeSDE (VE) | 0.044 | **34.67** | **20.14** | 17.05 | **0.013** | **0.023** | **7.64** | **7.05** | 0.139 | **6.88** | **6.79** | **1.273** |
| MoleculeSDE (VP) | **0.042** | **35.09** | **20.14** | **16.78** | **0.013** | **0.023** | 8.17 | **7.01** | 0.133 | **7.30** | **7.05** | 1.315 |

Table 17: Results on eight force prediction tasks from MD17, and the backbone model is PaiNN. We take 1K for training, 1K for validation, and 48K to 991K molecules for the test concerning different tasks. The evaluation is mean absolute error, and the best results are marked in **bold** and **bold**, respectively.

| Pretraining | Aspirin ↓ | Benzene ↓ | Ethanol ↓ | Malonaldehyde ↓ | Naphthalene ↓ | Salicylic ↓ | Toluene ↓ | Uracil ↓ |
|---|---|---|---|---|---|---|---|---|
| – | 0.572 | 0.053 | 0.230 | 0.338 | 0.132 | 0.288 | 0.141 | 0.201 |
| Supervised | 0.509 | 0.056 | 0.181 | 0.330 | – | 0.284 | 0.163 | – |
| Distance Prediction | 0.480 | 0.053 | 0.200 | 0.296 | 0.131 | 0.265 | 0.171 | 0.168 |
| 3D InfoGraph | 0.554 | 0.067 | 0.249 | 0.353 | 0.177 | 0.331 | 0.179 | 0.213 |
| GeoSSL-RR | 0.559 | 0.051 | 0.262 | 0.368 | 0.146 | 0.303 | 0.154 | 0.202 |
| GeoSSL-InfoNCE | 0.428 | 0.051 | 0.197 | 0.337 | 0.127 | 0.247 | 0.136 | 0.169 |
| GeoSSL-EBM-NCE | 0.435 | 0.048 | 0.198 | **0.295** | 0.143 | 0.245 | 0.132 | 0.172 |
| 3D InfoMax | 0.479 | 0.052 | 0.220 | 0.344 | 0.138 | 0.267 | 0.155 | 0.174 |
| GraphMVP | 0.465 | 0.050 | 0.205 | 0.316 | **0.119** | 0.242 | 0.136 | 0.168 |
| GeoSSL-DDM-1L | 0.436 | 0.048 | 0.209 | 0.320 | **0.119** | 0.249 | 0.132 | 0.177 |
| GeoSSL-DDM | **0.427** | 0.047 | **0.188** | 0.313 | 0.120 | **0.240** | **0.129** | 0.167 |
| 3D-EMGP (Gaussian) | 0.487 | 0.048 | 0.217 | 0.329 | 0.151 | 0.299 | 0.141 | 0.182 |
| MoleculeSDE (VE) | **0.421** | **0.043** | 0.195 | **0.284** | **0.105** | **0.236** | **0.123** | **0.158** |
| MoleculeSDE (VP) | 0.443 | **0.045** | **0.191** | 0.301 | 0.131 | 0.261 | 0.140 | **0.159** |

# J  Ablation Studies

We have the following challenges in the literature: (1) Different data preprocessors, including data augmentations and normalization strategies. (2) Different data splits, *i.e.*, with different seeds or different train-valid-test sizes. (3) Different running epochs. (4) Different optimizers (SGD, Adam, or EMA) and learning rate schedulers (cosine annealing or cyclic). These factors can significantly affect performance, and Geom3D is a useful tool for careful scrutinization. In this section, we are mainly focusing on the first point, *i.e.*, the tricks that are mainly related to the specific applications. For the following factors, we adopt a fixed setting, *i.e.*, the same seeds for tasks if using random splits, fixed epochs for most of the geometric modelings, Adam as the optimizer, and cosine annealing learning rate scheduler. We would like to acknowledge that the EMA optimizer and cyclic learning rate scheduler can be beneficial for certain geometric models, yet this is more related to the optimization process and is beyond the scope of this work. We will explore this in future work.

## J.1 Ablation Studies on the Effect of Latent Dimension $d$

Recent works [117, 121] have found that the latent dimensions play an important role in molecule pretraining, and here we list the comparison between latent dimension $d = 128$ and latent dimension $d = 300$.

- The performance comparison for QM9 is in Table 18, and we visually plot the performance gap MAE($d = 128$) - MAE($d = 300$) in Fig. 9. The results with $d = 300$ are reported in Table 1.
- The performance **(w/ normalization)** comparison for MD17 and rMD17 is in Tables 19 to 22. The results with $d = 300$ are reported in Tables 2, 13 and 14 except NequIP and Allegro. Their results in Appendix J.3 **(w/ normalization)** are reported Table 2.
- The performance comparison for COLL is in Table 23, and results with $d = 300$ are reported in Table 3.
- The performance comparison for LBA & LEP is in Tables 24 and 25, and results with $d = 300$ are reported in Table 4.

Table 18: Ablation studies of latent dimension $d$ on QM9. 110K for training, 10K for validation, and 11K for testing. The evaluation metric is the mean absolute error (MAE).

| Model | $d$ | $\alpha \downarrow$ | $\nabla \mathcal{E} \downarrow$ | $\mathcal{E}_{\text{HOMO}} \downarrow$ | $\mathcal{E}_{\text{LUMO}} \downarrow$ | $\mu \downarrow$ | $C_v \downarrow$ | $G \downarrow$ | $H \downarrow$ | $R^2 \downarrow$ | $U \downarrow$ | $U_0 \downarrow$ | ZPVE $\downarrow$ |
|---|---|---|---|---|---|---|---|---|---|---|---|---|---|
| SchNet | 128 | 0.068 | 49.66 | 31.91 | 26.09 | 0.030 | 0.032 | 14.17 | 14.16 | 0.126 | 14.11 | 14.27 | 1.684 |
| | 300 | 0.060 | 44.13 | 27.64 | 22.55 | 0.028 | 0.031 | 14.19 | 14.05 | 0.133 | 13.93 | 13.27 | 1.749 |
| DimeNet++ | 128 | 0.046 | 37.93 | 20.99 | 17.50 | 0.028 | 0.022 | 7.33 | 6.72 | 0.299 | 6.38 | 7.26 | 1.260 |
| | 300 | 0.044 | 36.22 | 20.01 | 16.66 | 0.028 | 0.022 | 7.45 | 6.14 | 0.323 | 6.33 | 7.18 | 1.118 |
| SE(3)-Trans | 128 | 0.144 | 55.36 | 34.59 | 34.05 | 0.051 | 0.064 | 64.85 | 76.32 | 1.763 | 69.73 | 68.22 | 5.448 |
| | 300 | 0.137 | 56.52 | 34.65 | 34.41 | 0.050 | 0.063 | 65.28 | 70.70 | 1.747 | 68.92 | 68.88 | 5.428 |
| EGNN | 128 | 0.065 | 49.07 | 29.19 | 25.00 | 0.028 | 0.031 | 11.61 | 10.52 | 0.074 | 10.51 | 10.61 | 1.544 |
| | 300 | 0.062 | 49.56 | 30.08 | 24.98 | 0.028 | 0.030 | 10.01 | 9.14 | 0.089 | 9.28 | 9.08 | 1.519 |
| PaiNN | 128 | 0.049 | 44.02 | 25.92 | 20.87 | 0.016 | 0.025 | 10.32 | 7.30 | 0.126 | 7.60 | 7.51 | 1.295 |
| | 300 | 0.049 | 42.73 | 24.46 | 20.16 | 0.016 | 0.025 | 8.43 | 7.88 | 0.169 | 8.18 | 7.63 | 1.419 |
| GemNet-T | 128 | 0.042 | 34.49 | 17.82 | 14.80 | 0.020 | 0.021 | 8.48 | 7.05 | 0.246 | 6.94 | 6.97 | 1.201 |
| | 300 | 0.041 | 35.46 | 17.85 | 15.86 | 0.021 | 0.023 | 7.61 | 7.08 | 0.271 | 6.42 | 5.88 | 1.232 |
| SphereNet | 128 | 0.050 | 40.36 | 22.49 | 19.29 | 0.026 | 0.026 | 9.06 | 7.49 | 0.248 | 7.53 | 7.79 | 1.560 |
| | 300 | 0.047 | 38.93 | 21.45 | 18.25 | 0.027 | 0.025 | 8.16 | 13.68 | 0.288 | 6.77 | 7.43 | 1.295 |
| SEGNN | 128 | 0.056 | 41.40 | 22.40 | 20.77 | 0.024 | 0.029 | 13.11 | 12.99 | 0.481 | 13.82 | 13.71 | 1.596 |
| | 300 | 0.048 | 33.61 | 17.66 | 17.01 | 0.021 | 0.026 | 11.60 | 12.45 | 0.404 | 11.29 | 12.20 | 1.590 |
| Equiformer | 128 | 0.051 | 33.52 | 17.58 | 16.83 | 0.015 | 0.023 | 17.13 | 13.14 | 0.408 | 15.23 | 13.63 | 2.182 |
| | 300 | 0.051 | 33.46 | 17.93 | 16.85 | 0.015 | 0.023 | 14.49 | 14.60 | 0.433 | 14.88 | 13.78 | 2.342 |

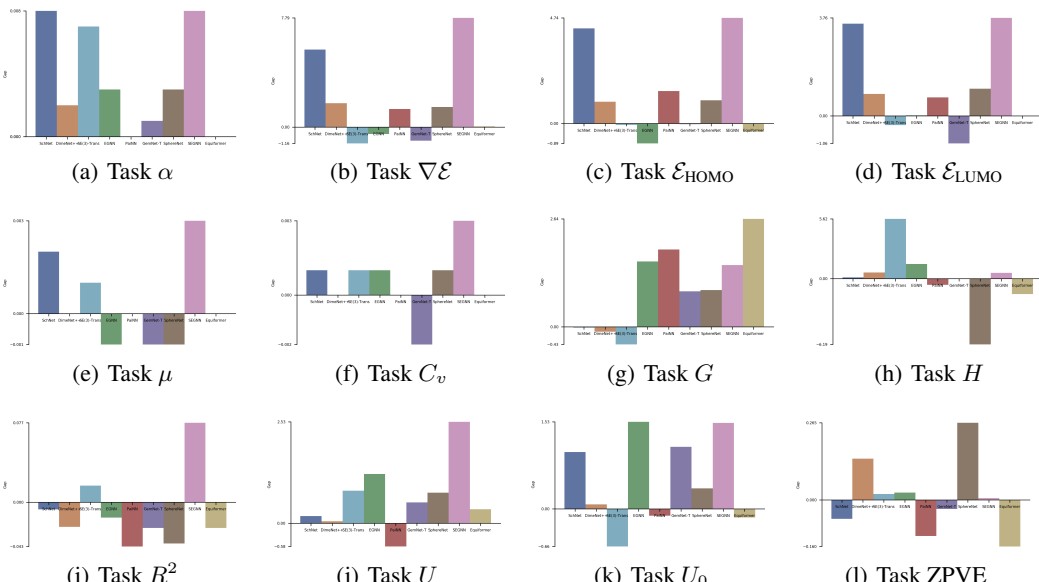

(a) Task $\alpha$    (b) Task $\nabla \mathcal{E}$    (c) Task $\mathcal{E}_{\text{HOMO}}$    (d) Task $\mathcal{E}_{\text{LUMO}}$

(e) Task $\mu$    (f) Task $C_v$    (g) Task $G$    (h) Task $H$

(i) Task $R^2$    (j) Task $U$    (k) Task $U_0$    (l) Task ZPVE

Figure 9: Performance gap of MAE($d = 128$) - MAE($d = 300$) in QM9.

Table 19: Ablation studies of latent dimension ($d = 128$) on MD17. The evaluation is the mean absolute error. No data normalization is used.

| Model | Energy / Force | Aspirin ↓ | Benzene ↓ | Ethanol ↓ | Malonaldehyde ↓ | Naphthalene ↓ | Salicylic ↓ | Toluene ↓ | Uracil ↓ |
|---|---|---|---|---|---|---|---|---|---|
| SchNet | Energy | 0.695 | 0.118 | 0.182 | 0.338 | 0.210 | 0.232 | 0.157 | 0.192 |
| | Force | 1.500 | 0.399 | 0.663 | 1.157 | 0.759 | 0.896 | 0.594 | 0.906 |
| DimeNet++ | Energy | 2.104 | 1.053 | 0.971 | 1.180 | 1.472 | 1.901 | 1.988 | 1.754 |
| | Force | 2.209 | 0.476 | 0.636 | 1.420 | 1.293 | 1.071 | 0.924 | 1.070 |
| EGNN | Energy | 91.490 | 0.663 | 1.439 | 1.385 | 17.064 | 31.006 | 7.190 | 1.409 |
| | Force | 19.211 | 1.049 | 1.983 | 2.380 | 2.185 | 3.957 | 2.453 | 2.172 |
| PaiNN | Energy | 0.209 | 0.097 | 0.070 | 0.093 | 0.235 | 0.127 | 0.133 | 0.107 |
| | Force | 0.549 | 0.053 | 0.198 | 0.328 | 0.134 | 0.284 | 0.146 | 0.180 |
| GemNet-T | Energy | 1.299 | 0.096 | 8.418 | 0.101 | 0.116 | 0.141 | 0.095 | 11.270 |
| | Force | 0.518 | 0.050 | 0.226 | 0.380 | 0.107 | 0.259 | 0.118 | 542.330 |
| SphereNet | Energy | 0.235 | 0.104 | 0.327 | 0.136 | 0.183 | 0.771 | 0.116 | 0.147 |
| | Force | 0.500 | 0.114 | 0.199 | 0.377 | 0.416 | 2.033 | 0.198 | 0.303 |
| SEGNN | Energy | 10.030 | 0.081 | 0.088 | 0.191 | 0.678 | 1.699 | 0.541 | 0.260 |
| | Force | 6.793 | 0.193 | 0.456 | 0.832 | 0.734 | 1.828 | 0.957 | 0.654 |
| Allegro | Energy | 2.380 | 0.278 | 0.386 | 0.583 | 0.732 | 1.131 | 0.615 | 1.357 |
| | Force | 6.537 | 1.777 | 1.916 | 2.572 | 3.359 | 5.063 | 3.022 | 6.974 |
| Equiformer | Energy | 0.708 | 0.076 | 0.056 | 0.102 | 0.097 | 0.191 | 0.094 | 0.103 |
| | Force | 0.282 | 0.044 | 0.142 | 0.229 | 0.068 | 0.202 | 0.080 | 0.140 |

Table 20: Ablation studies of latent dimension ($d = 300$) on MD17. The evaluation is the mean absolute error. No data normalization is used.

| Model | Energy/Force | Aspirin ↓ | Benzene ↓ | Ethanol ↓ | Malonaldehyde ↓ | Naphthalene ↓ | Salicylic ↓ | Toluene ↓ | Uracil ↓ |
|---|---|---|---|---|---|---|---|---|---|
| SchNet | Energy | 0.475 | 0.117 | 0.109 | 0.300 | 0.167 | 0.212 | 0.149 | 0.170 |
| | Force | 1.203 | 0.380 | 0.386 | 0.794 | 0.587 | 0.826 | 0.568 | 0.773 |
| DimeNet++ | Energy | 4.168 | 0.893 | 1.238 | 1.385 | 1.846 | 2.445 | 1.484 | 1.522 |
| | Force | 7.212 | 0.603 | 0.753 | 1.842 | 8.515 | 1.752 | 1.037 | 1.632 |
| EGNN | Energy | 17.892 | 1.142 | 0.436 | 0.896 | 12.177 | 6.964 | 4.051 | 0.854 |
| | Force | 3.042 | 0.736 | 0.924 | 1.566 | 1.136 | 1.177 | 1.202 | 1.367 |
| PaiNN | Energy | 27.626 | 0.095 | 0.063 | 0.102 | 0.622 | 0.371 | 0.165 | 0.111 |
| | Force | 0.572 | 0.053 | 0.230 | 0.338 | 0.132 | 0.288 | 0.141 | 0.201 |
| GemNet-T | Energy | 0.684 | 0.097 | 4.598 | 4.966 | 0.482 | 0.128 | 0.098 | 1.349 |
| | Force | 0.558 | 0.089 | 0.219 | 0.433 | 0.212 | 0.326 | 0.174 | 486.612 |
| SphereNet | Energy | 0.244 | 0.107 | 1.603 | 1.559 | 0.167 | 0.188 | 0.113 | 7.115 |
| | Force | 0.546 | 0.135 | 0.168 | 0.667 | 0.315 | 0.479 | 0.194 | 0.442 |
| SEGNN | Energy | 17.774 | 0.086 | 0.151 | 0.247 | 0.655 | 2.173 | 0.624 | 0.259 |
| | Force | 9.003 | 0.265 | 0.893 | 1.249 | 0.895 | 2.220 | 1.138 | 0.948 |
| Allegro | Energy | 1.577 | 0.117 | 0.308 | 0.481 | 0.899 | 1.088 | 0.406 | 0.490 |
| | Force | 4.328 | 0.358 | 1.613 | 2.185 | 3.841 | 4.731 | 1.866 | 2.627 |
| Equiformer | Energy | 0.308 | 0.075 | 0.096 | 0.183 | 0.097 | 0.189 | 0.209 | 0.106 |
| | Force | 0.286 | 0.045 | 0.142 | 0.230 | 0.068 | 0.200 | 0.080 | 0.141 |

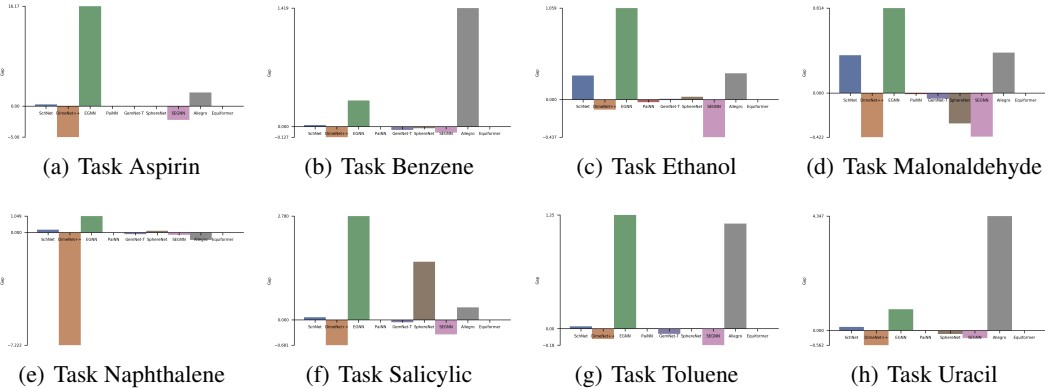

(a) Task Aspirin  (b) Task Benzene  (c) Task Ethanol  (d) Task Malonaldehyde

(e) Task Naphthalene  (f) Task Salicylic  (g) Task Toluene  (h) Task Uracil

Figure 10: Performance gap of MAE($d = 128$) - MAE($d = 300$) in MD17.

Table 21:   Ablation studies of latent dimension (dim=128) on rMD17. The evaluation is the mean absolute error. No data normalization is used.

| model | Energy / Force | Aspirin ↓ | Azobenzene ↓ | Benzene ↓ | Ethanol ↓ | Malonaldehyde ↓ | Naphthalene ↓ | Paracetamol ↓ | Salicylic ↓ | Toluene ↓ | Uracil ↓ |
|---|---|---|---|---|---|---|---|---|---|---|---|
| SchNet | Energy | 0.764 | 1.332 | 0.388 | 0.226 | 0.380 | 0.205 | 1.298 | 1.759 | 0.166 | 0.173 |
| | Force | 1.662 | 3.071 | 0.318 | 0.782 | 1.109 | 0.805 | 2.344 | 0.950 | 0.693 | 0.943 |
| DimeNet++ | Energy | 1.719 | 4.806 | 0.506 | 10.867 | 0.845 | 1.209 | 10.876 | 2.020 | 1.519 | 4227.668 |
| | Force | 1.253 | 1.033 | 0.307 | 20.860 | 0.632 | 0.602 | 1.123 | 1.022 | 0.991 | 33549.676 |
| EGNN | Energy | 89.661 | 51.554 | 4.893 | 1.065 | 9.339 | 32.901 | 77.996 | 27.114 | 12.766 | 4.519 |
| | Force | 20.531 | 4.436 | 0.912 | 2.305 | 3.056 | 2.287 | 9.484 | 13.117 | 2.567 | 2.482 |
| PaiNN | Energy | 1.949 | 5.733 | 0.036 | 0.606 | 1.626 | 2.610 | 0.541 | 0.831 | 0.158 | 0.181 |
| | Force | 3.189 | 0.940 | 0.143 | 0.727 | 1.158 | 0.851 | 1.636 | 1.450 | 0.682 | 0.875 |
| GemNet-T | Energy | 1.546 | 0.073 | 0.006 | 1.060 | 6.610 | 0.025 | 1.972 | 14.837 | 0.023 | 36.966 |
| | Force | 0.555 | 0.265 | 0.026 | 0.211 | 0.425 | 0.112 | 0.368 | 0.308 | 0.120 | 0.233 |
| SphereNet | Energy | 21.142 | 0.542 | 0.678 | 1.226 | 0.423 | 0.176 | 0.255 | 6.218 | 0.119 | 0.143 |
| | Force | 0.666 | 0.781 | 0.102 | 0.313 | 0.419 | 0.500 | 0.659 | 2.244 | 0.334 | 0.425 |
| SEGNN | Energy | 11.828 | 2.729 | 0.018 | 0.081 | 0.161 | 1.333 | 3.982 | 1.476 | 1.443 | 0.221 |
| | Force | 7.543 | 2.014 | 0.139 | 0.509 | 0.934 | 0.845 | 3.338 | 1.934 | 1.028 | 0.723 |
| Allegro | Energy | 6.142 | 2.221 | 0.094 | 0.465 | 0.592 | 1.320 | 2.196 | 1.239 | 0.584 | 1.739 |
| | Force | 4.891 | 5.727 | 0.960 | 2.166 | 2.630 | 3.546 | 4.571 | 5.949 | 2.885 | 6.610 |
| Equiformer | Energy | 0.480 | 0.119 | 0.031 | 0.085 | 0.098 | 0.065 | 0.848 | 0.261 | 0.082 | 0.214 |
| | Force | 0.303 | 0.132 | 0.020 | 0.163 | 0.242 | 0.069 | 0.260 | 0.217 | 0.077 | 0.150 |

Table 22:   Ablation studies of latent dimension ($d = 300$) on rMD17. The evaluation is the mean absolute error. No data normalization is used.

| Model | Energy/Force | Aspirin ↓ | Azobenzene ↓ | Benzene ↓ | Ethanol ↓ | Malonaldehyde ↓ | Naphthalene ↓ | Paracetamol ↓ | Salicylic ↓ | Toluene ↓ | Uracil ↓ |
|---|---|---|---|---|---|---|---|---|---|---|---|
| SchNet | Energy | 0.534 | 1.818 | 0.111 | 1.757 | 0.260 | 0.124 | 8.138 | 2.618 | 0.119 | 7.029 |
| | Force | 1.243 | 3.596 | 0.233 | 0.449 | 0.862 | 0.587 | 2.320 | 0.878 | 0.574 | 0.762 |
| DimeNet++ | Energy | 2.438 | 3.955 | 0.741 | 1.456 | 2.317 | 1.648 | 2.261 | 1.555 | 1.210 | 2.320 |
| | Force | 2.009 | 1.243 | 0.340 | 1.213 | 7.029 | 0.629 | 1.047 | 0.934 | 0.921 | 3.181 |
| EGNN | Energy | 17.350 | 21.333 | 0.315 | 0.402 | 0.534 | 12.164 | 26.902 | 7.794 | 15.021 | 1.669 |
| | Force | 3.825 | 2.330 | 0.529 | 0.989 | 1.334 | 1.183 | 2.313 | 1.571 | 1.165 | 1.323 |
| PaiNN | Energy | 30.156 | 0.107 | 0.010 | 1.170 | 0.070 | 5.297 | 0.117 | 5.219 | 0.045 | 2.478 |
| | Force | 0.573 | 0.326 | 0.032 | 0.316 | 0.377 | 0.161 | 0.440 | 0.321 | 0.231 | 0.235 |
| GemNet-T | Energy | 5.389 | 7.770 | 0.007 | 1.615 | 9.496 | 0.031 | 2.173 | 21.411 | 959.745 | 994.036 |
| | Force | 0.555 | 0.347 | 0.033 | 0.233 | 0.337 | 0.154 | 0.388 | 0.371 | 0.400 | 1.165 |
| SphereNet | Energy | 0.304 | 0.257 | 0.052 | 0.072 | 0.138 | 0.093 | 0.183 | 0.771 | 20.479 | 12.211 |
| | Force | 0.622 | 0.532 | 0.076 | 0.217 | 0.500 | 0.279 | 0.482 | 2.088 | 0.254 | 0.959 |
| SEGNN | Energy | 15.721 | 3.474 | 0.270 | 0.130 | 0.182 | 1.110 | 4.197 | 1.494 | 0.814 | 1.115 |
| | Force | 8.549 | 2.579 | 0.174 | 0.846 | 1.185 | 0.926 | 3.191 | 2.056 | 1.241 | 0.966 |
| Allegro | Energy | 1.339 | 2.441 | 0.049 | 0.339 | 0.651 | 3.781 | 0.978 | 1.356 | 0.451 | 2.497 |
| | Force | 3.861 | 4.609 | 0.467 | 1.579 | 1.816 | 3.428 | 3.693 | 5.086 | 2.241 | 5.183 |
| Equiformer | Energy | 0.375 | 0.127 | 0.027 | 0.064 | 0.085 | 0.069 | 0.215 | 0.143 | 0.104 | 0.200 |
| | Force | 0.305 | 0.132 | 0.020 | 0.162 | 0.240 | 0.070 | 0.258 | 0.218 | 0.077 | 0.149 |

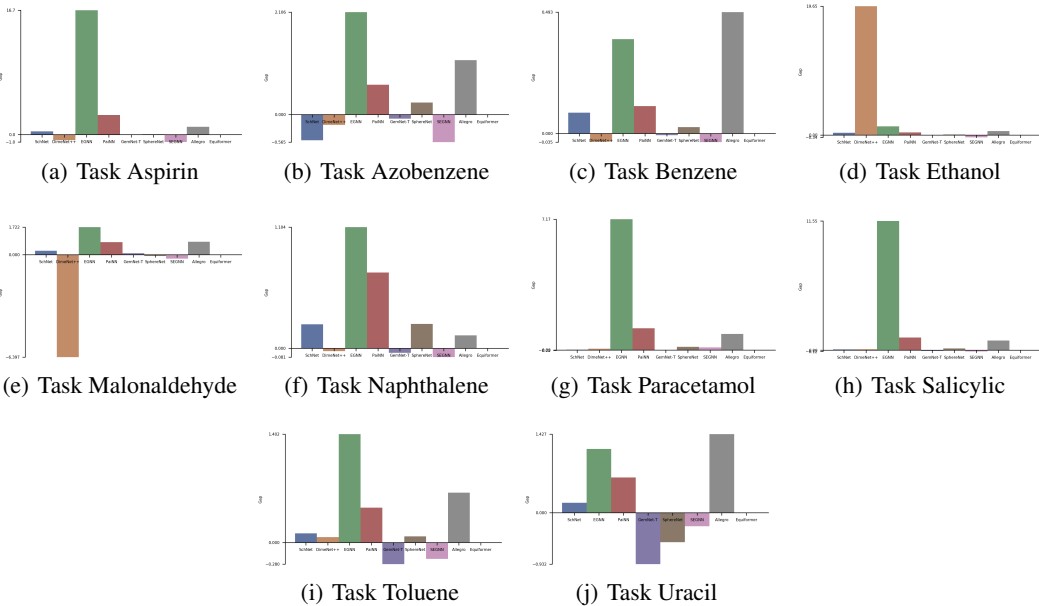

(a) Task Aspirin  (b) Task Azobenzene  (c) Task Benzene  (d) Task Ethanol

(e) Task Malonaldehyde  (f) Task Naphthalene  (g) Task Paracetamol  (h) Task Salicylic

(i) Task Toluene  (j) Task Uracil

Figure 11:  Performance gap of MAE($d = 128$) - MAE($d = 300$) in rMD17.

Table 23: Ablation studies of latent dimension on COLL. 120k for training, 10k for val, 9.48k for test. The evaluation metric is the mean absolute error (MAE).

| Model | $d$ | energy ↓ | force ↓ |
|---|---|---|---|
| SchNet | 128 | 0.171 | 0.135 |
| | 300 | 0.178 | 0.130 |
| DimeNet++ | 128 | 0.049 | 0.058 |
| | 300 | 0.036 | 0.049 |
| EGNN | 128 | 0.786 | 0.151 |
| | 300 | 1.808 | 0.234 |
| PaiNN | 128 | 0.047 | 0.066 |
| | 300 | 0.030 | 0.052 |
| GemNet-T | 128 | 0.022 | 0.035 |
| | 300 | 0.017 | 0.028 |
| SphereNet | 128 | 0.039 | 0.049 |
| | 300 | 0.032 | 0.047 |
| SEGNN | 128 | 7.054 | 0.511 |
| | 300 | 7.085 | 0.642 |
| Equiformer | 128 | 0.034 | 0.030 |
| | 300 | 0.036 | 0.030 |

Table 24: Ablation studies of latent dimension ($d = 128$) on 2 binding affinity prediction tasks. We select three evaluation metrics for LBA: the root mean squared error (RMSD), the Pearson correlation ($R_p$) and the Spearman correlation ($R_S$). LEP is a binary classification task, and we use the area under the curve for receiver operating characteristics (ROC) and precision-recall (PR) for evaluation. We run cross validation with 5 seeds and report the mean and std.

| Model | LBA | | | LEP | |
|---|---|---|---|---|---|
| | RMSD ↓ | $R_P$ ↑ | $R_C$ ↑ | ROC ↑ | PR ↑ |
| SchNet | 1.509 ± 0.05 | 0.510 ± 0.02 | 0.487 ± 0.01 | 0.444 ± 0.03 | 0.391 ± 0.02 |
| DimeNet++ | 1.808 ± 0.46 | 0.557 ± 0.01 | 0.566 ± 0.01 | 0.582 ± 0.06 | 0.494 ± 0.03 |
| EGNN | 1.531 ± 0.02 | 0.452 ± 0.01 | 0.419 ± 0.01 | 0.702 ± 0.05 | 0.603 ± 0.07 |
| PaiNN | 1.460 ± 0.03 | 0.569 ± 0.01 | 0.564 ± 0.01 | 0.627 ± 0.07 | 0.499 ± 0.09 |
| GemNet | 130.621 ± 13.90 | -0.114 ± 0.54 | -0.116 ± 0.55 | 0.623 ± 0.05 | 0.552 ± 0.05 |
| SphereNet | 1.605 ± 0.02 | 0.533 ± 0.00 | 0.527 ± 0.00 | 0.556 ± 0.05 | 0.471 ± 0.05 |
| SEGNN | 1.422 ± 0.04 | 0.560 ± 0.02 | 0.537 ± 0.03 | 0.582 ± 0.08 | 0.517 ± 0.09 |
| Equiformer | 1.490 ± 0.03 | 0.552 ± 0.01 | 0.543 ± 0.01 | 0.626 ± 0.08 | 0.530 ± 0.05 |

Table 25: Ablation studies of latent dimension ($d = 300$) on 2 binding affinity prediction tasks. We select three evaluation metrics for LBA: the root mean squared error (RMSD), the Pearson correlation ($R_p$) and the Spearman correlation ($R_S$). LEP is a binary classification task, and we use the area under the curve for receiver operating characteristics (ROC) and precision-recall (PR) for evaluation. We run cross validation with 5 seeds and report the mean and std.

| Model | LBA | | | LEP | |
|---|---|---|---|---|---|
| | RMSD ↓ | $R_P$ ↑ | $R_C$ ↑ | ROC ↑ | PR ↑ |
| SchNet | 1.521 ± 0.02 | 0.474 ± 0.01 | 0.452 ± 0.01 | 0.450 ± 0.03 | 0.379 ± 0.03 |
| DimeNet++ | 1.672 ± 0.09 | 0.550 ± 0.01 | 0.556 ± 0.01 | 0.590 ± 0.06 | 0.496 ± 0.05 |
| EGNN | 1.494 ± 0.04 | 0.503 ± 0.04 | 0.483 ± 0.05 | 0.657 ± 0.05 | 0.559 ± 0.05 |
| PaiNN | 1.434 ± 0.02 | 0.583 ± 0.02 | 0.580 ± 0.02 | 0.585 ± 0.02 | 0.432 ± 0.03 |
| GemNet | 269.427 ± 148.62 | 0.029 ± 0.50 | 0.036 ± 0.51 | 0.674 ± 0.04 | 0.565 ± 0.05 |
| SphereNet | 1.581 ± 0.02 | 0.538 ± 0.01 | 0.529 ± 0.01 | 0.523 ± 0.04 | 0.432 ± 0.05 |
| SEGNN | 1.416 ± 0.03 | 0.566 ± 0.02 | 0.550 ± 0.02 | 0.574 ± 0.03 | 0.485 ± 0.03 |
| Equiformer | 1.392 ± 0.03 | 0.598 ± 0.02 | 0.578 ± 0.02 | 0.618 ± 0.06 | 0.510 ± 0.05 |

## J.2 Ablation Study on Data Normalization for Molecular Dynamics Prediction

Allegro [95] and NequIP [3] introduce a normalization strategy for molecular dynamics (energy and force) prediction on MD17 and rMD17 datasets:

$$\hat{y}_E = y_E * \text{Force Mean} + \text{Energy Mean} * \# \text{Atom}, \tag{28}$$

where $y_E$ is the original predicted energy, and $\hat{y}_E$ is the normalized prediction. We find this trick important and would like to systematically test it here. Notice that as shown in Appendix J.1, the latent dimension is an important factor, and here we would like to conduct the ablation studies on both factors.

- MD17 w/o normalization and $d = 128$ in Table 19, $d = 300$ in Table 20. rMD17 w/o normalization and $d = 128$ in Table 21, $d = 300$ in Table 22.
- In the following tables, we test: MD17 w/ normalization and $d = 128$ in Table 26, $d = 300$ in Table 27. rMD17 w/ normalization and $d = 128$ in Table 28, $d = 300$ in Table 29.

Table 26: Ablation studies of latent dimension ($d = 128$) on MD17. The evaluation is the mean absolute error. Data normalization is used.

| Model | Energy / Force | Aspirin ↓ | Benzene ↓ | Ethanol ↓ | Malonaldehyde ↓ | Naphthalene ↓ | Salicylic ↓ | Toluene ↓ | Uracil ↓ |
|---|---|---|---|---|---|---|---|---|---|
| SchNet | Energy | 0.588 | 0.099 | 0.072 | 0.111 | 0.125 | 0.207 | 0.110 | 0.118 |
| | Force | 1.008 | 0.200 | 0.297 | 0.491 | 0.299 | 0.547 | 0.346 | 0.383 |
| DimeNet++ | Energy | 0.370 | 0.154 | 28.604 | 57144.066 | 0.289 | 15.497 | 0.206 | 0.317 |
| | Force | 0.578 | 0.110 | 89.512 | 2119653.000 | 0.930 | 90.846 | 0.540 | 0.535 |
| EGNN | Energy | 0.668 | 0.144 | 0.470 | 0.238 | 0.481 | 0.462 | 0.234 | 0.429 |
| | Force | 1.249 | 0.461 | 1.042 | 0.827 | 0.913 | 0.927 | 0.631 | 1.227 |
| PaiNN | Energy | 0.146 | 0.095 | 0.057 | 0.083 | 0.113 | 0.110 | 0.095 | 0.104 |
| | Force | 0.315 | 0.034 | 0.157 | 0.244 | 0.074 | 0.177 | 0.093 | 0.120 |
| GemNet-T | Energy | 0.175 | 0.097 | 0.055 | 0.080 | 0.130 | 0.112 | 0.093 | 0.105 |
| | Force | 0.284 | 0.042 | 0.141 | 0.191 | 0.082 | 0.167 | 0.080 | 0.120 |
| SphereNet | Energy | 0.168 | 0.095 | 0.061 | 0.110 | 0.115 | 0.120 | 0.095 | 0.113 |
| | Force | 0.305 | 0.042 | 0.173 | 0.280 | 0.083 | 0.219 | 0.088 | 0.189 |
| SEGNN | Energy | 0.337 | 0.069 | 0.060 | 0.092 | 0.101 | 0.151 | 0.092 | 0.104 |
| | Force | 0.879 | 0.077 | 0.236 | 0.365 | 0.251 | 0.564 | 0.307 | 0.281 |
| Allegro | Energy | 0.290 | 0.096 | 0.064 | 0.105 | 0.143 | 0.151 | 0.123 | 0.112 |
| | Force | 0.646 | 0.073 | 0.228 | 0.346 | 0.285 | 0.407 | 0.265 | 0.245 |
| Equiformer | Energy | 0.140 | 0.072 | 0.056 | 0.085 | 0.090 | 0.112 | 0.078 | 0.101 |
| | Force | 0.315 | 0.057 | 0.159 | 0.250 | 0.069 | 0.204 | 0.083 | 0.156 |

Table 27: Ablation studies of latent dimension ($d = 300$) on MD17. The evaluation is the mean absolute error. Data normalization is used.

| Model | Energy / Force | Aspirin ↓ | Benzene ↓ | Ethanol ↓ | Malonaldehyde ↓ | Naphthalene ↓ | Salicylic ↓ | Toluene ↓ | Uracil ↓ |
|---|---|---|---|---|---|---|---|---|---|
| SchNet | Energy | 0.321 | 0.099 | 0.074 | 0.125 | 0.129 | 0.155 | 0.130 | 0.171 |
| | Force | 1.055 | 0.191 | 0.318 | 0.522 | 0.328 | 0.597 | 0.387 | 0.401 |
| DimeNet++ | Energy | 0.628 | 56451512.000 | 0.192 | 0.480 | 14056.564 | 0.421 | 27644.078 | 7522.200 |
| | Force | 2.632 | 688219840.000 | 1.029 | 1.703 | 173932.344 | 0.621 | 972773.375 | 16002.980 |
| EGNN | Energy | 0.393 | 0.125 | 0.072 | 0.112 | 0.249 | 0.257 | 0.158 | 0.164 |
| | Force | 0.695 | 0.442 | 0.269 | 0.415 | 0.439 | 0.641 | 0.447 | 0.536 |
| PaiNN | Energy | 0.149 | 0.102 | 0.056 | 0.083 | 0.118 | 0.113 | 0.093 | 0.104 |
| | Force | 0.331 | 0.037 | 0.163 | 0.252 | 0.082 | 0.187 | 0.097 | 0.122 |
| GemNet-T | Energy | 0.162 | 0.142 | 0.068 | 0.089 | 0.136 | 0.115 | 0.095 | 0.106 |
| | Force | 0.329 | 0.052 | 0.206 | 0.262 | 0.101 | 0.234 | 0.091 | 0.146 |
| SphereNet | Energy | 0.212 | 0.096 | 0.081 | 0.101 | 0.116 | 0.145 | 0.099 | 0.120 |
| | Force | 0.334 | 0.047 | 0.177 | 0.309 | 0.087 | 0.238 | 0.097 | 0.212 |
| SEGNN | Energy | 0.345 | 0.069 | 0.072 | 0.097 | 0.096 | 0.354 | 0.093 | 0.110 |
| | Force | 1.023 | 0.080 | 0.331 | 0.452 | 0.227 | 0.803 | 0.314 | 0.327 |
| Allegro | Energy | 0.256 | 0.096 | 0.060 | 0.088 | 0.131 | 0.139 | 0.114 | 0.110 |
| | Force | 0.579 | 0.064 | 0.198 | 0.292 | 0.233 | 0.349 | 0.233 | 0.216 |
| Equiformer | Energy | 0.143 | 0.073 | 0.061 | 0.085 | 0.090 | 0.107 | 0.077 | 0.100 |
| | Force | 0.315 | 0.058 | 0.158 | 0.251 | 0.069 | 0.204 | 0.083 | 0.156 |

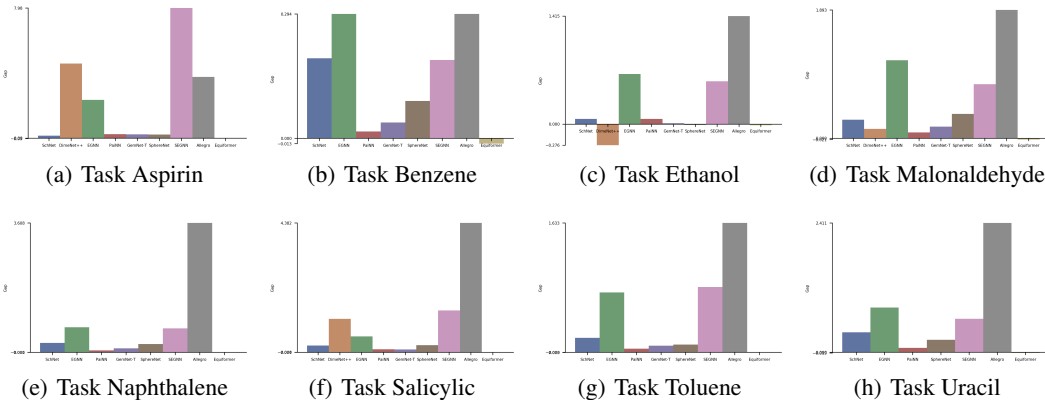

(a) Task Aspirin    (b) Task Benzene    (c) Task Ethanol    (d) Task Malonaldehyde

(e) Task Naphthalene    (f) Task Salicylic    (g) Task Toluene    (h) Task Uracil

Figure 12: Performance gap of MAE(force prediction, $d = 300$ and w/o normalization) - MAE(force prediction, $d = 300$ and w/ normalization) in MD17.

Table 28: Ablation studies of latent dimension (dim=128) on rMD17. The evaluation is the mean absolute error. Data normalization is used.

| Model | Energy / Force | Aspirin ↓ | Azobenzene ↓ | Benzene ↓ | Ethanol ↓ | Malonaldehyde ↓ | Naphthalene ↓ | Paracetamol ↓ | Salicylic ↓ | Toluene ↓ | Uracil ↓ |
|---|---|---|---|---|---|---|---|---|---|---|---|
| SchNet | Energy | 0.702 | 0.583 | 0.013 | 0.055 | 0.079 | 0.058 | 0.179 | 0.103 | 0.064 | 0.054 |
|  | Force | 1.028 | 0.780 | 0.137 | 0.311 | 0.488 | 0.314 | 0.757 | 0.578 | 0.371 | 0.373 |
| DimeNet++ | Energy | 0.321 | 0.560 | 0.050 | 0.093 | 0.142 | 0.157 | 0.299 | 0.353 | 0.143 | 0.318 |
|  | Force | 0.536 | 0.424 | 0.102 | 0.284 | 0.420 | 0.239 | 0.447 | 0.639 | 0.231 | 0.341 |
| EGNN | Energy | 0.653 | 0.684 | 0.056 | 0.275 | 0.238 | 0.440 | 0.476 | 0.514 | 0.233 | 0.395 |
|  | Force | 1.102 | 1.003 | 0.275 | 0.939 | 0.955 | 0.826 | 0.971 | 0.911 | 0.560 | 1.031 |
| PaiNN | Energy | 0.187 | 0.076 | 0.006 | 0.046 | 0.076 | 0.048 | 0.109 | 0.063 | 0.033 | 0.040 |
|  | Force | 0.551 | 0.260 | 0.035 | 0.282 | 0.396 | 0.151 | 0.407 | 0.328 | 0.177 | 0.238 |
| GemNet-T | Energy | 0.116 | 0.058 | 0.002 | 0.038 | 0.078 | 0.018 | 0.082 | 0.047 | 0.017 | 0.023 |
|  | Force | 0.329 | 0.198 | 0.020 | 0.179 | 0.328 | 0.094 | 0.267 | 0.226 | 0.090 | 0.155 |
| SphereNet | Energy | 0.124 | 0.069 | 0.019 | 0.039 | 0.074 | 0.040 | 0.096 | 0.063 | 0.042 | 0.061 |
|  | Force | 0.325 | 0.189 | 0.028 | 0.174 | 0.282 | 0.091 | 0.265 | 0.226 | 0.095 | 0.191 |
| SEGNN | Energy | 0.509 | 0.171 | 0.005 | 0.039 | 0.056 | 0.052 | 0.194 | 0.150 | 0.080 | 0.045 |
|  | Force | 1.129 | 0.603 | 0.054 | 0.279 | 0.394 | 0.254 | 0.792 | 0.682 | 0.365 | 0.327 |
| Allegro | Energy | 0.348 | 0.183 | 0.005 | 0.046 | 0.081 | 0.094 | 0.190 | 0.131 | 0.080 | 0.046 |
|  | Force | 0.673 | 0.385 | 0.039 | 0.249 | 0.371 | 0.289 | 0.476 | 0.430 | 0.270 | 0.254 |
| Equiformer | Energy | 0.106 | 0.044 | 0.002 | 0.030 | 0.038 | 0.016 | 0.112 | 0.050 | 0.021 | 0.025 |
|  | Force | 0.321 | 0.134 | 0.026 | 0.183 | 0.264 | 0.070 | 0.284 | 0.222 | 0.079 | 0.164 |

Table 29: Ablation studies of latent dimension (dim=300) on rMD17. The evaluation is the mean absolute error. Data normalization is used.

| Model | Energy / Force | Aspirin ↓ | Azobenzene ↓ | Benzene ↓ | Ethanol ↓ | Malonaldehyde ↓ | Naphthalene ↓ | Paracetamol ↓ | Salicylic ↓ | Toluene ↓ | Uracil ↓ |
|---|---|---|---|---|---|---|---|---|---|---|---|
| SchNet | Energy | 0.556 | 0.482 | 0.013 | 0.059 | 0.107 | 0.067 | 0.218 | 0.122 | 0.119 | 0.064 |
|  | Force | 1.115 | 0.824 | 0.094 | 0.338 | 0.536 | 0.349 | 0.783 | 0.636 | 0.397 | 0.391 |
| DimeNet++ | Energy | 0.339 | 0.257 | 10.026 | 0.118 | 0.201 | 0.135 | 0.550 | 0.213 | 0.156 | 1.382 |
|  | Force | 0.588 | 0.456 | 378.561 | 0.313 | 0.453 | 0.263 | 0.493 | 0.601 | 0.262 | 4.510 |
| EGNN | Energy | 0.455 | 0.522 | 0.048 | 0.070 | 0.068 | 0.212 | 0.313 | 0.233 | 0.359 | 0.150 |
|  | Force | 0.738 | 0.720 | 0.234 | 0.314 | 0.391 | 0.515 | 0.684 | 0.618 | 0.682 | 0.603 |
| PaiNN | Energy | 0.127 | 0.056 | 0.002 | 0.037 | 0.056 | 0.017 | 0.078 | 0.044 | 0.022 | 0.024 |
|  | Force | 0.443 | 0.183 | 0.019 | 0.237 | 0.331 | 0.095 | 0.331 | 0.248 | 0.126 | 0.171 |
| GemNet-T | Energy | 0.116 | 0.058 | 0.002 | 0.038 | 0.078 | 0.018 | 0.082 | 0.047 | 0.017 | 0.023 |
|  | Force | 0.329 | 0.198 | 0.020 | 0.179 | 0.328 | 0.094 | 0.267 | 0.226 | 0.090 | 0.155 |
| SphereNet | Energy | 0.132 | 0.087 | 0.010 | 0.048 | 0.123 | 0.027 | 0.101 | 0.079 | 0.027 | 0.066 |
|  | Force | 0.348 | 0.203 | 0.023 | 0.194 | 0.315 | 0.090 | 0.283 | 0.248 | 0.094 | 0.206 |
| SEGNN | Energy | 0.570 | 0.300 | 0.005 | 0.037 | 0.064 | 0.061 | 0.283 | 0.210 | 0.096 | 0.062 |
|  | Force | 1.313 | 0.732 | 0.055 | 0.264 | 0.499 | 0.285 | 1.003 | 0.782 | 0.346 | 0.410 |
| Allegro | Energy | 0.294 | 0.167 | 0.004 | 0.043 | 0.056 | 0.070 | 0.170 | 0.093 | 0.063 | 0.037 |
|  | Force | 0.597 | 0.347 | 0.034 | 0.212 | 0.312 | 0.237 | 0.435 | 0.367 | 0.233 | 0.217 |
| Equiformer | Energy | 0.101 | 0.044 | 0.002 | 0.030 | 0.041 | 0.016 | 0.090 | 0.045 | 0.020 | 0.024 |
|  | Force | 0.321 | 0.134 | 0.026 | 0.180 | 0.265 | 0.070 | 0.284 | 0.223 | 0.079 | 0.164 |

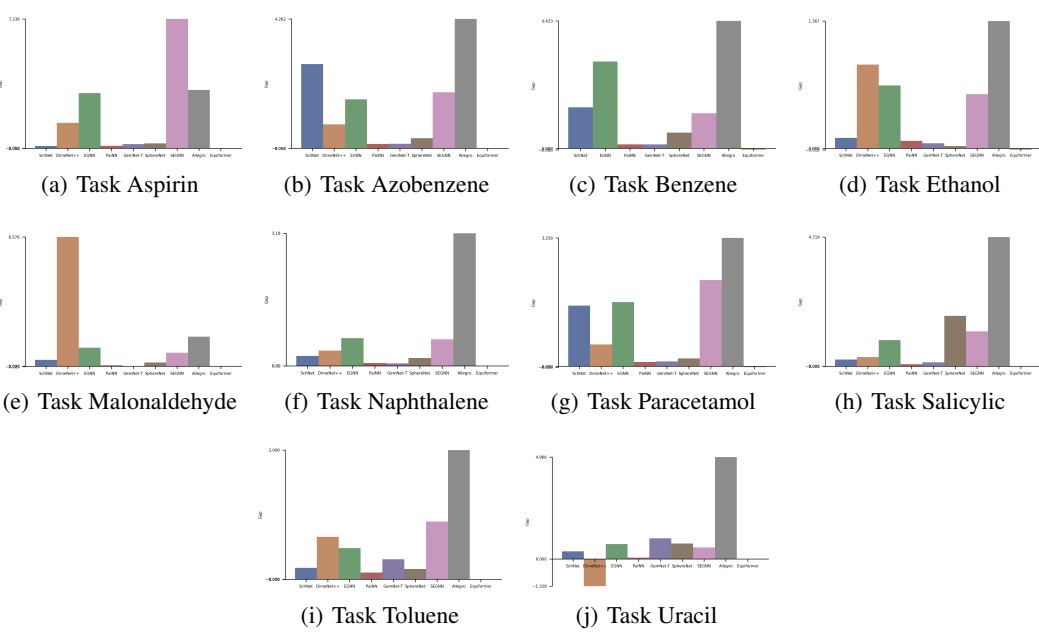

(a) Task Aspirin    (b) Task Azobenzene    (c) Task Benzene    (d) Task Ethanol

(e) Task Malonaldehyde    (f) Task Naphthalene    (g) Task Paracetamol    (h) Task Salicylic

(i) Task Toluene    (j) Task Uracil

Figure 13: Performance gap of MAE(force prediction, $d = 300$ and w/o normalization) - MAE(force prediction, $d = 300$ and w/ normalization) in rMD17.

## J.3 Ablation Studies on Reproduced Results of NequIP and Allegro

Here we would like to further discuss NequIP and Allegro.

- NequIP has no explicit molecule-level representation, and we directly put its results below.
- Allegro adopts $d = 512$ by default (by far we are mainly checking $d = 128$ and $d = 300$).
- We can reproduce NequIP and Allegro results w/ data normalization, as shown below.

Table 30: Ablation study of data normalization on NequIP and Allegro on MD17. The evaluation is the mean absolute error. Here Allegro uses $d = 512$, and both NequIP and Allegro can match the reported results [3, 95] w/ normalization.

| Model | Normalization | Energy / Force | Aspirin ↓ | Benzene ↓ | Ethanol ↓ | Malonaldehyde ↓ | Naphthalene ↓ | Salicylic ↓ | Toluene ↓ | Uracil ↓ |
|---|---|---|---|---|---|---|---|---|---|---|
| NequIP | w/o Normalization | Energy | 8.333 | 0.355 | 0.971 | 2.293 | 1.032 | 2.952 | 1.303 | 1.266 |
| | | Force | 23.769 | 2.383 | 5.832 | 12.099 | 5.247 | 14.048 | 6.800 | 8.060 |
| | w/ Normalization | Energy | 0.175 | 0.095 | 0.058 | 0.089 | 0.114 | 0.114 | 0.094 | 0.105 |
| | | Force | 0.383 | 0.039 | 0.195 | 0.294 | 0.091 | 0.212 | 0.106 | 0.136 |
| Allegro | w/o Normalization | Energy | 1.138 | 0.154 | 0.258 | 1.330 | 0.824 | 1.114 | 0.441 | 0.613 |
| | | Force | 3.405 | 0.823 | 1.412 | 4.191 | 3.743 | 4.934 | 1.968 | 3.544 |
| | w/ Normalization | Energy | 0.240 | 0.096 | 0.058 | 0.085 | 0.128 | 0.130 | 0.107 | 0.107 |
| | | Force | 0.553 | 0.058 | 0.179 | 0.259 | 0.207 | 0.311 | 0.203 | 0.184 |

Table 31: Ablation study of data normalization on NequIP and Allegro on rMD17. The evaluation is the mean absolute error. Here Allegro uses $d = 512$.

| Model | Normalization | Energy / Force | Aspirin ↓ | Azobenzene ↓ | Benzene ↓ | Ethanol ↓ | Malonaldehyde ↓ | Naphthalene ↓ | Paracetamol ↓ | Salicylic ↓ | Toluene ↓ | Uracil ↓ |
|---|---|---|---|---|---|---|---|---|---|---|---|---|
| NequIP | w/o Normalization | Energy | 9.618 | 1.993 | 3.048 | 0.936 | 2.313 | 2.089 | 5.136 | 3.302 | 1.306 | 1.738 |
| | | Force | 22.904 | 6.406 | 1.523 | 6.027 | 12.372 | 5.529 | 17.574 | 15.693 | 7.094 | 10.220 |
| | w/ Normalization | Energy | 0.147 | 0.049 | 0.003 | 0.034 | 0.061 | 0.018 | 0.078 | 0.047 | 0.020 | 0.021 |
| | | Force | 0.407 | 0.176 | 0.019 | 0.218 | 0.310 | 0.092 | 0.308 | 0.230 | 0.113 | 0.142 |
| Allegro | w/o Normalization | Energy | 1.366 | 0.872 | 0.029 | 1.002 | 0.417 | 1.756 | 0.944 | 1.035 | 0.437 | 0.387 |
| | | Force | 3.186 | 2.763 | 0.237 | 2.799 | 2.125 | 3.815 | 3.081 | 4.781 | 2.048 | 1.939 |
| | w/ Normalization | Energy | 0.223 | 0.146 | 0.003 | 0.033 | 0.053 | 0.060 | 0.156 | 0.079 | 0.054 | 0.031 |
| | | Force | 0.558 | 0.308 | 0.029 | 0.198 | 0.264 | 0.207 | 0.409 | 0.331 | 0.210 | 0.187 |

## J.4 Ablation Study on the Data Split of Crystalline Material

In the main paper, we report the results on MatBench with 60%-20%-20% for train-valid-test split. To verify the reproducibility correctness of Geom3D, we carry on an ablation study with the same setting as MatBench [22]. Notice that MatBench adopts the setting in KGCNN [105]: with seed 18012019 and 80% for training and 20% for the test. The reproduced results are in Table 32.

The mean evaluation metrics of SchNet and DimeNet++ with cross-validation are reported in MatBench leaderboard and KGCNN leaderboard, and evaluation metrics of the PaiNN are reported in KGCNN leaderboard.

Table 32: Reproduced results on 8 MatBench tasks.

| Model | Per. $E_{\text{form}}$ ↓ 18,928 | Dielectric ↓ 4,764 | $log_{10}G$ ↓ 10,987 | $log_{10}K$ ↓ 10,987 | $E_{\text{exfo}}$ ↓ 636 | Phonons ↓ 1,265 | Band Gap ↓ 106,113 | $E_{\text{form}}$ ↓ 132,752 |
|---|---|---|---|---|---|---|---|---|
| SchNet (MatBench) | 0.0342 | 0.3277 | 0.0796 | 0.0590 | 42.6637 | 38.9636 | 0.2352 | 0.0218 |
| SchNet (KGCNN) | 0.0347 | 0.3241 | 0.0798 | 0.0584 | 48.0629 | 40.2982 | 0.9351 | 0.0215 |
| SchNet (Geom3D, ours) | 0.035 | 0.334 | 0.080 | 0.060 | 49.363 | 35.172 | 0.226 | 0.023 |
| DimeNet++ (MatBench) | 0.0376 | 0.3400 | 0.0792 | 0.0572 | 49.0243 | 37.4619 | 0.1993 | 0.0235 |
| DimeNet++ (KGCNN) | 0.0373 | 0.3337 | 0.0805 | 0.0579 | 49.2113 | 36.7288 | 0.2089 | 0.0233 |
| DimeNet++ (Geom3D, ours) | 0.033 | 0.340 | 0.080 | 0.060 | 47.700 | 33.564 | 0.207 | 0.022 |
| PaiNN (KGCNN) | 0.0456 | 0.3587 | 0.0851 | 0.0646 | 50.5886 | 47.2212 | 0.2220 | 0.0244 |
| PaiNN (Geom3D, ours) | 0.033 | 0.323 | 0.081 | 0.053 | 42.325 | 38.859 | 0.192 | 0.022 |

## J.5 Ablation Study on the Data Augmentation of Crystalline Material

The default latent dimension $d = 300$ for most of the models, except for EGNN and SEGNN, which lead to the out-of-memory exception.

Table 33: Ablation study on data augmentation (DA) on MatBench and QMOF.

| Model | DA | MatBench | | | | | | | | QMOF |
| | | Per. $E_{\text{form}} \downarrow$ 18,928 | Dielectric $\downarrow$ 4,764 | $log_{10}G \downarrow$ 10,987 | $log_{10}K \downarrow$ 10,987 | $E_{\text{exfo}} \downarrow$ 636 | Phonons $\downarrow$ 1,265 | Band Gap $\downarrow$ 106,113 | $E_{\text{form}} \downarrow$ 132,752 | Band Gap $\downarrow$ 20,425 |
|---|---|---|---|---|---|---|---|---|---|---|
| SchNet | gathered | 0.040 | 0.334 | 0.081 | 0.060 | 65.201 | 42.586 | 0.327 | 0.026 | 0.236 |
| | expanded | 0.048 | 0.338 | 0.086 | 0.066 | 62.991 | 46.301 | 0.253 | 0.042 | 0.278 |
| DimeNet++ | gathered | 0.037 | 0.357 | 0.081 | 0.058 | 68.685 | 38.339 | 0.208 | 0.025 | 0.234 |
| | expanded | 0.042 | 0.334 | 0.088 | 0.064 | 69.579 | 45.223 | 0.235 | 0.041 | 0.243 |
| EGNN | gathered | 0.407 | 0.329 | 0.128 | 0.088 | 76.247 | 87.201 | 0.304 | 0.097 | 0.483 |
| | expanded | 0.038 | 0.331 | 0.087 | 0.064 | 78.015 | 74.846 | 0.211 | 0.026 | 0.256 |
| PaiNN | gathered | 0.038 | 0.317 | 0.080 | 0.053 | 67.752 | 44.602 | 0.190 | 0.022 | 0.207 |
| | expanded | 0.038 | 0.327 | 0.083 | 0.056 | 73.224 | 59.930 | 0.203 | 0.029 | 0.229 |
| GemNet-T | gathered | 0.042 | 0.325 | 0.088 | 0.061 | 68.425 | 48.986 | 0.186 | 0.026 | 0.207 |
| | expanded | 0.042 | 0.364 | 0.090 | 0.063 | 68.376 | 57.316 | 0.195 | 0.036 | 0.230 |
| SphereNet | gathered | 0.043 | 0.388 | 0.087 | 0.061 | 72.987 | 36.300 | 0.217 | 0.029 | 0.251 |
| | expanded | 0.047 | 0.359 | 0.090 | 0.062 | 69.267 | 49.401 | 0.233 | 0.039 | 0.268 |
| SEGNN | gathered | 0.073 | 0.334 | 0.126 | 0.089 | 69.534 | 95.438 | 0.508 | 0.127 | 0.492 |
| | expanded | 0.046 | 0.360 | 0.087 | 0.059 | 65.052 | 43.638 | 0.330 | 0.047 | 0.330 |
| Equiformer | gathered | 0.046 | 0.280 | 0.087 | 0.057 | 62.977 | 37.381 | 0.202 | 0.027 | 0.234 |
| | expanded | 0.047 | 0.314 | 0.086 | 0.061 | 69.845 | 54.087 | 0.226 | 0.036 | 0.258 |

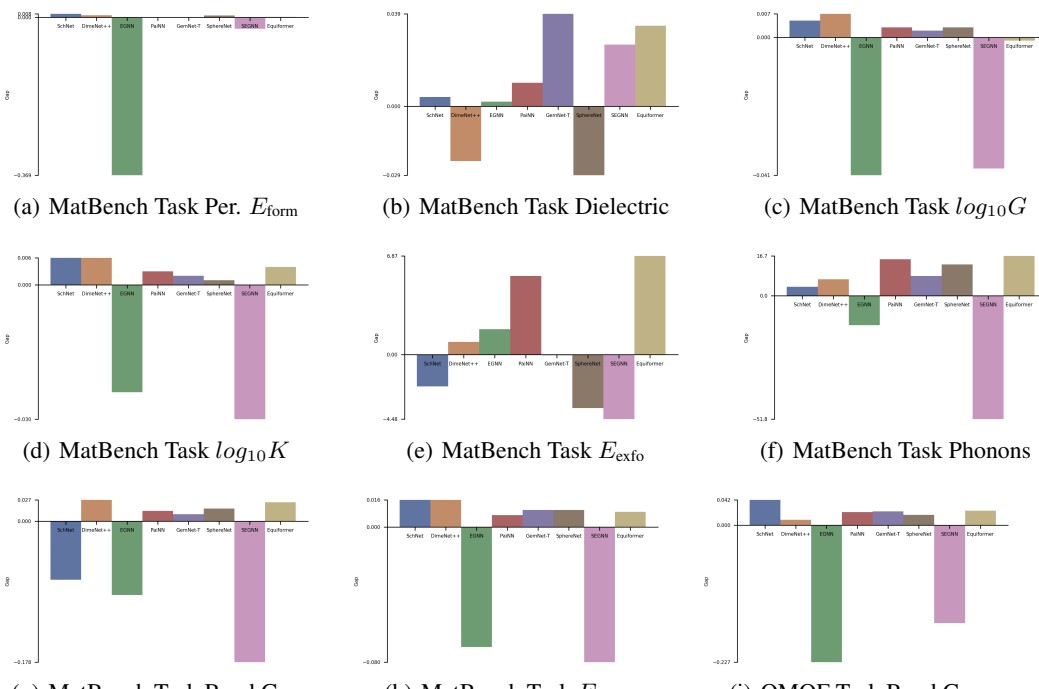

(a) MatBench Task Per. $E_{\text{form}}$     (b) MatBench Task Dielectric     (c) MatBench Task $log_{10}G$

(d) MatBench Task $log_{10}K$     (e) MatBench Task $E_{\text{exfo}}$     (f) MatBench Task Phonons

(g) MatBench Task Band Gap     (h) MatBench Task $E_{\text{form}}$     (i) QMOF Task Band Gap

Figure 14: Performance gap of DA: MAE(expanded DA) - MAE(gathered DA), in MatBench and QMOF.

## J.6 An Evidence Example On The Importance of Atom Types and Atom Coordinates

First, it has been widely acknowledged [28] that the atom positions or molecule shapes are important factors to the quantum properties. Here we carry out an evidence example to empirically verify this. The goal here is to make predictions on 12 quantum properties in QM9.

The molecule geometric data includes two main components as input features: the atom types and atom coordinates. Other key information can be inferred accordingly, including the pairwise distances and torsion angles. We consider corruption on each of the component to empirically test their importance accordingly.

- Atom type corruption. There are in total 118 types of atom types, and the standard embedding option is to apply the one-hot encoding. In the corruption case, we replace all the atom types with a hold-out index, *i.e.*, index 119.
- Atom coordinate corruption. Originally QM9 includes atom coordinates that are in the stable state, and now we replace them with the coordinates generated with MMFF [42] from RDKit [72].

Table 34: An evidence example of molecular data. The goal is to predict 12 quantum properties (regression tasks) of 3D molecules (with 3D coordinates on each atom). The evaluation metric is MAE.

| Model | Mode | $\alpha \downarrow$ | $\nabla \mathcal{E} \downarrow$ | $\mathcal{E}_{\text{HOMO}} \downarrow$ | $\mathcal{E}_{\text{LUMO}} \downarrow$ | $\mu \downarrow$ | $C_v \downarrow$ | $G \downarrow$ | $H \downarrow$ | $R^2 \downarrow$ | $U \downarrow$ | $U_0 \downarrow$ | ZPVE $\downarrow$ |
|---|---|---|---|---|---|---|---|---|---|---|---|---|---|
| SchNet | Stable Geometry | 0.070 | 50.59 | 32.53 | 26.33 | 0.029 | 0.032 | 14.68 | 14.85 | 0.122 | 14.70 | 14.44 | 1.698 |
| | Type Corruption | 0.074 | 52.07 | 33.64 | 26.75 | 0.032 | 0.032 | 21.68 | 22.93 | 0.231 | 23.01 | 22.99 | 1.677 |
| | Coordinate Corruption | 0.265 | 110.59 | 79.92 | 78.59 | 0.422 | 0.113 | 57.07 | 58.92 | 18.649 | 60.71 | 59.32 | 5.151 |
| DimeNet++ | Stable Geometry | 0.046 | 37.41 | 20.89 | 17.54 | 0.030 | 0.023 | 7.89 | 6.71 | 0.310 | 6.74 | 6.94 | 1.193 |
| | Type Corruption | 0.052 | 40.05 | 24.42 | 19.33 | 0.031 | 0.024 | 9.57 | 8.53 | 0.322 | 8.84 | 8.34 | 1.299 |
| | Coordinate Corruption | 0.257 | 202.34 | 88.33 | 167.63 | 0.514 | 0.115 | 77.95 | 628.73 | 19.923 | 72.92 | 804.56 | 5.950 |
| SphereNet | Stable Geometry | 0.048 | 39.98 | 22.69 | 18.98 | 0.026 | 0.027 | 6.95 | 6.95 | 0.234 | 7.33 | 7.34 | 1.620 |
| | Type Corruption | 0.049 | 41.09 | 23.56 | 20.08 | 0.028 | 0.028 | 13.21 | 14.63 | 0.287 | 16.35 | 13.74 | 2.063 |
| | Coordinate Corruption | 0.228 | 100.25 | 69.89 | 70.12 | 0.379 | 0.094 | 52.04 | 56.86 | 17.539 | 55.61 | 55.12 | 4.684 |
| PaiNN | Stable Geometry | 0.048 | 44.50 | 26.00 | 21.11 | 0.016 | 0.025 | 8.31 | 7.67 | 0.132 | 7.77 | 7.89 | 1.322 |
| | Type Corruption | 0.057 | 45.61 | 27.22 | 22.16 | 0.016 | 0.025 | 11.48 | 11.60 | 0.181 | 11.15 | 10.89 | 1.339 |
| | Coordinate Corruption | 0.223 | 108.31 | 73.43 | 72.35 | 0.391 | 0.095 | 48.40 | 51.82 | 16.828 | 51.43 | 48.95 | 4.395 |

We take SchNet and PaiNN as the backbone 3D GNN models, and the results are in Table 34. We can observe that (1) Both corruption examples lead to performance decrease. (2) The atom coordinate corruption may lead to more severe performance decrease than the atom type corruption. To put this into another way is that, when we corrupt the atom types with the same hold-out type, it is equivalently to removing the atom type information. Thus, this can be viewed as using the equilibrium atom coordinates alone, and the property prediction is comparatively robust. This observation can also be supported from the domain perspective. According to the valence bond theory, the atom type information can be implicitly and roughly inferred from the atom coordinates.

Therefore, by combining all the above observations and analysis, one can draw the conclusion that, *for molecule geometry data, the atom coordinates reveal more fundamental information for representation learning*.

## J.7 Ablation on the Effect of Residue Type

As discussed in Sec. 2 and appendix A, proteins have four levels of backbone structures. In Appendix J.6, we carefully check the effect of atom types and atom coordinates in small molecules, and here we would like to check the effect of side residue type in protein geometry-related tasks.

For experiments, we take one of the most recent works, CDConv [29], as the backbone geometric model. The ablation study results are as in Table 35. We observe that the performance drops on all the tasks, and the performance drops on Sup and Fam are much more significant. This reveals that the effect of residue type may differ for different tasks, yet it is preferred to have them encoded for geometric modeling.

Table 35: The effect of residue type on the performance of CDConv.

| Model | Residue Type | EC | Fold | | | |
|---|---|---|---|---|---|---|
| | | | Fold | Sup | Fam | Avg |
| CDConv | w/ residue type | 86.887 | 60.028 | 79.904 | 99.528 | 79.820 |
| CDConv | w/o residue type | 86.144 | 41.783 | 61.164 | 95.598 | 66.182 |

# K  Resources

We use a single GPU (V100 or A100) for each task. Note that we try to run all the models with the same epoch numbers, yet some models are too large in terms of computational memory and time, so we have to reduce the computational time. Thus, we list the running time for the main tasks below for readers to check.

As shown in Tables 36 to 38, in total, it takes over 652 GPU days (without any hyperparameter tuning, random seeds, or ablation studies). It takes at least 1,384 GPU days if we include ablation studies discussed in Appendix J.

We would also like to acknowledge the following nice implementations and tutorials of geometric models:

- e3nn: Euclidean Neural Networks, by Tess [37]
- TFN [112]
- MaterialProject [55] and MatBench [21]
- Keras Graph Convolution Neural Networks (KGCNN) [105]
- DIG [105]
- TorchDrug [145]

Table 36: Running time for each (model, task, epoch) per epoch on small molecules and crystal materials. There are eight tasks in MatBench with various dataset sizes, and we take 2 times $E_{form}$ for illustration here. For NequIP and Allegro, as you can find in the GitHub repository, we do tune their hyperparameters on QM9, yet not being able to reproduce the results. So we may as well report their numbers here.

| Model | | QM9 | MD17 | rMD17 | COLL | LBA | LEP | MatBench | QMOF | Total |
|---|---|---|---|---|---|---|---|---|---|---|
| SchNet [109] | epochs | 1,000 | 1,000 | 1,000 | 1000 | 300 | 300 | 1000 | 300 | 9.3 days |
| | time | 36s | 9s | 8s | 46s | 7s | 5s | 77s | 53s | |
| DimeNet++ [68] | epochs | 500 | 800 | 800 | 1000 | 300 | 300 | 300 | 300 | 53.3 days |
| | time | 185s | 200s | 200s | 288s | 58s | 52s | 470s | 45s | |
| SE(3)-Trans [35] | epochs | 100 | – | – | – | – | – | – | – | 24.2 days |
| | time | 1740s | – | – | – | – | – | – | – | |
| EGNN [108] | epochs | 1000 | 1000 | 1000 | 1000 | 300 | 300 | 800 | 300 | 22.5 days |
| | time | 85s | 12s | 12s | 100s | 18s | 14s | 319s | 300s | |
| PaiNN [110] | epochs | 1000 | 1000 | 1000 | 1000 | 300 | 300 | 1000 | 300 | 13.3 days |
| | time | 46s | 8s | 7s | 61s | 12s | 8s | 176s | 150s | |
| GemNet-T [67] | epochs | 1000 | 1000 | 1000 | 1000s | 300 | 300 | 150 | 200 | 56.8 days |
| | time | 273s | 52s | 48s | 412s | 75s | 82s | 600s | 480s | |
| SphereNet [89] | epochs | 1000 | 1000 | 1000 | 300 | 300 | 300 | 300 | 300 | 78.7 days |
| | time | 250s | 185s | 180s | 418s | 14s | 14s | 480s | 340s | |
| SEGNN [4] | epochs | 500 | 800 | 800 | 100 | 300 | 300 | 40 | 60 | 81.7 days |
| | time | 470s | 245s | 234s | 1450s | 370s | 324s | 3500s | 2750s | |
| NequIP [3] | epochs | 1000 | 1000 | 1000 | 300 | 300 | 300 | – | – | 21.2 days |
| | time | 106s | 29s | 27s | 147s | 25s | 15s | – | – | |
| Allegro [95] | epochs | 1000 | 1000 | 1000 | 300 | 300 | 300 | – | – | 22.6 days |
| | time | 133s | 17s | 17s | 131s | 25s | 22s | – | – | |
| Equiformer [73] | epochs | 300 | 1000 | 1000 | 100 | 300 | 300 | 100 | 150 | 58.6 days |
| | time | 739s | 87s | 126s | 660s | 193s | 109s | 1130s | 936s | |

Table 37: Running time for each (model, task, epoch) per epoch on proteins.

| Model | | ECSingle | ECMultiple | Fold | GO-MF | GO-BP | GO-CC | MSP | PSR | Total |
|---|---|---|---|---|---|---|---|---|---|---|
| IEConv [45] | epochs | – | – | 200 | – | – | – | – | – | 0.85 days |
| | time | – | – | 368s | – | – | – | – | – | |
| GVP-GNN [62] | epochs | 300 | 200 | 400 | 200 | 200 | 200 | 300 | 300 | 3.38 days |
| | time | 150s | 86s | 21s | 160s | 133s | 116s | 241s | 224s | |
| GearNet [142] | epochs | 300 | 200 | 400 | 200 | 200 | 200 | – | – | 2.12 days |
| | time | 61s | 62s | 21s | 88s | 239s | 217s | – | – | |
| ProNet [122] | epochs | 400 | 300 | 1000 | 300 | 300 | 300 | 300 | 300 | 2.85 days |
| | time | 60s | 33s | 19s | 57s | 62s | 57s | 217s | 256s | |
| CDConv [29] | epochs | 150 | 200 | 400 | 200 | 200 | 200 | 300 | 300 | 4.86 days |
| | time | 175s | 138s | 104s | 249s | 253s | 251s | 259s | 325s | |

Table 38: Running time for each (pretraining algorithm, dataset, backbone model) per epoch.

| Dataset | | PCQM4Mv2 (w/ SchNet) | PCQM4Mv2 (w/ PaiNN) | Total |
|---|---|---|---|---|
| Supervised | epochs | 100 | 100 | 13.4 days |
| | time | 426s | 560s | |
| Type Prediction | epochs | 100 | 100 | 13.5 days |
| | time | 433s | 572s | |
| Distance Prediction | epochs | 100 | 100 | 13.4 days |
| | time | 403s | 530s | |
| Angle Prediction | epochs | 100 | − | 6.4 days |
| | time | 479s | − | |
| 3D InfoGraph [81] | epochs | 100 | 100 | 13.5 days |
| | time | 448s | 592s | |
| GraphMVP [86] | epochs | 100 | 100 | 14.0 days |
| | time | 701s | 754s | |
| 3D InfoMax [86, 114] | epochs | 100 | 100 | 13.5 days |
| | time | 493s | 584s | |
| GeoSSL-RR [81] | epochs | 100 | 100 | 14.2 days |
| | time | 680s | 924s | |
| GeoSSL-EBM-NCE [81] | epochs | 100 | 100 | 14.2 days |
| | time | 630s | 980s | |
| GeoSSL-InfoNCE [81] | epochs | 100 | 100 | 14.1 days |
| | time | 598s | 952s | |
| GeoSSL-DDM [81] | epochs | 100 | 100 | 15.0 days |
| | time | 1100s | 1200s | |
| GeoSSL-DDM-1L [136] | epochs | 100 | 100 | 14.4 days |
| | time | 780s | 1010s | |
| 3D-EMGP [59] | epochs | − | 100 | 7.6 days |
| | time | − | 980s | |
| MoleculeSDE-VE [79] | epochs | 50 | 50 | 14.5 days |
| | time | 1906s | 1933s | |
| MoleculeSDE-VP [79] | epochs | 50 | 50 | 14.5 days |
| | time | 1906s | 1933s | |