# OpenReview forum: "Symmetry-Informed Geometric Representation for Molecules, Proteins, and Crystalline Materials"
_NeurIPS.cc/2023/Track/Datasets_and_Benchmarks — NeurIPS 2023 Datasets and Benchmarks Poster_

### Official Review · Reviewer_1hwb · 2023-07-04
**Large-scale ablation**

**Rating:** 5
**Confidence:** 2
**Correctness:** I noticed no mistakes.

**Strengths:**

* The framework gives useful insights into the various heuristics that are employed within the studied models, which helps us better understand what makes a model useful. This is a valuable contribution.
* It is useful to have a framework to compare different geometric models in a consistent manner.


**Additional Feedback:**

See above

**Clarity:**

The paper generally reads well, but at times information is missing. E.g. type-0 and type-1 features are discussion, but it is never explained what they are.

**Documentation:**

An installation guide is provided, which seems to contain sufficient detail.

Example scripts are provided, but e.g. no tutorials could be found (perhaps I missed them?). Given that the paper is proposing a software framework for others to incorporate into their work, then I think the amount of documentation discussing the following topics is too limited:
* the software architecture of the framework
* how the framework is to be used
* how the framework should be extended.


**Ethics:**

I do not see ethical issues here.

**Limitations:**

To the best of my abilities, I think the authors discuss limitations. This is, however, a bit outside my area of expertise, so I may be missing something.

**Opportunities For Improvement:**

* The name of the framework, Geom3D, is rather generic. This is also evident from the fact that several existing software packages carry the same name and these are used for quite different purposes than the proposed framework. I strongly recommend picking a more informative name that reflects what the framework is used for.
* The paper is motivated by AI for Science, which is always nice. However, as I read the paper, the main audience is actually researchers working on geometric models, and only to a lesser extent is the paper aimed at scientists looking to leverage AI. I think it's perfectly fine that the framework targets geometric machine learners, but I think the paper should be more upfront about this.
* The bulk of the paper (the first ~5.5 pages) is essentially devoted to background material. While this material can be valuable, the current structure leaves very little room for actual contributions. My recommendation is to trim the background material in order to discuss the framework to a greater extent.


**Relation To Prior Work:**

I think so.

**Summary And Contributions:**

The paper develops a framework, Geom3D, that is used to compare different geometric machine learning models on different tasks related to protein modeling. The paper gathers a collection of existing data sets alongside diverse tasks to be solved. These are then studied across a range of geometric deep learning models to determine when which models perform well. The framework is also useful for ablation studies, and the paper sheds light on which engineering tricks are useful.

---

> ### Author Response · Authors · 2023-08-15
> **Response to Reviewer 1hwb**
>
>
> Thank you for acknowledging our work as insightful and helpful in understanding the geometric model. Regarding the issues you raised concerning platform naming, the central theme of our work, and the organization of the paper, we have taken diligent steps to address these points in the following. We are hoping that these explanations effectively alleviate your concerns, and we greatly value your reconsideration of the evaluation.
>
> **Q: Naming of Geom3D.**
>
> Thank you for raising this. Geom3D stands for *geometric modeling on 3D structures*, which is indeed our main goal, and we also include the word `geometry/geometric` as your next comment. Besides, after searching the term `Geom3D github` on google, the existing platforms [1](https://github.com/J-F-Liu/geom3d), [2](https://github.com/ctralie/Geom3D), [3](https://github.com/jwezorek/Geom3D) are no longer activated or maintained.
>
>
> **Q: Being upfront about geometric machine learning.**
>
> Thank you for raising this, while we are a little confused and would like to double-check with you. We have been actively presenting our platform centering around **geometric modeling**. You can find it in our method naming (Geom3D), title, abstract, and all the other sections. If you can help specifically point out which parts are not clear, we will be very happy to address them.
>
>
> **Q: Contributions of our work.**
>
> Thank you for raising this. We want to politely point out that these pages are not just background materials, but they also include our key contributions.
> 1. The unified and novel aspect of the geometric model is one of the core contributions of our paper. We discussed this in Sec 1 & Sec 3 (lines 39-57, 138-167, and Figure 2). Specifically, the understanding and categorization of symmetry-informed geometric modeling is novel: we categorize existing geometric models into three venues: invariant, spherical frame-based equivariant, and vector frame-based equivariant models.
> 2. The unified and comprehensive platform (data structure and data preprocessing) of small molecules, proteins, and materials is also the core contribution of our paper. We discussed this in lines 58-80, Figure 1, and Figure 4.
>    - We also provide the most concise data structure descriptions of these applications and what features will be used for geometric modeling in Sec 2. This is the only preliminary information that is also critical for readers as a dataset and benchmark paper.
> 4. We also explicitly point out that data preprocessing and optimization tricks are important when running geometric models in Sec 1 (lines 81-89), and more ablation studies on them are in Sec J.
> 5. Additionally, the goal of Geom3D is to serve as building blocks for more advanced applications. For example, the geometric pretraining in Sec 3.4 (lines 169-183) can illustrate this quite well. We also provide a [tutorial](https://github.com/chao1224/Geom3D/blob/main/tutorials/tutorial_04_geometric_pretraining.ipynb), and you can see how users can inject the geometric models in the geometric pretraining algorithm (in Step 6).
>
> To sum up, we acknowledge that Sec 2 (around 0.5 page) is about preliminary information, yet all the other 5 pages actually discuss the insights that we want to deliver to the community, and they are our core contributions.
>
>
> **Q: Clarity on type-0 and type-1**
>
> We have explained them in lines 140-143 and lines 148-149, and we also provided the reference after these terminologies for readers to check. Now we added more explanations in the revised manuscript, Sec C (lines 981-987).
>
> To help you understand them better, we expand into more details as follows. Different orders of spherical harmonics exhibit varying degrees of angular variation across the sphere's surface. This variation refers to how quickly the function changes as you move around different directions on the sphere. Lower-order spherical harmonics have smoother angular patterns with slower rates of change, while higher-order harmonics have more rapid changes in angular directions.
>
>
>
> **Q: Tutorials and documents**
>
> Thank you for raising this point.
> - We have added tutorials of using Geom3D on [customized data](https://github.com/chao1224/Geom3D/blob/main/tutorials/tutorial_01_customized_your_own_data.ipynb), [energy prediction](https://github.com/chao1224/Geom3D/blob/main/tutorials/tutorial_02_QM9_energy_prediction.ipynb), [force prediction](https://github.com/chao1224/Geom3D/blob/main/tutorials/tutorial_03_MD17_energy_force_prediction.ipynb), and [geometric pretraining](https://github.com/chao1224/Geom3D/blob/main/tutorials/tutorial_04_geometric_pretraining.ipynb).
> - We hope you can understand that the documentation may require more effort, and it is impossible to provide detailed documentation during the short rebuttal period. We appreciate your suggestion and will release them in the near future.

---

> > ### Comment · Reviewer_1hwb · 2023-08-15
> > **Thanks**
> >
> > Thanks for the clarifications etc.
> >
> > Regarding the name *Geom3D*, then I would also argue that "geometric modeling on 3D structures" is quite a bit more generic than what the proposed framework achieves. Such a title could equally well describe a tool for finite element modeling or something similar. The word "geometry" means different things to different people and different communities. I understand why the name makes sense to you, but I would still recommend something less generic.
> >
> > Regarding the second question, then I read the introduction of the paper to indicate that the emphasis was on "AI4Science". I was not aiming to be critical of the work with this remark but merely wanted to communicate that I found the introduction of the paper to be aimed at a different community than the proposed tool.
> >
> > For now, I will retain my low confidence score.

---

> > > ### Author Response · Authors · 2023-08-15
> > > **Thank You for the Follow-up**
> > >
> > > Thank you for the prompt reply. We are happy to communicate as well, which is the main goal of the rebuttal. We carefully read your latest comments, and address them as follows.
> > >
> > > 1. We see your point, and we agree that the method naming can be more specific. We will change it to `Symmetric3D` since we are benchmarking the equivariant models on 3D point clouds.
> > >     - We cannot change this now because this may affect the replies to other reviewers (the specific line numbers will change accordingly), and we will do this once we get the first-round replies from all the other reviewers.
> > >     - Thank you for raising the finite element modeling. We assume that you are talking about PDE solvers, and that is indeed another usage of Symmetric3D: all the encoders in Symmetric3D can serve as the building blocks for other applications, including a neural PDE (solving PDE using neural networks).
> > > 2. About the topic of `AI4Science` in Sec 1, thank you for raising this concern.
> > >     - We briefly mentioned this in the first sentence (line 23).
> > >     - After this line (lines 24-89), we stick to the topic of Symmetric3D: geometric modeling on small molecules, proteins, and crystal materials using SE(3)-equivariance.
> > >     - If there's any sentence that you think is too generic or too `AI4Science`, feel free to let us know. We will be happy to explain them further.

---

> > > > ### Author Response · Authors · 2023-08-26
> > > > **Inquiry for Feedback**
> > > >
> > > > Dear Reviewer 1hwb,
> > > >
> > > > We hope this message finds you well.
> > > >
> > > > We would like to begin by expressing our sincere gratitude for your invaluable feedback on our submission. Rest assured, we have carefully considered your insights and have incorporated them into our revised manuscript.
> > > >
> > > > We are committed to addressing all of your concerns comprehensively. If there are additional questions or comments you would like us to consider, we welcome your input. Your expertise is greatly appreciated, and we aim to ensure that all aspects of our work meet the highest standards of clarity and rigor.
> > > >
> > > > We are deeply grateful for the time and effort you have invested in the review process. Any further feedback from you will certainly contribute to enhancing the quality of our work.
> > > >
> > > > Thank you once again for your continued involvement and guidance. We eagerly await your additional feedback.
> > > >
> > > > Warm regards,
> > > >
> > > > Authors

---

### Official Review · Reviewer_XV1e · 2023-07-12
**Benching marking symmetry-informed geometric representation learning methods**

**Rating:** 7
**Confidence:** 5
**Clarity:** In general, it is well written.

**Strengths:**

In the past few years, many geometric representation learning models have been reported. However, it is hard to compare them as their experimental results were not always comparable. This study attempts to solve this problem. The community needs such a work.
Show that it is important to consider data preprocessing and optimization tricks for specific datasets/tasks.

**Additional Feedback:**

None.

**Correctness:**

Regarding the claim: "A unified and novel aspect in understanding symmetry-informed geometric models.". Please elaborate "unified", "novel", and "understanding". Especially, how does this study advances our understanding.

"The geometric models in Geom3D can serve as a building block for exploring extensive ML tasks". Please elaborate how users can achieve this. Does Geom3D simply import the codes of those models or re-organize those codes so that users can use components of those models to build a new model?

**Documentation:**

Can be improved.

**Ethics:**

None.

**Limitations:**

It is not clear how new methods can be fairly compared with those methods tried in this manuscript. Many other existing methods were not included in this benchmarking study. It is not clear how to expand the comparison to them.

No negative societal impact detected.

**Opportunities For Improvement:**

Fonts in figures are too small and are hard to read.
Experimental settings can be better explained. More experiments are needed. Table 1: How was the split decided? How about using random split? Tables 2/3/4: Why didn't compare all models in Table 1 in these experiments?

**Relation To Prior Work:**

Yes.

**Summary And Contributions:**

This manuscript presents a benchmarking study on geometrical representation learning for molecules, proteins and crystalline materials. Experiments were carried out using 16 symmetry-informed geometric representation learning models and 14 geometric pretraining methods on 46 datasets.

---

> ### Author Response · Authors · 2023-08-15
> **Response to Reviewer XV1e (1/2)**
>
> Thank you for acknowledging our work as necessary for the community in providing a comparable setting for geometric experiments and careful examination of the data preprocessing and optimization tricks. We also deeply appreciate you for listing the rigorous concerns. We believe that we have attempted to solve them during the rebuttal and in the revised manuscript. We have listed the details below.
>
> **Q: Fonts too small.**
>
> We have raised the font sizes in Figure 6. If you think the font size of other figures should be increased, feel free to let us know. We will be happy to address them.
>
> **Q: Experimental settings can be better explained.**
>
> Due to the space limitation, we were only able to provide a high-level description of each task in Sec 4, and more detailed explanations on each dataset can be found in Sec B. If there is any statement particularly unclear to you, we are happy to address them further.
>
>
> **Q: More experiments are needed.**
>
> By far, we have included 46 datasets/tasks in Geom3D with 16 geometric models in the main text, and more ablation studies are in SI (Sec J.1 - Sec J.7). We will be happy to discuss if you can provide additional details on what extra experiments should be included here.
>
>
> **Q: Data splitting on QM9.**
>
> The data splitting QM9 is random splitting, with 110K, 10K, and 11K for training, validation, and testing, respectively. Please check Sec B (lines 850-852) for more details.
>
>
> **Q: Compare all models in Table 1 in these experiments in Tables 2/3/4.**
>
> Thank you for noticing such details. We have now added all of them in Tables 1, 2, 3, and 4 in the revised version. The only exception is SE3-Trans, and the reason is that SE3-Trans requires the covalent bond type as the edge input. However, this is only provided in the QM9 datasets with RDKit parsing. The other datasets by default only include the 3D geometry without the covalent bond (2D topology).
>
>
> **Q: Clarification on comparison with other methods.**
>
> Thank you for raising this concern. However, we need to double-check with you on the question to better solve this issue.
>
> - **Q: It is not clear how new methods can be fairly compared with those methods tried in this manuscript.** If you are asking how to add new geometric models for comparison with the models in this manuscript, you can simply accomplish this by replacing the model architecture and without modifying the data preprocessing or optimization tricks. We also provided **three tutorials** ([tutorial on customized data](https://github.com/chao1224/Geom3D/blob/main/tutorials/tutorial_01_customized_your_own_data.ipynb), [tutorial on energy prediction](https://github.com/chao1224/Geom3D/blob/main/tutorials/tutorial_02_QM9_energy_prediction.ipynb) and [tutorial on force prediction](https://github.com/chao1224/Geom3D/blob/main/tutorials/tutorial_03_MD17_energy_force_prediction.ipynb)).
> - **Q: Many other existing methods were not included in this benchmarking study.**  The geometric research line is growing fast, and we are pretty sure there are many geometric models missed in the current version of Geom3D and including all of them is impossible in one single work (especially with the discoveries on the data preprocessing and optimization tricks). However, now we have covered 16 most recent geometric models and 14 geometric pretraining algorithms, which to the best of our knowledge, is the largest platform in the community. Additionally, we are happy to add more models if you can help list them with GitHub links.
> - **Q: It is not clear how to expand the comparison to them.** We may need your help in explaining this question a little further. If you mean how to run the codes or how to add new geometric models, then it can be done following the instructions listed in the above tutorials.

---

> > ### Author Response · Authors · 2023-08-15
> > **Response to Reviewer XV1e (2/2)**
> >
> > **Q: On the unified, novel, and understanding.**
> >
> > Thank you for raising this. This is a good question and actually reveals **the most important conceptual aspect of Geom3D, from the aspect of machine learning and physics**.
> > - First, we want to clarify that this question can be better explained in the realm of **physics and ML**. The whole paragraph from lines 39-57 discusses this.
> > - Then, more concretely, Geom3D provides a unified and novel understanding because:
> >   - Previously, there have been many geometric models, yet they are compared differently in each work. We studied such existing works and **categorized them into three venues**: invariant model, spherical frame basis equivariant model, and vector frame basis equivariant model.
> >   - Besides, we have provided the theoretical differences between these three categories in Sec C, D, E.
> >   - Such categorization is novel and unifies the existing geometric literature in a more organized way, especially for ML and physics learners to enter this field.
> >
> > **Q: Generalize Geom3D as a building block for extensive ML tasks.**
> >
> > We want to clarify that, in this case, Geom3D will serve as a building block for other uses to use.
> > - For example, if people have a new small molecule or crystal material dataset on energy prediction, then they can use Geom3D by (1) simply constructing their own `Dataset` class and (2) running the models as shown in the tutorials above.
> > - Another example is geometric pretraining. Similarly, the (1) first step is to create a `Dataset` class, and the (2) second step is to utilize the existing geometric models in Geom3D with customized pretraining algorithms. We also provided a [pretraining tutorial here](https://github.com/chao1224/Geom3D/blob/main/tutorials/tutorial_04_geometric_pretraining.ipynb), and you can see how you can plug-and-play different geometric models into the GeoSSL pretraining algorithm (in Step 6).
> > - Besides, Geom3D can be easily generalized to arbitrary tasks, like conformation generation. The high-level idea is that Geom3D makes the geometric models as **basic building blocks**, and users can easily **switch** among different models and test which can be better adapted to their own tasks.

---

> > > ### Author Response · Authors · 2023-08-26
> > > **Inquiry for Feedback**
> > >
> > > Dear Reviewer XV1e,
> > >
> > > We hope this message finds you well.
> > >
> > > We would like to begin by expressing our sincere gratitude for your invaluable feedback on our submission. Rest assured, we have carefully considered your insights and have incorporated them into our revised manuscript.
> > >
> > > We are committed to addressing all of your concerns comprehensively. If there are additional questions or comments you would like us to consider, we welcome your input. Your expertise is greatly appreciated, and we aim to ensure that all aspects of our work meet the highest standards of clarity and rigor.
> > >
> > > We are deeply grateful for the time and effort you have invested in the review process. Any further feedback from you will certainly contribute to enhancing the quality of our work.
> > >
> > > Thank you once again for your continued involvement and guidance. We eagerly await your additional feedback.
> > >
> > > Warm regards,
> > >
> > > Authors

---

> > ### Comment · Reviewer_XV1e · 2023-08-29
> >
> > Thanks for the updates. I like the direction this work is heading, and have raised my rating. The readers can benefit if all random seeds for splitting each datasets are reported, along with the mean and std of each model. It can be biased to compare models using just one random split on each dataset. It will also be great if the details of hyperparameter tuning for each model are reported. It requires substantial efforts to run multiple random splits and conduct hyperprameter tuning. I believe that such results should greatly benefit the whole community. I am willing to raise my rating further if the authors have done the above and are ready to include them.

---

> > > ### Author Response · Authors · 2023-08-29
> > > **Thank You**
> > >
> > > Dear Reviewer XV1e,
> > >
> > > We appreciate you liking the direction of our work, and we are glad that we have addressed most of your concerns. We also appreciate your detailed and careful questions on the random seeds and hyperparameters, and we will address them below.
> > >
> > > - About random seeds.
> > >     - For almost all the experiments, the default seed is 42 (in [this line](https://github.com/chao1224/Geom3D/blob/main/examples_3D/config.py#L7)). We put this as the first hyperparameter, and it is fixed during our benchmarking.
> > >     - The only exception is LBA & LEP, where we take the seeds as 12, 22, 32, 42, and 52.
> > >     - Benchmarking with multiple seeds is the ideal case, which we totally agree. However, one main limitation is the computational resources. Even in the current manuscript, benchmarking these datasets and models took **1372 GPU days and two-year effort in writing the codes and having the jobs run on the cluster**. So if we do five-seed running, that would be roughly **6.8K GPU days and ten-year efforts on our side**, which seems impossible to finish in a short period. We would appreciate it if you could understand our obstacles here.
> > >     - Meanwhile, we also hope that once we release this platform, larger groups with more resources can help build this. This is also another goal to release our package and may require effort from the whole community. We are open to collaborating on our side.
> > > - About hyperparameters.
> > >     - We have listed the hyperparameters for all the models on task QM9 [here](https://github.com/chao1224/Geom3D/tree/main/scripts/QM9). We will release the detailed hyperparameters for other tasks soon.
> > >     - Yet, we also want to highlight that, during our benchmarking, all the important hyperparameters (including the hyperparameters for optimization and data processing tricks) are discussed in Sec J.
> > > - We cherish the value of reproducibility in the ML community, and we promise that we will release all the hyperparameters and checkpoints to the huggingface.
> > >
> > > BTW. We will add the detailed points above in the revised version. We cannot change this now because this may affect the replies to other reviewers (the specific line numbers will change accordingly).

---

### Official Review · Reviewer_pq7Y · 2023-07-20
**Initial review**

**Rating:** 6
**Confidence:** 3
**Correctness:** The claims made in the submission see…

**Strengths:**

The main strength of the work lies in the great variety of models and datasets that the authors have brought under a common framework. It takes a substantial amount of effort to unify such a large number of disparate architectures, and having these available for the research community to test for their own applications and datasets should be immensely helpful.

**Additional Feedback:**

1. Line 168: It is unclear what the authors mean by "higher-order particles."
2. Consistently making use of highlighting optimal values in tables would be ideal; it feels weird to have it only in Table 7.

**Clarity:**

The clarity of the paper could be improved in many areas; in particular, I think that many explanations of the non-machine-learning terms (such as the discussion of potential energy surfaces in line 92, discussing tertiary protein structure around line 104, and the discussion of various types of internal and free energies around line 205) are presented in a way that is more unhelpful than helpful for non-domain-expert readers.

**Documentation:**

Including documentation on the architecture of the project would be good to help users keep track of what is available in the project as it evolves. The example scripts are helpful, and there are often docstrings pointing to references or implementation sources, but I think that having every dataset and architecture documented in a navigatable way with references to source paper(s) and implementation(s) should be a high priority.

**Ethics:**

I have no ethical concerns for this work.

**Limitations:**

The limitations of the work are adequately addressed.

**Opportunities For Improvement:**

It would be good to clarify how much development effort was performed by the authors specifically for this project and how much was sourced from open-source releases of other developers; for example, if no adaptations were needed for the nequip class of models, it seems preferable to me to have the nequip package be a dependency and import its implementation directly, or link a particular version of the nequip repository as a git submodule.

Maybe it is inevitable simply based on the breadth of input domains in this area, but looking at the example code, it seems that the user still needs to perform a significant amount of dataset- and model-specific preprocessing and reshaping. Having the models available in the first place is very useful, but I wonder if some additional abstractions could help make it a more painless process to try a variety of architectures more easily, or swap out datasets for the same architecture(s).

On another note, I applaud the calculation of errors over random replicas in Table 4, but it seems strange to include them in only one table. It would be great to include a more thorough treatment of randomization for all the data in the paper, but I don't want to discourage it happening in the one place.

**Relation To Prior Work:**

The authors present a broad overview of various applications of deep learning to the domains of chemistry, materials science, and biology, but it would be good to include some discussion of the various other model/dataset abstraction projects that exist (for example, the open catalyst project contains implementations of several different equivariant models much like Geom3D does).

**Summary And Contributions:**

This paper showcases the authors' efforts to build a unified model and data library for deep learning on three-dimensional systems commonly found in chemistry, materials science, and biology. This type of work which integrates multiple methods and datasets and exposes them in a common framework is very helpful for end-users and new method developers, so I believe that this paper would be overall beneficial to accept.

---

> ### Author Response · Authors · 2023-08-15
> **Response to Reviewer pq7Y (1/2)**
>
>
> Thank you for acknowledging our work as a unified and common framework valuable for the research community, especially in extending to more diverse tasks. You list certain constructive comments, and we also appreciate you for acknowledging them as open questions. They are indeed challenging and may require efforts from the whole community. Yet, we are happy to discuss with you and share our insights. We hope that we have adequately addressed your concerns.
>
> **Q: Clarify on the development effort.**
>
> Thank you for raising this question. We would like to illustrate the development effort of our work in the following aspects.
>
> - **Reproducibility.** The main development efforts lie in reproducibility, especially in a unified framework where we have **the same data preprocessing steps and optimizers**. Certain models inject data preprocessing steps into the modeling (e.g., adding data normalization) and advanced optimizers, and we separate these modules to examine the effectiveness of each geometric model better. In other words, we **first separate and decompose the key modules in each model**, then we run **comprehensive tuning to test the effect of each of these modules**.
> - **Data preprocessing.** The dataset preprocessing can be domain-specific, e.g., the QM9 needs certain unit transformations. For protein datasets like EC and FOLD, we need special care on the missing positions. For crystal material datasets, the preprocessing is even more challenging, and we elaborated on them in Sec A.3. Plus, another challenge is that different platforms may have different preprocessing steps. We carefully check and compare them (e.g., the effect of two data augmentations), and unify all these preprocessing steps into Geom3D.
> - **Unified dataloader class.** Once we unify the data preprocessing steps in the PyG Dataset class, we also need to update the Dataloader accordingly. For example, the LEP dataset and GemNet model have very special data structures that need to be handled particularly in the collate function.
> - **Optimization analysis.** Optimization is another important module in Geom3D. For example, the number of training epochs is very important, and the performance of SchNet has been long underestimated because it only runs for 300 epochs, while more advanced models are running for 1K epochs. Similarly, for other options like optimizers, we added a more detailed discussion in Sec J (lines 1308-1313).
> - For the example in NequIP, it is performing quite well on MD17 tasks. When we dig further, we find that the NequIP has a key module for utilizing data normalization. Then we apply this to all models, and all of them get improved performance. We elaborate on them in Sec J.2 - J.3. Thus, in Geom3D, we study further on what makes NequIP outstanding, and **we believe that, compared to reaching SOTA performance, such insights can also be valuable for the community.**
> - To sum up, **in addition to getting a good performance**, what we are interested in Geom3D is a more elaborate comparison among geometric models and correspondence effect of data preprocessing and optimization modules, to **better understand the geometric modeling.** And all the work that have been done in this work is centered around this fundamental goal.
>
> **Q: Significant amount of dataset- and model-specific preprocessing and reshaping**
>
> Thank you for raising this question, and we appreciate you also acknowledging this question as inevitablely simple. We are happy to discuss it with you as follows.
> - The Geom3D serves as a platform, and users can easily try their customized dataset with the following supports:
>     - Supportive functions for transforming the small molecules, proteins, and crystal materials into PyG graph (a specific data structure in PyTorch).
>     - The Dataset class on all the datasets mentioned above.
>     - The Dataloader class is specific for certain datasets (LEP) and models (GemNet).
>     - A unified API for 16 advanced geometric models.
>     - Besides, we have also provide  [tutorial](https://github.com/chao1224/Geom3D/blob/main/tutorials/tutorial_01_customized_your_own_data.ipynb) on how to run models on customized datasets.
>
> Specifically for the datasets used in Geom3D, we just pushed both the raw and processed dataset in [this HuggingFace link](https://huggingface.co/datasets/chao1224/Geom3D_data) for reproducibility.
>
>
> **Q: Huge computational cost for error bars.**
>
> Thank you for raising this question. The main reason is due to the huge computational resources. As shown in Table 36, that training all these results (without any hyperparameter tuning or random seeds) take 641 GPU days. If we repeat that for 5 random seeds for each task, that would take around **3.2K GPU days**, which is impossible for us to afford. This is also the main reason why most of the existing geometric models, including SchNet, EGNN, PaiNN, SphereNet, etc, do not consider providing error bars.

---

> > ### Author Response · Authors · 2023-08-15
> > **Response to Reviewer pq7Y (2/2)**
> >
> > **Q: Clarity.**
> >
> > Thank you for raising this concern. Some of the terminologies are actually from math/physics. Thus, it is non-trivial to explain them in plain text without the equations. In Geom3D, we have tried our best to explain these terminologies, and we are happy to explain more in detail in the revised manuscript and summarize them as below:
> > - The **potential energy surfaces (PES)** describe the dynamics of molecules. What we care about most is the molecules at the equilibrium/stable state, and such stable geometries are called conformations. We illustrated them in Figure 3(b): the blue curve is the PES, and the red and yellow regions stand for the conformations.
> > - About the protein tertiary structure, we have a detailed description in Sec B.2. We also revised them in Sec 2 (lines 105-112) in the latest manuscript.
> > - About the discussions on internal and gree energies on QM9, we have them revised in lines 202-203.
> >
> > We understand that such domain-specific information is non-trivial for ML experts, and we have tried our best to explain them in plain text and figures (especially Figure 3) and provide references. If there are any parts confusing to you, please feel free to let us know. We will be happy to explain in more detail.
> >
> >
> > **Q: Relation to prior work.**
> >
> > Thank you for mentioning this. Actually, we have also briefly mentioned this in lines 316-318. The [OCP](https://github.com/Open-Catalyst-Project/ocp) is definitely a large community and has also done a lot in geometric modeling. As shown in the above link, we share about 7 geometric models, yet there are two main differences:
> > 1. OCP includes **1 extra 2D model and 5 extra 3D models**; Geom3D includes **7 1D models, 11 2D models, 12 extra 3D models, and 14 pretraining algorithms**.
> > 2. OCP focuses on few but large-scale catalyst tasks, while Geom3D provides a platform for 46 diverse tasks, covering small molecules, proteins, ligand-pocket binding, and materials.
> >
> > To sum up, we have credited OCP and other great work that contributed to the community. We will be happy to include more based on your suggestions.
> >
> >
> > **Q: Documentation.**
> >
> > Thank you for mentioning this. We acknowledge that documentation is helpful for building up the community. We are working on this now, yet we hope you can understand that crafting robust documentation during the rebuttal phase poses a significant challenge.
> >
> > Meanwhile, we have provided four tutorials on using Geom3D on [customized data](https://github.com/chao1224/Geom3D/blob/main/tutorials/tutorial_01_customized_your_own_data.ipynb), [energy prediction](https://github.com/chao1224/Geom3D/blob/main/tutorials/tutorial_02_QM9_energy_prediction.ipynb), [force prediction](https://github.com/chao1224/Geom3D/blob/main/tutorials/tutorial_03_MD17_energy_force_prediction.ipynb), and [geometric pretraining](https://github.com/chao1224/Geom3D/blob/main/tutorials/tutorial_04_geometric_pretraining.ipynb). We hope these tutorials can also be helpful for users in understanding our goal.
> >
> > **Q: Additional feedback.**
> >
> > Thank you for reading such details. We have them fixed as follows:
> > - The higher-order particles are physical terms, which may not be easily explained in ML terms. In the previous manuscript, we listed [95] for reference. We also added explanations in Sec C (lines 981-987) in the revised manuscript. Here we would like to expand on them in more detail.
> >     - Different orders of spherical harmonics exhibit varying degrees of angular variation across the sphere's surface. This variation refers to how quickly the function changes as you move around different directions on the sphere. Lower-order spherical harmonics have smoother angular patterns with slower rates of change, while higher-order harmonics have more rapid changes in angular directions.
> > - Thank you for pointing this out. We have highlighted the optimal results for each main table in the revised manuscript.

---

### Official Review · Reviewer_m8T2 · 2023-07-20
**A Comprehensive Benchmark for Representation Learning on Molecules**

**Rating:** 8
**Confidence:** 5
**Correctness:** Yes.
**Clarity:** Yes.

**Strengths:**

1. provide a novel overview perspective on existing geometric models as symmetry-informed and classify them into three categories
2. comprehensive over the area of molecular geometric representation learning, take all small molecule, protein and material into consideration
3. extensive experiments with 16 geometric models and 14 geometric pretraining methods across 46 different tasks. The inclusion of ablation studies on training tricks further enhances the paper's credibility, providing robust evidence to support the findings and insights.
4. build a well-organized and easy-to-use platform with methods, datasets, evaluation, and training tricks, serving as a valuable tool for future researchers. This practical aspect facilitates the replication of experiments and encourages further exploration and development in the field.


**Additional Feedback:**

N/A

**Documentation:**

Yes.

**Limitations:**

Yes, the authors have acknowledged the limitations of this study and plan to address them in future versions of this paper.

**Opportunities For Improvement:**

The filtering methods for these datasets are not explained in the paper. Regarding the protein datasets EC and Fold, the paper does not elaborate on how the authors handle situations where certain atom's structure information is missing, e.g., the coordinates of an alpha carbon atom are missing, and in this case, how did the authors align the structure information with sequential information? This data processing detail could be essential to model performances.

**Relation To Prior Work:**

Yes.

**Summary And Contributions:**

This review paper presents an extensive survey and benchmark for the development in the area of AI for molecular geometric representation learning. The authors offer a platform featuring commonly used public datasets, state-of-the-art methods, training tricks, and evaluation metrics, alongside baseline results. Notably, the paper introduces a novel perspective on symmetry-informed geometric models, classifying them into three categories: invariant models, SE(3)-equivariant models with spherical frame basis, and vector frame basis.

The research involves conducting experiments with 16 geometric models and 14 geometric pretraining methods across 46 tasks. Additionally, ablation studies on training tricks are performed to better understand their impact on performance. The results demonstrate that each model benefits differently from various training tricks, offering valuable insights into enhancing machine learning models for scientific discovery.

The platform provided in this paper is well-organized, user-friendly, and serves as a valuable tool for future researchers. Its potential extends to machine learning researchers exploring scientific problems, as well as researchers in computational chemistry, structural biology, and materials science. The representation techniques highlighted herein can assist researchers in making informed decisions when selecting the most suitable methods for specific applications, ultimately benefiting society at large.

---

> ### Author Response · Authors · 2023-08-15
> **Response to Reviewer m8T2**
>
>
> Thank you for your endorsement of our work! We are glad that you find our work as (1) a novel overview aspect, (2) comprehensive application areas, (3) extensive results, and (4) a well-organized and easy-to-use platform. We appreciate your concerns about data preprocessing and hope that our explanations can adequately address your concerns.
>
> **Q: Details on QM9 filtering.**
>
> Thank you for reading our paper carefully. We have added the explanation in the revised version (lines 850-851). The 3,054 molecules are labeled as `uncharacterized` and `filtered out`  because they are rearranged during geometry optimization ([reference](https://moldis-group.github.io/curatedQM9)). The codes are in [these lines](https://github.com/chao1224/Geom3D/blob/main/Geom3D/datasets/dataset_QM9.py#L213-L215) in the Geom3D GitHub.
>
> **Q: Details on the data preprocessing on proteins.**
>
> Thank you for raising this critical question. Generally, we utilized the cleaned dataset provided by existing works, where there is no key atom structure missing. Yet, there are certain methods to handle the missing coordinates issue. More specifically:
> - The FOLD dataset is reported by [Hou et al. (2018)](https://academic.oup.com/bioinformatics/article/34/8/1295/4708302) and [Hermosilla et al. (2021)](https://openreview.net/pdf?id=l0mSUROpwY).
> - The EC dataset is from [Hermosilla et al. (2021)](https://openreview.net/pdf?id=l0mSUROpwY).
> - These two datasets are present in hdf5 files and neither of them presents the issues of missing $C_\alpha$ coordinates.
> - However, these studies do not provide details on how they would manage missing data when cleaning the datasets. We have emailed the corresponding authors about their dataset cleaning details, but they haven't responded yet.
> - Meanwhile, we would like to point out that there are certain potential solutions to address this issue.  For example, we can discard the missing residue or assign a per-defined coordinate for the $C_\alpha$. CDConv has performed experiments for the latter. As indicated in their [paper](https://openreview.net/pdf?id=P5Z-Zl9XJ7) Appendix P, they managed missing coordinates by defaulting them to a pre-defined value (0,0,0). An ablation study on this methodology was conducted, and Table 12 reveals that CDConv maintains good performance, even when up to 20% of the coordinates are absent.

---

> > ### Comment · Reviewer_m8T2 · 2023-08-28
> > **Thank you for the response**
> >
> > I have read the reply and appreciate the author's reply. Thanks!

---

> > > ### Author Response · Authors · 2023-08-29
> > > **Thank you for the support**
> > >
> > > Hi reviewer m8T2,
> > >
> > > We are glad that your concerns are addressed, and thank you for your kind support of our work!
> > >
> > > Regards,
> > >
> > > Authors

---

### Official Review · Reviewer_8p61 · 2023-07-24
**Comprehensive Geometric Representation Benchmark**

**Rating:** 6
**Confidence:** 4
**Correctness:** Yes.
**Clarity:** Yes.

**Strengths:**

1. The presented framework lays a solid foundation for further research on ML tasks, particularly in the field of deep learning.
2. The pipeline's natural and intuitive design encompasses dataset preprocessing, feature extraction, geometric pretraining, representation, and target tasks.
3. The experiments conducted cover a diverse array of fields and applications, enriching numerous scientific research projects.

**Additional Feedback:**

n/a

**Documentation:**

Yes

**Limitations:**

See weakness.

**Opportunities For Improvement:**

1. The distinction and comparison between the application of symmetry-informed geometric models in different fields lack clarity. Further elaboration and additional experiments are necessary.
2. Ambiguities arise within the experimental section:
   a. The corresponding applications and aspects lack a demonstration of their representativeness in the experiments.
   b. Some of the experimental figures, such as Figure 5, lack clarity.
   c. The framework and logic of the experimental part require better organization, as it currently appears somewhat disorderly.

Overall, the paper presents a valuable contribution with a well-structured framework for SE(3)-equivariant models and their implementations. However, there is room for improvement in clarifying application differences and providing more clarity and organization in the experimental section.

**Relation To Prior Work:**

No

**Summary And Contributions:**

This paper proposes a unified view on SE(3)-equivariant models, accompanied by their implementations. These models serve as essential building blocks for various tasks, including geometric pretraining and conformation generation. The primary contributions of this work lie in establishing a unified platform that spans various scientific domains and providing a versatile framework applicable to a wide range of ML tasks, data preprocessing, and optimization techniques.

---

> ### Author Response · Authors · 2023-08-15
> **Response to Reviewer 8p61**
>
> We appreciate your recognition of our work as a strong foundation in the ML community with diverse applications. Your primary concerns revolve around the clarifications regarding Geom3D, which we have elaborated upon below. We trust that these explanations adequately address the points of concern you raised.
>
> **Q: The distinction and comparison between the application of symmetry-informed geometric models.**
>
> Thank you for raising this question. We have already included related discussion in Sec H, and here we added additional comparison information in Sec A (lines 769-781, lines 787-789, lines 840-841). We summarize the main messages below:
> - The three categories in Figure 2 are from a high-level aspect, and all of them are application-agnostic. In other words, they can be naturally applied to small molecules, proteins, and crystal materials.
> - In terms of the specific modeling, the **small molecules and crystal materials** share the same geometric modeling methods on **atoms**.
> - The **proteins** are more challenging because of the large volume, i.e., thousands of amino acids and over tens of thousands of atoms. Thus, existing geometric models on proteins consider modeling either in the **backbone-level** or **residue-level**. Recall that, as shown in Figure 3, backbone-level and residual-level mean that we model only the most important atoms (e.g., $C_\alpha, C, N$) in proteins.
> - We provided more details on the comparison of each geometric model in Sec H.
>
>
> **Q: Ambiguities arise within the experimental section.**
>
> Thank you for raising this question. Yet, we found this question is not clear and would like to double-check with you as below:
> - We want to double-check with you on the `lack a demonstration of their representativeness in the experiments`. If you mean the best performance of each table should be highlighted, then we have highlighted them in Tables 1, 2, 3, 4, 5, and 6 in the revised version.
> - The Figure 5 caption explains that this figure provides the performance difference of MAE(force pred w/o normalization) - MAE(force pred w/ normalization), and the data normalization is explained in lines 286-301. Perhaps you could provide additional insights into the specific aspect of Figure 5 that remains unclear, if any. We will be very happy to address them.
> - Thank you for raising the concerns on the framework and logic of Sec 4. It would be helpful if you could explicitly identify the particular paragraphs or subsections in Sec 4 that appear confusing to you. We will be very happy to address them once we get your detailed feedback.

---

### Author Response · Authors · 2023-08-15
**General Response**

We thank the reviwers for providing constructive and interesting reviews. Specifically, we appreciate reviewers for acknowleding our work as solid foundation and pipeline for the ML community (reviewer 8p61, m8T2, pq7Y, XV1e, 1hwb), novel perspective on geometric modeling (reviewer m8T2, 1hwb), comprehensive applications and experiments (reviewer 8p61, m8T2, pq7Y, XV1e, 1hwb), and versatile studies on data preprocessing and optimization (reviewer 8p61, m8T2, XV1e, 1hwb).

Additionally here, we would like to provide clarifications and resolutions for common issues that have been raised by the reviewers. By addressing these concerns directly, we aim to enhance the overall quality and understanding of our work.

- We added four tutorials on using Geom3D on [customized data](https://github.com/chao1224/Geom3D/blob/main/tutorials/tutorial_01_customized_your_own_data.ipynb), [energy prediction](https://github.com/chao1224/Geom3D/blob/main/tutorials/tutorial_02_QM9_energy_prediction.ipynb), [force prediction](https://github.com/chao1224/Geom3D/blob/main/tutorials/tutorial_03_MD17_energy_force_prediction.ipynb), and [geometric pretraining](https://github.com/chao1224/Geom3D/blob/main/tutorials/tutorial_04_geometric_pretraining.ipynb).
- We uploaded both the raw data and preprocessed data to [this HuggingFace link](https://huggingface.co/datasets/chao1224/Geom3D_data) for reproducibility. To download the data into your own workspace, please refer to [this instruction](https://github.com/chao1224/Geom3D/tree/main/data) and [this python script](https://github.com/chao1224/Geom3D/blob/main/data/download_data.py).
- We have added 33 additional experiments into Tables 1, 2, 3, and 4 as requested by reviewer XV1e. We want to highlight that our previous observations stay the same. Please check the revised manuscript for more details.
- We carefully revised the manuscript (marked in blue) to incorporate the comments suggested by the reviewers. More concretely:
    - We revised the explanations to protein tertiary structures in lines 105-112.
    - We revised the QM9 descriptions in lines 202-203.
    - We added 33 experiment results in Tables 1, 2, 3, and 4.
    - We scaled Table 6 and Figure 6 to enlarge the font size.
    - We highlighted the best performance (bold and underlined) in Tables 1, 2, 3, 4, 5, and 6.
    - We added more explanations to the data structures of molecules, proteins, and crystal materials in Sec A (lines 769-781, 787-789, 840-841).
    - We added explanations to the QM9 filtering step in Sec B (lines 850-851).
    - We added explanations to type-0, type-1, and higher-order particles in Sec C (lines 981-987).
    - We added explanations to protein geometric models in Sec H (lines 1195-1198, lines 1266-1269).
    - We added explanations to the optimization options in Sec J (lines 1308-1313).

---

> ### Author Response · Authors · 2023-08-21
> **Your Additional Feedback Is Encouraged**
>
> Dear Esteemed Reviewers,
>
> We extend our appreciation for the time and effort you invested in reviewing our paper. Your perceptive comments and recommendations have significantly enhanced the quality and coherence of our work. By carefully addressing each of your questions, we have taken your concerns into account. As we move into the rebuttal phase, we eagerly await your post-rebuttal feedback. Please do not hesitate to share your thoughts if you need further clarification or have any additional points you would like us to discuss. We are committed to ensuring that all your concerns are fully addressed.
>
> Regards,
>
> Authors

---

### Decision · Program_Chairs · 2023-09-22

**Decision:**

Accept (Poster)

**Comment:**

This paper constitutes a substantive contribution to molecular geometric representation learning in AI. The authors offer an exhaustive survey, benchmark, and the Geom3D platform, which integrates numerous public datasets, cutting-edge methods, and evaluative metrics. This consolidated model facilitates deep learning on 3D systems relevant to chemistry, materials science, and biology, and its utility extends across diverse scientific disciplines.

The paper introduces a new viewpoint on symmetry-informed geometric models, categorized into three types, and performs robust experiments with 16 geometric models and 14 geometric pretraining methods across 46 tasks. Inclusion of ablation studies on various training techniques provides essential insights into their performance impacts. The findings of some guidance for method selection for specific applications.

The paper's integration of multiple methods and datasets into a common framework will be useful to method developers and end-users. Given its potential benefits to the wider research community and society, the paper is recommended for acceptance.